



# Dead Sea evaporation by eddy covariance measurements versus aerodynamic, energy budget, Priestley-Taylor, and Penman estimates

Jutta Metzger[1], Manuela Nied[1], Ulrich Corsmeier[1], Jörg Kleffmann[2], and Christoph Kottmeier[1]

[1]Institute of Meteorology and Climate Research, Karlsruhe Institute of Technology (KIT), POB 3640, 76021 Karlsruhe, Germany
[2]Physikalische und Theoretische Chemie, Fakultät Mathematik und Naturwissenschaften, Bergische Universität Wuppertal, 42097 Wuppertal, Germany

*Correspondence to:* J. Metzger (jutta.metzger@kit.edu)

**Abstract.** The Dead Sea water budget is no longer in equilibrium. The lake level decline exceeds $1\,\mathrm{m\,a^{-1}}$ and causes severe environmental problems, such as a shifting of the fresh/saline groundwater interface and climatic changes. As the Dead Sea is a terminal lake, located in an arid environment, evaporation is the key component of the Dead Sea water budget and accounts for the main loss of water. However, the actual amount of evaporation as well as the governing factors are unknown. Therefore, for

the first time, long-term eddy covariance measurements were performed for a period of one year, starting in March 2014. The total annual amount measured at this location was $994\pm81\,\mathrm{mm\,a^{-1}}$. The median daily evaporation rate reaches $4.3\,\mathrm{mm\,d^{-1}}$ in July and only $1.1\,\mathrm{mm\,d^{-1}}$ in December. The wind velocity and vapour pressure deficit were identified as the main governing factors of evaporation throughout the year. Consequently, the local wind systems define the diurnal evaporation cycle. In the evening, strong downslope winds govern the wind field and cause evaporation amounts which are up to $100\,\%$ higher than

during daytime, and also during the night evaporation rates are accelerated compared to daytime evaporation, due to strong northerly along-valley flows. Furthermore, a robust and reliable regression model is presented to calculate sub-daily and multi-day evaporation values with a linear function of wind velocity and vapour pressure deficit. An overall correlation coefficient of 0.8 is achieved and the cross validation results in a prediction error of $4.8\,\%$.

Finally, indirect evaporation approaches were tested for their applicability for the Dead Sea and compared to the measurements.

The aerodynamic approach is applicable for sub-daily and multi-day calculations and attains correlation coefficients between 0.85 and 0.99. For the application of the Bowen-Ratio-Energy-Balance (BREB) method and the Priestley-Taylor method, measurements of the heat storage term are inevitable to calculate evaporation on time scales up to one month. Without the heat storage term, the equations yield strong seasonal biases and over- or underestimate daily evaporation rates by up $100\,\%$. The usage of an empirically gained linear function or a hysteresis model depending on the net radiation to estimate the heat storage

term was not accurate enough to provide reliable evaporation amounts. The Penman equation was adapted to calculate realistic evaporation amounts, by using an empirically gained linear function for the heat storage term. The correlation coefficients are above 0.9, the daily mean difference is only $0.5\,\mathrm{mm\,d^{-1}}$ and the estimated annual amount is within the range of the measurement uncertainties.



In summary, this study provides the first directly measured amounts of Dead Sea evaporation and applicable methods to calculate evaporation.

# 1 Introduction

Since several years, the lake level of the Dead Sea declines by over $1\,\mathrm{m\,a^{-1}}$ (approx. $600-700\cdot10^6\,\mathrm{m^3\,a^{-1}}$), meaning
that the balance of the Dead Sea water budget is no longer sustained. The main water inflow to the Dead Sea is the Jordan river, but through anthropogenic interferences the discharge of the Jordan river into the Dead Sea decreased by $90\,\%$ to $60-400\cdot10^6\,\mathrm{m^3\,a^{-1}}$ (Asmar and Ergenzinger, 2002; Holtzman et al., 2005). Further natural inflow by groundwater discharge and surface runoff is in the range of $235-243\cdot10^6\,\mathrm{m^3\,a^{-1}}$ (Siebert et al., 2014). As the Dead Sea is a terminal lake, no natural outflow exists, but water is withdrawn from the lake for mineral and potash production. The total amount is about $250\cdot10^6\,\mathrm{m^3\,a^{-1}}$ (Lensky et al., 2005). Thus, evaporation has to be the main loss of water from the Dead Sea. Even though evaporation is of particular importance for the Dead Sea water balance, the variation in evaporation estimates is high. The spread of the evaporation estimates ranges from 1.05 to $2\,\mathrm{m\,a^{-1}}$ (Stanhill, 1994; Salameh and El-Naser, 1999), comparable to a volume loss of $700-1400\cdot10^6\,\mathrm{m^3\,a^{-1}}$ (Gavrieli et al., 2006). Evaporation is not only a loss of water from the Dead Sea and an important water balance budget component, the thereby resulting lake level decline also causes severe environmental problems. It influences the adjacent aquifers, their groundwater tables and flow paths (Siebert et al., 2016), and results in a shifting fresh/saline groundwater interface (Yechieli et al., 2006), which is connected to the development of sinkholes (Yechieli et al., 2006; Abelson et al., 2006). Since the 1980s, over 4000 sinkholes have formed at the western shore of the Dead Sea, which affect industrial, agricultural, and environmentally protected areas, leading to a substantial economic loss (Arkin and Gilat, 2000). Furthermore, evaporation influences the climatic conditions through a considerable change of the fraction of land and water surface. The changing fraction of water and land surfaces leads to a changing partitioning of the net radiation into sensible and latent heat flux. This results in a weaker horizontal gradient of the air temperature between the air masses over the water and land surface, resulting in a weaker pressure gradient, and thus weakens the lake breeze. As the lake breeze has an attenuating effect on the diurnal temperature amplitude, and advects humidity towards the land, a weaker lake breeze results in higher maximum temperatures and decreasing humidity in the southern part of the valley (Alpert et al., 1997). Furthermore, it increases the diurnal penetration of the westerly winds into the valley in the afternoon. Alpert et al. (1997) showed that in the 1940, before the lake level and thus the water surface started to decrease, the much stronger easterly lake breeze delayed the penetration of the westerly winds considerably. The changing atmospheric conditions, together with the changing groundwater tables result in a severe dieback of vegetation and the drying up of springs, endangering the unique flora and fauna in the Dead Sea region. Especially the unique fish population around Ein Feshkha (Goren and Ortal, 1999; Lipchin et al., 2009) is affected by the reduced water supply.

In view of these environmental changes, resulting from the lake level decline, more accurate estimates of the Dead Sea evaporation are required (Kottmeier et al., 2016). Previous studies on the Dead Sea evaporation used indirect methods, such as water budget calculations (Salameh, 1996; Salameh and El-Naser, 2000), the energy balance approach (Stanhill, 1994; Lensky



et al., 2005), aerodynamic methods (Salhotra et al., 1985; Oroud, 1994), or the combination of the latter two methods, called combination approach (Calder and Neal, 1984; Asmar and Ergenzinger, 1999; Oroud, 2011). Variations in evaporation estimates between the studies result from assumptions on single water budget components such as groundwater inflow, different lengths of the time series of input variables, different measurement locations, and measurement uncertainties. To minimise

the spread of 1.05 to $2\,\mathrm{m\,a^{-1}}$ (Stanhill, 1994; Salameh and El-Naser, 1999) and reduce uncertainties, direct measurements of the Dead Sea evaporation are required. Furthermore, the governing factors of the Dead Sea evaporation, e.g. wind velocity, vapour pressure deficit, or net radiation, have to be identified, to validate the indirect methods. The eddy covariance technique is the only method to obtain direct evaporation measurements, in high temporal resolution, which can be linked to meteorological variables afterwards. It is considered the most accurate and reliable method to estimate evaporation (Rimmer et al.,

2009). However, it is quite expensive and difficult to perform as it requires highly accurate instruments and their continues maintenance. Various studies using eddy covariance measurements have been conducted around the world and also in Israel. Assouline and Mahrer (1993) measured evaporation from Lake Kinneret and Tanny et al. (2008) measured evaporation from a small reservoir in the North of the Dead Sea using eddy covariance systems. However, as to the authors knowledge, no eddy covariance measurements were performed at the Dead Sea, where the environmental problems are severe. Therefore,

long-term eddy covariance measurements are conducted at the Dead Sea shore to measure the latent and sensible heat fluxes, as well as temperature, humidity, precipitation, radiation, wind speed, and wind direction. The measurement location directly at the shoreline provides flux data from the water surface for onshore wind conditions. The station is complemented by two additional eddy covariance stations in close vicinity to provide additional data from the homogeneous desert land surface and from a vegetated area. The comprehensive data set of the station is analysed in this paper with the following aims: (i) Providing

an applicable method for measuring evaporation from the Dead Sea water surface, using a station located at the shoreline. (ii) Evaluating the actual evaporation rate of the Dead Sea and its diurnal and intra-annual variability, and (iii) evaluating the applicability of the commonly used indirect methods to calculate evaporation from the Dead Sea.

## 2 Measurement site

The Dead Sea is a hypersaline terminal lake, located at the lowest point of the Jordan rift valley. It is surrounded by the

Judean Mountains to the West and the Moab Mountains to the East (Fig. 1 a). Nowadays, the Dead Sea consists of two basins, the northern basin with approximately $600\,\mathrm{km^2}$ and the shallow artificial evaporation ponds in the south with approximately $280\,\mathrm{km^2}$, which are used for potash and mineral production. Since the 1950s, the lake level of the northern basin dropped by over 30 m, from -395 m AMSL to the current -429.9 m AMSL (Givati and Tal, 2016). The southern basin is held on a constant level by pumping water from the northern basin to the south. The area between the lake and the eastern and western mountain

chains is rocky desert. When freshwater springs emerge along the shore line, sufficient water is available for plants to grow. Although the total area of these vegetated areas is very small compared to the area covered by water or desert, these vegetated areas are very important for the diversity of the local ecosystems.

To measure the energy balance components of the water surface, a fully equipped energy balance station (EBS) was installed





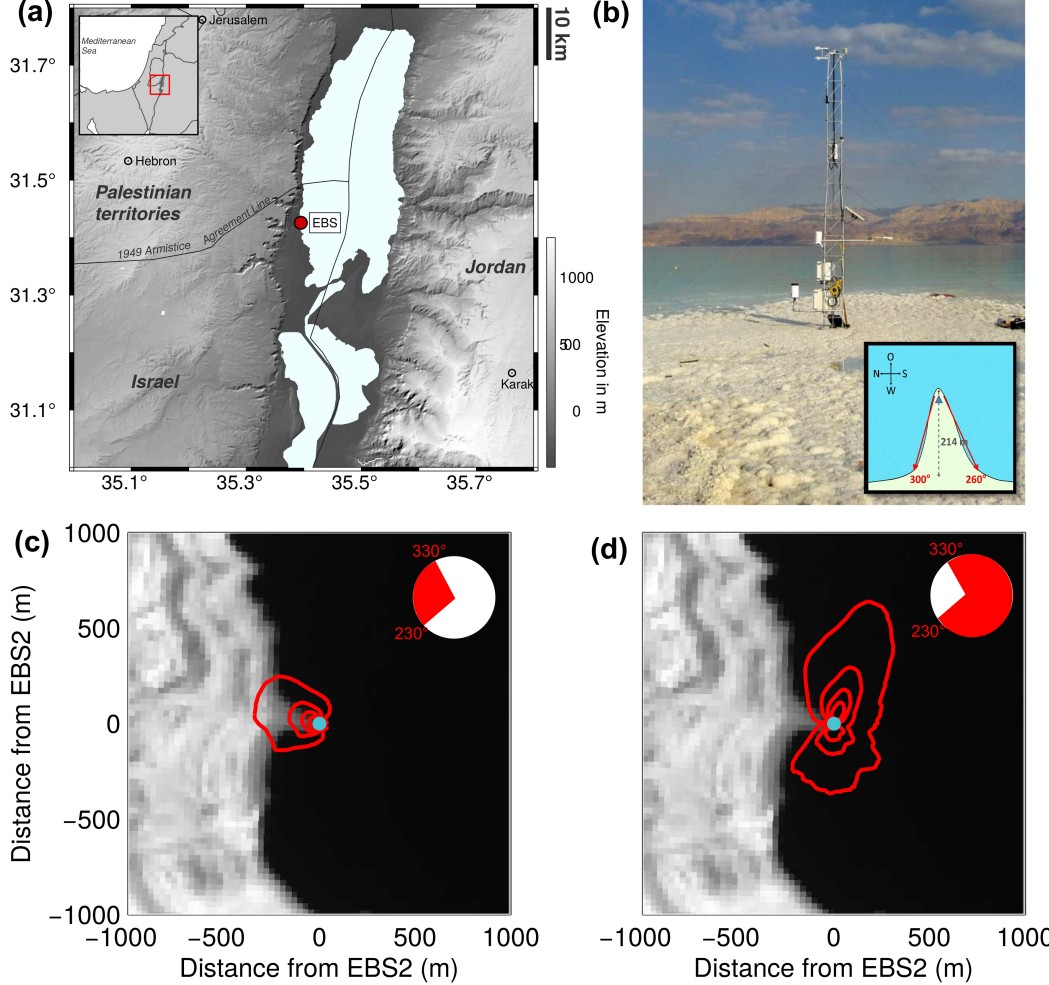

**Figure 1.** Map of the research area and location of the measurement site (a), image of the measurement site and sketch of the headland (inlet) (b), landsat 8 images of the headland with location of the EBS (blue dot), containing the results of the footprint analysis (c,d). Contour lines (from inside to outside) represent 20 %, 40 %, 60 %, and 80 % of the flux footprint area calculated with the footprint model of Kljun et al. (2015) for offshore wind conditions with wind direction between 230° and 330° (c) and for the other wind directions (d). Satellite data provided by the U.S. Geological Survey.

directly at the shoreline (Fig. 1 b). The station, was located 3 km south of Ein Gedi on the tip of a headland at the western shore of the Dead Sea (Fig. 1 a). At the time of the measurements, the station was located at -428 m AMSL, the headland was 214 m long and was surrounded by water from 300° to 260° (insert in Fig. 1 b).



## 3  Data and methods

To achieve the research aims, measurement data from March 2014 until March 2015 were analysed.
At the station the following meteorological variables were measured and averaged over 10 min: temperature and humidity at 2 m height (HC2S3, Rototronics), temperature at 6 m (100KGA1A, BetaTherm), longwave and shortwave radiation components of

the upper and lower half space (CNR4, Kipp&Zonen), precipitation (552202, Young), and atmospheric pressure (PTR330, Vaisala). With a temporal resolution of 20 Hz, water vapour, $CO_2$ concentration, sonic temperature and the three wind components were measured with an integrated gas analyzer and sonic anemometer (IRGASON) from Campbell Scientific at 6 m height.

From the 20 Hz data evaporation was calculated using the eddy covariance technique. The principle of the eddy covariance

theory, the applied post-processing steps and the used objective method for data quality control are presented in the following Sec. 3.1. Furthermore, a multiple regression model was used to calculate evaporation for offshore wind conditions and it was validated using the Monte-Carlo cross validation technique, which is explained in Sec. 3.2. The indirect methods which are evaluated for calculating evaporation from the Dead Sea water surface and the performed sensitivity studies are presented in Sec. 3.3.

### 3.1  Calculation of sensible and latent heat flux

To calculate the sensible and latent heat flux from the wind, temperature, and humidity data measured by the IRGASON, the eddy covariance technique is used. This method uses the fluctuations of the vertical wind velocity and temperature around a temporal mean, here 30 min, to calculate the sensible heat flux:

$$H = c_p \cdot \rho_a \cdot \overline{w'T_a'}, \tag{1}$$

and of the vertical wind velocity and the absolute humidity to calculate the latent heat flux:

$$LE = L_v \cdot \overline{w'a'}. \tag{2}$$

The overbar represents the time average over 30 min, $c_p$ is the specific heat at constant pressure in $\mathrm{J\,K^{-1}\,kg^{-1}}$, $\rho_a$ is the density of the air in $\mathrm{kg\,m^{-3}}$, $w'$ is the deviation of the vertical wind speed from the mean vertical wind speed in $\mathrm{m\,s^{-1}}$, $T_a'$ is the deviation of the temperature from the mean air temperature in K, and $a'$ is the deviation of absolute humidity from the mean

absolute humidity in $\mathrm{kg\,m^{-3}}$. $L_v$ is the latent heat of vaporisation in $\mathrm{kJ\,kg^{-1}}$,

$$L_v = 3148.4 - 2.37 \cdot T_w, \tag{3}$$

which depends on water temperature $T_w$ in K. For salt water $L_v$ increases with increasing salinity (Steiner, 1948). Therefore, for the calculation of the latent heat flux of the Dead Sea water, the salinity of the water has to be considered. To get the dependency of $L_v$ on surface water temperature of the Dead Sea, respective measurements were undertaken. The vapour

pressure of the Dead Sea water was measured as a function of water temperature with a calibrated capacitance manometer (see





Appendix A). The following equation for the Dead Sea water was derived with the same units as in equation 3:

$$L_v = 5150.6561 - 13.9530 \cdot T_w + 0.0162 \cdot T_w^2. \tag{4}$$

As evaporation takes place directly at the water surface of the lake, in a layer of approximately $10\,\mu m$ (Emery et al., 2001), surface water temperature $T_{0m}$ should be used for the calculation of $L_v$. The surface water temperature is not measured directly and is thus calculated using the Monin-Obukhov theory (see Appendix B) and is therefore referred to as $T_{MO}$.

### 3.1.1 Post-processing of eddy covariance data

Post-processing of eddy covariance data is essential as field measurements generally do not fulfil all the theoretical concepts and assumptions of the eddy covariance theory. In particular, measurement limitations, non-stationary conditions over the averaging period, as well as horizontal heterogeneity have to be considered (Foken et al., 2012). Therefore, the following post-processing steps were applied to the data set using the software package TK3 (Mauder and Foken, 2011). First, data were checked on plausibility using individual thresholds for each meteorological variable. Then, a spike detection, using the algorithm after Mauder et al. (2013), was applied. No fluxes were calculated if more than 10 % of the data in the corresponding 30 min interval were missing. To account for a not perfectly levelled sonic anemometer, meaning that the vertical axis is not perpendicular to the surface, and thus the vertical wind measurements are affected by the horizontal wind components, the coordinate system of the sonic anemometer was rotated using the planar fit method after Wilczak et al. (2001). It rotates the coordinate system to the main wind direction and then rotates the system around the y-axis, such that the z-axis is positioned perpendicular to the horizontal plan and that the mean vertical wind is $0\,\mathrm{m\,s^{-1}}$.

The influence of humidity on sonic temperature plays an important role for the calculation of the sensible heat flux. To account for this influence, the Schotanus correction (Schotanus et al., 1983) was applied. This correction is particularly important for flux calculations for sites with high humidity fluctuations, such as over the water surface. The water vapour measurements are influenced by temperature and humidity changes, as only the molar density of water vapour is measured and not the mass mixing ratio. To consider the density fluctuations, corrections after Webb et al. (1980) were applied. Finally, spectral corrections were performed to account for the loss of energy for high frequencies, due to path-length averaging and limited sensor frequency response, following the approach after Mauder and Foken (2011).

### 3.1.2 Quality control and data coverage

The overall performance of the system was very good, and only 2.1 % of the sensible heat flux data and 2.4 % of the latent heat flux data were missing. To assure data quality of the flux measurements, several quality criteria were applied. Latent heat flux data were rejected when the signal strength of the radiation source to measure the water vapour was below 0.5, when the variability of the signal was higher than 0.6 % within a 30 min time interval, and during precipitation events, as a disturbance of the water vapour measurements was expected for these conditions. Due to these quality criteria 10 % of the latent heat flux data were rejected. Further quality control was performed using the steady state test after Foken and Wichura (1996), which analyses each 30 min time interval on stationarity. The integral turbulence characteristics (ITC) test after Foken et al. (2012)





checks data on fully developed turbulent conditions and, therefore, compares modelled ITC to the actually measured ones, and if the deviation is less than 30 % very good data quality is assumed. A combined quality flag considering the steady state test and the ITC test (Foken, 1999) was used to classify the data into nine classes. Class 1 to 6 describe data which can be used for the analysis and classes 7 to 9 were rejected. After the quality control, data availability was 86.3 % for sensible and 78.5 %

for latent heat flux data. Furthermore, the flux footprint has to be considered. A footprint analysis was performed, using the model after Kljun et al. (2015). Results show that flux data for wind directions between 230° and 330° have to be rejected as the fetch is over land, while the aim of this station is to measure evaporation from the water surface (Fig.1 c). For southerly wind directions, the fetch is over water and 80 % of the flux footprint is in the range of 300 m away from the headland (Fig.1 d). For northerly wind directions the fetch was in the range of about 600 m away from the headland. The amount of flux data

rejected due to the footprint was about 19 %. The total available flux data from the water surface was thus 67.1 % for sensible and 59.2 % for latent heat flux. This is reasonable good compared to other eddy covariance studies at other lakes. Jonsson et al. (2008) reported a data availability of 46 % and Mammarella et al. (2015) had a data availability of 63 % for sensible and 53 % for latent heat flux data.

### 3.2   Multiple regression model for the latent heat flux

Through the installation of the EBS at the shoreline flux data from the water surface are only available for onshore wind conditions and all data for offshore wind conditions, i.e. wind directions between 230° and 330°, have to be rejected for further analysis (Fig. 1 c). However, for the analysis of the diurnal and intra-annual variability of the evaporation, estimates of the fluxes for these wind directions are important, as otherwise evaporation in the afternoon, when westerly downslope winds prevail would be missing. Therefore, a multiple regression model is applied to find a suitable relationship between the turbulent

fluxes and governing meteorological variables, such as wind speed, vapour pressure deficit, net radiation, and surface water temperature. Vapour pressure deficit is calculated using the surface water temperature, which is gained by the Monin-Obukhov theory (see Appendix B). A Monte-Carlo cross validation (MCCV), first introduced by Picard and Cook (1984), is performed to test the model's robustness and get an estimate of the model error. The work flow is as follows: (i) data between 230° and 330° are removed from the data set. (ii) Two approaches are used to divide the data in a training and validation data set. The

first approach uses randomly chosen data points of about about 15 % of the total data set as validation data and the second approach uses a randomly chosen wind sector of 45° as validation data. The usage of these two approaches allows the general test of the model on robustness but also its sensitivity on a certain wind sector. (iii) After each division a regression model is build with the training data set and then applied on the data of the validation group. The deviation of the calculated from the measured flux values yield the model error of one realisation. (iv) After multiple applications, in this case 500 times, the model

error is averaged and results in the prediction error of the regression model. A large prediction error indicates a dependency of the model on the choice of the training data set and is therefore rejected.



**Table 1.** Selection of commonly used equations to calculate evaporation ($Ev$). Equations are shown in the form like they are used for the comparison as default versions (V0) in Sec. 4.4.

| Method | Name | Reference | Equation |
|---|---|---|---|
| Aerodynamic | Aerodynamic | Brutsaert (1982) | $Ev = K_E \cdot v_a \cdot (E_w - e_a)$ |
| Energy Budget | BREB (simplified) | Dingman (2002) | $Ev = \frac{Rn}{\rho_w \cdot L_v \cdot (1+\beta)}$ |
| Combination | Penman | Van Bavel (1966) | $Ev = \frac{\Delta \cdot Rn + \gamma \cdot K_E \cdot v_a \cdot \rho_w \cdot L_v \cdot (E_a - e_a)}{\rho_w \cdot L_v \cdot (\Delta + \gamma)}$ |
| Combination | Priestley-Taylor | Priestley (1972) | $Ev = c_{PT} \frac{\Delta \cdot Rn}{\rho_w \cdot L_v \cdot (\Delta + \gamma)}$ |
| | Bowen ratio | | $\beta = \frac{c_p \cdot p}{0.622 \cdot L_v} \cdot \frac{T_{0m} - T_a}{E_w - e_a} = \gamma \cdot \frac{T_{0m} - T_a}{E_w - e_a}$ |
| | Wind function | Brutsaert (1982) | $K_E = \frac{0.622 \cdot \rho_a \cdot \kappa^2}{p \cdot \rho_w \left( \ln(\frac{z_m - z_d}{z_0}) \right)^2}$ |

$c_p$ = specific heat capacity at constant pressure
$c_{PT} = 1.26$ = Priestley-Taylor coefficient
$e_a$ = vapour pressure at air temperature
$E_a$ = saturation vapour pressure at air temperature
$E_w$ = saturation vapour pressure at water surface temperature
$Ev$ = evaporation
$F_n$ = net advected heat flux
$G$ = ground heat flux
$K_E$ = wind function
$L_v$ = latent heat of vaporisation
$p$ = air pressure
$Rn$ = net radiation
$T_{0m}$ = surface water temperature
$T_a$ = air temperature

$v_a$ = wind velocity
$z_m$ = measurement height
$z_0$ = roughness length
$z_d$ = displacement height
$\beta$ = Bowen ratio
$\gamma$ = psychometric constant
$\Delta$ = slope of the saturation vapour pressure versus temperature curve
$\Delta Q$ = heat storage of the lake
$\Delta t$ = time interval
$\kappa = 0.4$ = Kármán constant
$\rho_a$ = air density
$\rho_w$ = water density

## 3.3 Indirect methods to estimate evaporation

Four commonly used indirect methods to estimate evaporation (Table 1) will be evaluated in this paper by comparing their results to the eddy covariance measurements. An aerodynamic approach, the energy budget method, and two combination approaches, namely the Penman equation and the Priestley-Taylor equation. Additionally, sensitivity studies are performed to quantify the influence of simplification within the approaches, which are often made in literature. An overview which sensitivity study is applied for the different methods is given in Table 2.

The first method is an aerodynamic approach after Brutsaert (1982), where only wind speed and vapour pressure deficit are required. Brutsaert (1982) used a logarithmic wind profile and did not consider near surface atmospheric stability (Table 1). A sensitivity study is performed for the aerodynamic approach considering atmospheric stability in the wind function $K_E$, afterwards referred to as V1.

The second method is the energy budget method expressed as the Bowen Ratio Energy Budget (BREB). For this approach several of the input variables are difficult to obtain. The amount of net advected heat into the water body, $F_n$, meaning the heat



**Table 2.** Overview of the sensitivity studies performed for the evaporation equations.

| Version | Explanation | Aerodynamic | BREB | Priestley-Taylor | Penman |
|---|---|---|---|---|---|
| 0 | Default (Table 1) | X | X | X | X |
| 1 | Atmospheric Stability | X | – | – | X |
| 2 | Heat storage term derived with hysteresis model | – | X | X | X |
| 3 | Heat storage term derived as a linear function of $Rn$ from V0 | – | X | X | X |
| 4 | Removal of $T_s$ | – | – | – | X |
| 5 | Removal of $T_s$ and heat storage term with hysteresis model | – | – | – | X |
| 6 | Removal of $T_s$ and heat storage term derived as a fraction of $Rn$ from V4 | – | – | – | X |

advected into the lake by water inflow and precipitation, as well as the loss of heat by water outflow, have to be known. If the in- and outflows are small compared to the size of the water body, or water temperatures are similar the terms can be neglected (Dingman, 2002; Rosenberry et al., 2007). Moreover, the ground heat flux $G$, meaning the heat exchange at the bottom of the lake, is required. It can usually be neglected, as the amount is small compared to the other components (Henderson-Sellers,

1986). Another component difficult to obtain is the heat storage of the lake, $\Delta Q$. It requires measurements of lake temperature at different depths from a raft station or a ship. On longer time scale it can often be neglected, which is used in many studies. Thus, for the default version (V0) of the BREB method the net advected heat, the ground heat flux and the heat storage term are neglected (Table 1). Using this version, only net radiation, surface water temperature, and air temperature have to be known, which are relatively easy to obtain and thus an easy approach to calculate evaporation. Sensitivity analyses for the BREB

method are performed regarding the consideration of the heat storage in the equation. For this purpose $Rn$ is replaced with $(Rn - \Delta Q)$. Duan and Bastiaanssen (2015) proposed a hysteresis model to calculate the heat storage term, depending only on the net radiation. This approach is used in sensitivity version V2. Another approach to account for the heat storage term is the simple assumption that the heat storage is directly proportional to the net radiation and that the deviation of the default version from the measurements equals the heat storage term. This is tested as version 3 (V3).

The third method to calculate evaporation is the combination approach, considering the energy balance and the aerodynamic influence. Priestley (1972) proposed an equation which considers the aerodynamic influence by using an empirically gained coefficient of $c_{PT} = 1.26$ (Table 1). In this equation the heat storage is also not considered, and hence the same sensitivity studies as for the BREB method are performed. Penman (1948) combined the energy balance equation with the aerodynamic approach. In his approach he neglected net advected heat, the ground heat flux, and the heat storage (Table 1). The heat storage

can be considered in the Penman equation by replacing $Rn$ with $(Rn - \Delta Q)$ again. Sensitivity studies V2 and V3 are tested for the Penman equation as well. Furthermore, Kohler and Parmele (1967) presented modified coefficients for $L_v$ and $\gamma$ to eliminate the need of surface water temperature $T_s$ from the equation. This further reduces the amount of necessary input parameters, which makes the equation more easily applicable. The modified parameters are tested in version V4. In version V5 they are applied together with the hysteresis model for the heat storage term. The last sensitivity test (V6) combines the

parameters with a linear function for the heat storage term, derived from V4.



## 4   Results

### 4.1   Meteorological conditions

In the Dead Sea valley the measured average annual air temperature was 26.5°C for the measurement period, which was slightly higher than the long term annual mean of 25.9 °C (Hecht and Gertman, 2003). Maximum daily air temperatures exceeded 40 °C in summer (Fig. 2) and the total precipitation amount was 273 mm. (Fig. 2). The total precipitation amount for the observation period is high compared to the annual precipitation normal of 80 mm (Goldreich, 2003). It resulted from a few heavy precipitation events in January 2015, which makes the observation period 2014/15 a relatively wet year for the area. The wind velocity didn't show a clear annual cycle. From March until October, mean, maximum, and minimum were relatively similar, only during the winter seasons a different behaviour was found (Fig. 2). The relative uniform wind velocities resulted from periodic local wind systems, governing the conditions in the valley. Between sunrise and sunset a lake breeze prevailed, leading to north-easterly winds at the western shore with a median wind velocity of $3\,\mathrm{m\,s^{-1}}$ (Fig. 2 and 3 a). The lake breeze occurred throughout the year, with an occurrence rate of over 70 % of the days in summer 2014, and 58 % and 48 % of the days in spring and autumn 2014, respectively. In winter, the synoptic conditions gained more influence and often superimposed the local wind field such that a north-easterly lake breeze was only observed at about 32 % of the days and a south-easterly flow at 26 % of the days. In the evening, downslope winds, often enhanced by the Mediterranean Sea Breeze (MSB) (Alpert et al., 1997; Naor et al., 2015), lead to accelerated wind velocities in the valley (Fig. 3 b). These downslope winds occurred at about 57 % of the days in summer, and still 28 % of the days in spring and 45 % of the days in autumn. The downslope winds regularly reached mean wind velocities of over $10\,\mathrm{m\,s^{-1}}$ (Fig. 3 b). During the night, a northerly along-valley flow prevailed mainly in spring and summer. The along-valley flow also reached wind velocities of over $10\,\mathrm{m\,s^{-1}}$ (Fig. 3 c). The difference between the saturation vapour pressure at the water surface and the actual vapour pressure of the air ($\Delta e$) had a mean value of 9.75 hPa. It had a clear annual cycle with maximum values above 30 hPa in summer. Individual peaks in winter were related to special synoptic conditions, e.g. at the beginning of November a Red Sea Trough with a central axis advected dry and warm air into the valley over the course of several days. The annual cycles of the energy balance components are also shown in Fig. 2. The net radiation reaches maximum values of over $900\,\mathrm{W\,m^{-2}}$ in summer and about $500\,\mathrm{W\,m^{-2}}$ in winter. The sensible heat flux is small throughout the year. The mean latent heat flux values are higher in summer compared to the winter months. However, at individual days in winter latent heat flux values even exceed the summer values. The heat storage is calculated as the residuum of the energy balance equation ($Rn = LE + H + \Delta Q$). As can be seen a considerable amount of energy is stored, but also released over the course of the day. On a seasonal basis the sensible heat flux accounts for about 5 to 10 % of the net radiation in spring, summer, and autumn, whereas it accounts for nearly 40 % in winter. The latent heat flux accounts for 43 % and 53 % of the net radiation, in spring and summer, leading to a high heat storage amount of 51 % and 42 %, respectively. This energy is used for heating the lake, which is stronger in spring than in summer. In autumn over 74 % of the net radiation is transformed into latent heat flux, such that the heat storage amount is small. In winter, the latent heat flux is in the range of 92 % of the net radiation, meaning that the heat storage term is negative, releasing the heat to the atmosphere, represented through the higher sensible heat flux. Similar behaviour of the flux components was found for other lakes, e.g. Giadrossich et al. (2015).





**Figure 2.** Daily precipitation amounts, 24 h running mean of air temperature, $T_a$, surface water temperature, $T_{MO}$, wind velocity, $v_a$, vapour pressure deficit $\Delta e_{MO}$, net radiation, $Rn$, latent heat flux, $LE$, sensible heat flux, $H$, and heat storage $\Delta Q$. The grey shaded area represents the range between daily minimum and maximum values of the respective variable.





**Figure 3.** Wind conditions between (a) 6:30 and 17:30 LT, (b) 17:30 and 20:30 LT, and (c) 20:30 and 6:30 LT. Data are shown for spring, summer, autumn, and winter 2014/15.

## 4.2 Multiple regression model for the latent heat flux

The footprint model showed that the fetch of the fluxes is over land for wind directions between 230° and 330°. The affected amount of latent heat flux data is 19 %. Through the predominant local wind systems, these wind directions occur almost exclusively in the evening between 17:30 to 20:30 LT from spring until autumn (Fig. 3) and, thus, at most of the days data within this time frame are excluded. For the analysis of the diurnal variability of the latent heat flux from the water surface, and also for the intra-annual and annual amounts, it is important to close these gaps. A multiple regression model was applied to calculate the latent heat flux for offshore wind conditions. The choice of the input variables for the multiple regression model was based on the analysis of the linear correlation between the latent heat flux and different meteorological variables. The correlation coefficients for the variables are shown in Table 3. For the latent heat flux highest correlation is achieved with wind





**Table 3.** Correlation coefficients for latent heat flux ($LE$) with wind speed ($v_a$), net radiation ($Rn$), surface water temperature ($T_{MO}$), and vapour pressure deficit calculated with surface water temperature ($\Delta e_{MO}$). Correlation coefficients over 0.5 are bold. Data are shown for the meteorological seasons 2014/15 and the entire data set.

|        | $v_a$  | $Rn$  | $T_{MO}$ | $\Delta e_{T_{MO}}$ |
|--------|--------|-------|----------|---------------------|
| Spring | **0.68** | -0.19 | 0.07 | 0.06 |
| Summer | **0.72** | -0.16 | 0.00 | -0.12 |
| Autumn | **0.53** | 0.16  | 0.36 | 0.46 |
| Winter | **0.81** | 0.27  | 0.19 | **0.56** |
| Total  | **0.60** | 0.03  | 0.42 | 0.38 |

speed, with correlation coefficients between 0.53 and 0.81 for the different seasons, followed by the vapour pressure deficit, and finally surface water temperature and net radiation. This is different from cooler climates where highest correlation was found with vapour pressured deficit (Blanken et al., 2000; Nordbo et al., 2011), and also from lakes in Mediterranean climate, where vapour pressure deficit had the same impact as wind speed (e.g. Bouin et al. (2012)). The influence of the vapour pressure

deficit varies strongly between the different seasons. In spring and summer no correlation exists between latent heat flux and the vapour pressure deficit, but in autumn, winter, and for the total data set correlation coefficients are between 0.38 and 0.56. Although correlation with individual meteorological variables is already good, none of the variables can fully explain the latent heat flux. A stepwise multiple regression model was applied with the following variables to find the best fitting solution for the latent heat flux:

$$X_{LE} = (v_a, \Delta e_{T_{MO}}, Rn, T_{MO}) \tag{5}$$

The model $X_{LE}$ gave the same dependency for all seasons. The latent heat flux depended on a linear combination of wind speed and vapour pressure deficit. The correlation coefficient ranged from 0.77 in spring and summer to 0.85 in winter (Table 4). The aerodynamic approach to estimate evaporation is based on the product of wind speed and vapour pressure deficit (Table 1), instead of a linear combination. For comparison, the correlation of the product of wind speed and vapour pressure deficit with

the latent heat flux was calculated additionally and resulted in nearly the same correlation coefficients (Table 4). The results of the MCCV analysis reveal that the model $X_{LE}$ results in small prediction errors. The prediction error varies between -0.16 and 2.94 % for randomly chosen data points and for randomly chosen control sectors between 0.02 and 8.61 %. The model with $v_a \cdot \Delta e_{T_{MO}}$ results in higher model errors varying between -0.17 and 4.72 % for randomly chosen data points and between -0.31 and 10.57 % for randomly chosen control sectors. Even though the correlation coefficients are similar for both models, model

$X_{LE}$ was chosen for the calculation of the latent heat flux, instead of the commonly used $\Delta e \cdot v_a$, because of the robustness and the smaller prediction error. The model coefficients are shown in Table 5.

In summary, the regression model $X_{LE}$ provides a suitable and robust method to calculate the latent heat flux for offshore wind conditions. 90 % of the originally rejected latent heat flux data due the fetch criteria could be calculated with the model. The total data availability was thus increased from 59.2 % to 76.8 %. The calculation of the latent heat flux for offshore wind





**Table 4.** Results of the stepwise linear regression model $X_{LE}$ for the latent heat flux. The corresponding correlation coefficient (R) of the model is shown if a variable is added to the model. If a variable is not added to the model, it is indicated with a minus sign. For the model with $v_a \cdot \Delta e_{MO}$, the correlation coefficient (R) is given. The prediction errors yielded by the MCCV with randomly chosen validation data points ($er_r$) and randomly chosen validation sectors ($er_s$) are shown for both models. Results are shown for the meteorological seasons and for the entire data set.

| | $X_{LE}$ | | | | | | $v_a \cdot \Delta e_{T_{MO}}$ | | |
| | $v_a$ | $\Delta e_{T_{MO}}$ | $Rn$ | $T_{MO}$ | $er_r(\%)$ | $er_s(\%)$ | R | $er_r(\%)$ | $er_s(\%)$ |
|---|---|---|---|---|---|---|---|---|---|
| Spring | 0.68 | 0.77 | – | – | 0.32 | 8.61 | 0.79 | -0.01 | 10.57 |
| Summer | 0.73 | 0.77 | – | – | 0.17 | 2.31 | 0.76 | -0.17 | 1.60 |
| Autumn | 0.53 | 0.82 | – | – | -0.16 | 1.25 | 0.84 | 0.42 | 6.17 |
| Winter | 0.81 | 0.85 | – | – | 2.94 | 0.02 | 0.85 | 4.72 | -0.31 |
| Total | 0.59 | 0.80 | – | – | 0.96 | 4.79 | 0.83 | 0.80 | 6.78 |

conditions is especially important for the analysis of the diurnal cycle of the latent heat flux, and also for its intra-annual variation. The comparison of the mean diurnal cycles of the measured fluxes (uncorrected) with the cycles including the calculated values for offshore wind conditions shows that during the day the differences are small (Fig. 4). As the prevailing wind direction is north-east, caused by the lake breeze, nearly no calculations are necessary, as the flux footprint is located over water (Fig. 1 d). However, in the evening, when downslope winds prevail in spring, summer, and autumn, the differences are quite large (Fig. 4). During this time period, the measured values represent the latent heat flux from the land surface, with values around or below $50\,\mathrm{W\,m^{-2}}$. In contrary the calculated values represent the latent heat flux from the water surface, with values up to $200\,\mathrm{W\,m^{-2}}$ in summer. Hence, the regression model allows a detailed analysis of the diurnal cycle of the fluxes, even though the station is located at the shoreline.

## 4.3 Diurnal and annual variability

The latent heat flux is the dominating turbulent flux at the water surface (Fig. 2). It has a strong diurnal cycle. During daytime, the latent heat flux reaches values of $100\,\mathrm{W\,m^{-2}}$ in summer and autumn, and $70\,\mathrm{W\,m^{-2}}$ in spring and winter (Fig. 4). The

**Table 5.** Coefficients of the model equations to calculate latent heat flux ($LE$). The equations have the general form: $LE = a + b \cdot v_a + c \cdot \Delta e$. Coefficients are shown for the meteorological seasons 2014/15 and the entire data set.

| | Spring | Summer | Autumn | Winter | Total |
|---|---|---|---|---|---|
| a | -32.52 | -25.41 | -58.91 | -15.29 | -36.92 |
| b | 13.33 | 18.41 | 16.21 | 11.07 | 14.31 |
| c | 5.51 | 4.61 | 7.56 | 4.46 | 6.13 |



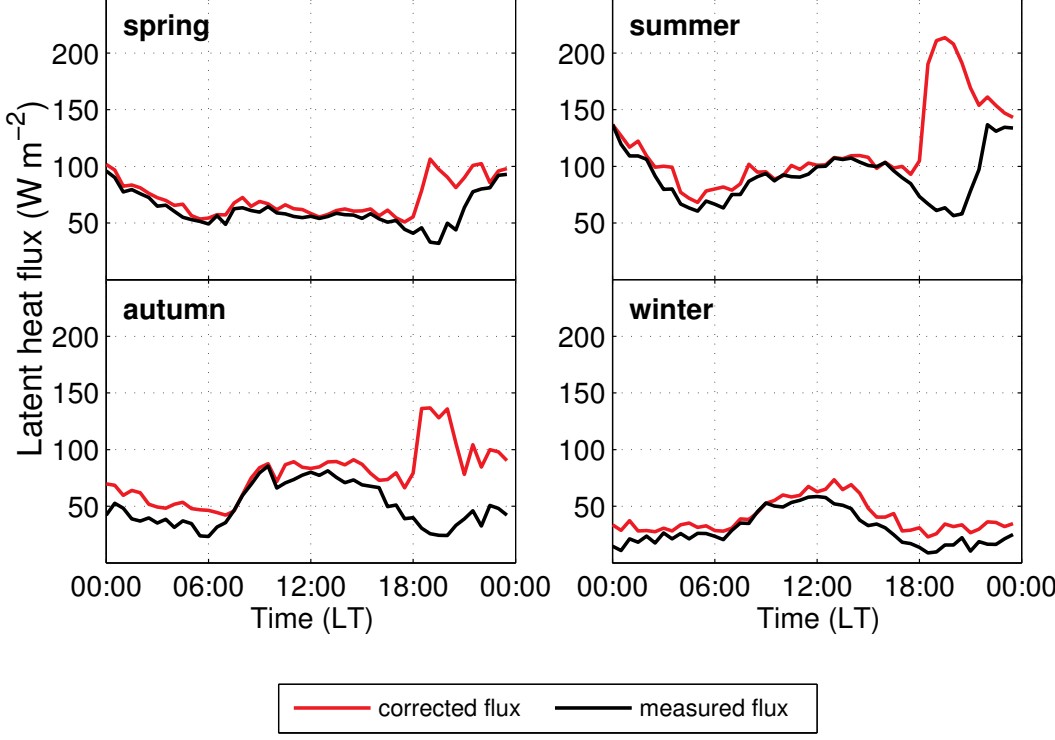

**Figure 4.** Median diurnal cycles of the measured latent heat flux (black lines) and the calculated latent heat flux for the water surface (red lines). Latent heat flux values for wind directions between 230 and 330° are calculated with the multiple regression model.

maximum values are reached after sunset around 19:00 LT in spring, summer, and autumn. In spring about $105 \, W \, m^{-2}$ are reached, in summer $213 \, W \, m^{-2}$, and in autumn $136 \, W \, m^{-2}$. During the night, the latent heat flux continues to be higher than during daytime and reaches minimum values shortly before sunrise. In winter, this late maximum is not observable and values during nighttime are lower than during daytime. The unusual diurnal cycle with highest latent heat flux values after sunset and

5 during the night are clearly connected to the diurnal cycle of wind speed and vapour pressure deficit, and thus to the wind systems. This is most pronounced in summer. During the day, the lake breeze with relatively low wind velocities, (Fig. 3 a), causes moderate latent heat flux rates. The downslope winds in the evening have generally high wind velocities (Fig. 3 b), and advect drier air into the valley, which results in high vapour pressure deficits and thus high latent heat flux values. The high values during night result from accelerated wind velocities (Fig. 3 c), rather than high vapour pressure deficits.

10 For the calculation of daily and yearly evaporation amounts, still existing data gaps were closed, using the median evaporation rate of the corresponding time step of the respective month. The uncertainty due to this gap filling method was estimated using the median absolute deviation (MAD), which is the median of the absolute deviations from the data's median.

In spring, evaporation values steadily increase until a maximum median evaporation rate of $4.3 \, mm \, d^{-1}$ is reached in July (Fig. 5). Afterwards, evaporation values decrease until a minimum median evaporation rate of $1.1 \, mm \, d^{-1}$ is reached in De-





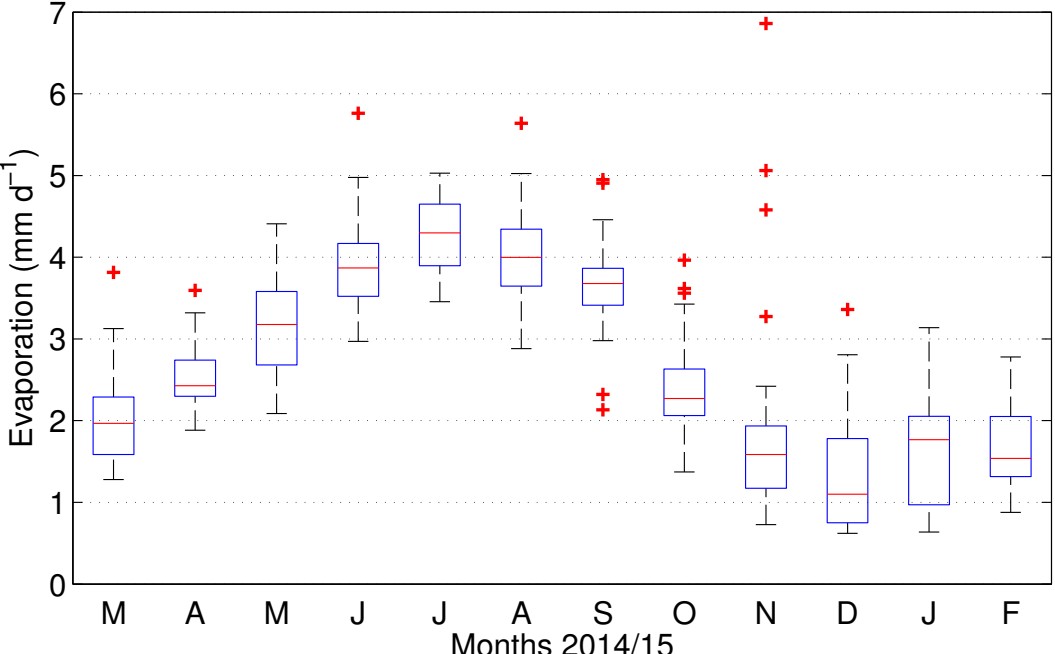

**Figure 5.** Boxplot of daily evaporation rates. Red lines indicate medians, the edges of the boxes are the 25th and 75th percentiles, the whiskers extend to the most extreme data points not considered outliers, and outliers are plotted individually by red crosses.

cember (Fig. 5). The annual cycle of evaporation follows the solar cycle with a time lag of about 1 month. Summing the evaporation values over the whole measurement period results in a total amount of $994.5 \pm 81.2$ mm. Also visible in Fig. 5 is the higher variation of the daily evaporation amounts between November and February. This is the so-called wet season when synoptic patterns gain more influence on the atmospheric conditions in the valley (Bitan, 1974, 1976). The governing factors of

evaporation, i.e. wind speed and vapour pressure deficit, are very variable during this time. On the one hand, winter storms with rain and high air humidity can reach the region, which decreases the evaporation rate. On the other hand, winter storms without rain but high wind velocities, which advect very dry air to the Dead Sea, can significantly increase the evaporation rate (Shafir and Alpert, 2011). The highest variability (not considering outliers) can be seen in January, with daily evaporation amounts between 0.6 and 3.1 mm d$^{-1}$. In November, evaporation values vary between 0.7 and 2.4 mm d$^{-1}$, but on three consecutive days

evaporation rates exceed these values. Evaporation rates of 5.1 mm d$^{-1}$, 6.9 mm d$^{-1}$, and 4.6 mm d$^{-1}$ are measured, which is the absolute maximum of the whole measurement period. These extreme evaporation values are caused by a Red Sea Trough with a central axis and a dominant high to the east, which causes south-easterly winds above the valley. It can be observed that through the complex orography a pressure driven channelling occurs along the valley axis, resulting in a near-surface northerly wind with constantly high averaged wind speed of over 10 m s$^{-1}$ (not shown). This leads to the advection of warm and very

dry air over the lake, which, together with the high wind velocities, increases the evaporation dramatically.



### 4.4 Indirect methods to estimate evaporation

For the calculation of evaporation several equations, based on different physical approaches, exist. Each approach connects evaporation to different meteorological parameters and is designed for different time intervals, ranging from sub-daily calculations to a time interval of at least 7 days. With the comprehensive data set of the measurements, it is possible, for the
first time, to evaluate four of the commonly used equations for their applicability for Dead Sea evaporation on different time scales (30 min, 1 d, 7 d, and 28 d) and perform a sensitivity analysis on simplifications and assumption used for the equations. The main goal is the identification of the best fitting equation, by using measurements purely made on land, as data from raft stations or buoys are often difficult to obtain (Giadrossich et al., 2015). The calculated evaporation amounts are compared to the eddy covariance measurements and evaluated in terms of their correlation coefficient, slope and offset of the regression
line, mean difference and monthly differences between the estimates and the measurements.

The first equation is the aerodynamic approach after Brutsaert (1982) (Table 1) and is afterwards named V0. This equation uses wind speed and vapour pressure deficit as governing factors. The aerodynamic approach is the only approach designed for sub-daily time intervals. The correlation coefficient for 30 min averages is 0.85 and it tends to overestimate evaporation amounts. The slope of the regression line is 1.26 (Table 6) and the mean difference is $0.92\pm0.54\,\mathrm{mm\,d^{-1}}$. For time intervals of
1 d and longer, the aerodynamic approach yields better results. The correlation coefficients vary between 0.94 for 1 d intervals and 0.99 for 28 d intervals, mean differences are smaller, and the slopes of the regression lines vary around 1.10 (Table 6, Fig. 6 a,V0). The mean differences are evenly distributed throughout the year, showing no seasonal bias, and the total annual evaporation amount is well represented (Fig. 7 a,V0). A sensitivity study was performed, considering the near surface stability (V1), using the stability factors after Cline (1997). However, the comparison shows that the inclusion of the stability has a
negligible effect on the daily evaporation amounts (Table 6). The BREB method is first used in the simplified version shown in Table 1, neglecting net advected heat fluxes, the ground heat flux, as well as the heat storage term. With this version (V0), only net radiation, surface water temperature, and air temperature have to be known. These variables are relatively easy to obtain and it would therefore be an easy approach to calculate evaporation. However, neglecting the heat storage term results in an strong bias of the evaporation amounts. The correlation coefficients range from only 0.67 for 1 d time intervals to 0.87
for 28 d intervals, the slope varies from 1.27 to 1.72, respectively, and the larges offset is -1.35 $\mathrm{mm\,d^{-1}}$ (Table 6). This indicates a strong overestimation of high evaporation amounts in spring and summer and an underestimation of small evaporation amounts mainly in winter (Fig. 6 b,V0), resulting in a clear seasonal bias. From April until September daily evaporation rates are overestimated by up to 3 $\mathrm{mm\,d^{-1}}$ and underestimated during the rest of the year (Fig. 7 b,V0). This seasonal bias was also observed in other studies, e.g. Winter et al. (1995); Rosenberry et al. (2007). Compared to the measured values, this results
in a overestimation of the annual evaporation amount by 22 %. The sensitivity study V2, which considers the heat storage of the lake using a hysteresis model, improved the results. Correlation coefficient are better and the mean differences are reduced (Table 6). The intra-annual performance also improved and evaporation estimates between November and April are quite good, however, evaporation rates are underestimated in summer and autumn by about 1 $\mathrm{mm\,d^{-1}}$ (Fig. 7 b,V2). This results in a underestimation of the total evaporation amount by about 11 %. In general, the slopes and the offsets indicate an overestimation of





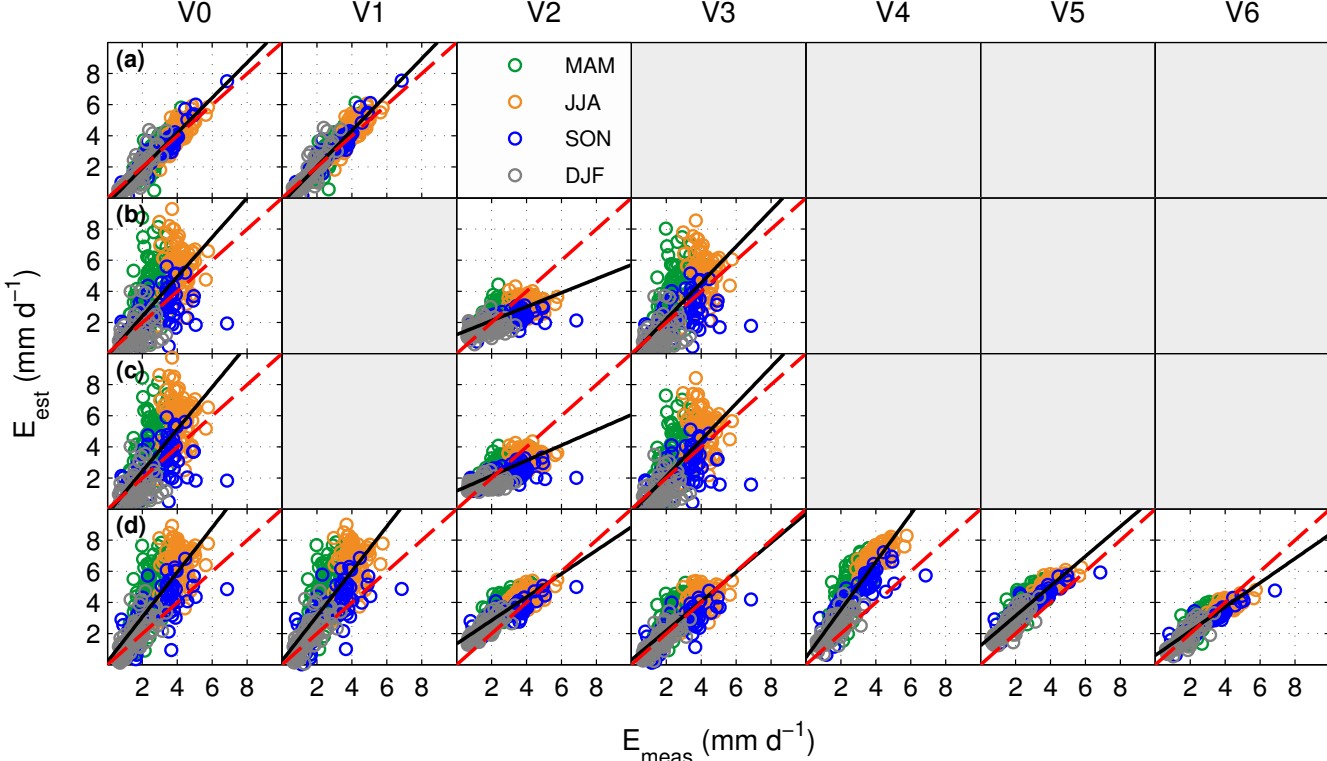

**Figure 6.** Correlation between estimated and measured daily evaporation amounts for (a) the aerodynamic approach, (b) the BREB method, (c) the Priestley-Taylor equation and (d) the Penman equation and their sensitivity studies (Table 2) calculated from 1 d averages. The colours indicate the meteorological seasons spring (MAM), summer (JJA), autumn (SON), and winter (DJF). The regression line is shown in black and the 1:1 line as dashed red line.

the small evaporation amounts and an underestimation of the high amounts (Fig. 6 b,V2). Sensitivity study V3 also accounts for the heat storage term, by using $\Delta Q = 0.08 \cdot Rn$, derived from the deviation of the V0 from the measurements. This approach can only slightly improve the correlation coefficient, slope, offset, and mean difference in comparison to V0 (Table 6). Only the total annual evaporation amount improves compared to the default version and overestimates evaporation by only 13 % instead

5   of 22 % (Fig. 7 b V3).

The Priestley-Taylor equation, as described in (Table 6), results in correlation coefficients between 0.69 for 1 d and 0.89 for a 28 d time interval. Like the BREB equation slopes are too high with values between 1.35 and 1.84 and offsets vary between -0.28 and -1.58 mm d$^{-1}$ (Table 6). By neglecting the heat storage term small evaporation rates are underestimated and large ones overestimated (Fig. 6 c,V0), resulting in a strong seasonal bias and an overestimation of the total evaporation amount by

10   26 % (Fig. 7 c,V0). Sensitivity test V2 yields similar results as for the BREB equation (Table 6). With the hysteresis model the seasonal bias shifts to an underestimation of evaporation in summer and autumn and relatively good results for winter and spring, resulting in a total underestimation of the annual amount by 8 % (Fig. 7 c,V2). In V3 the heat storage is considered as



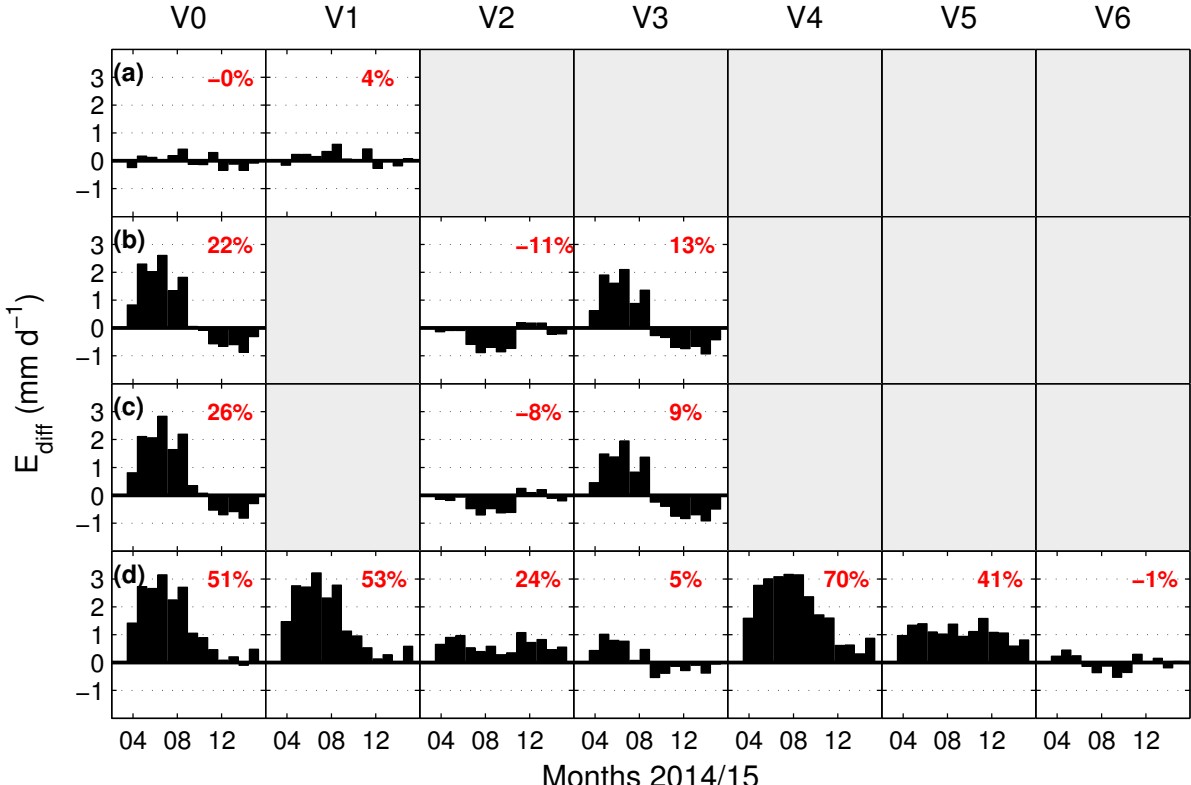

**Figure 7.** Differences between the estimated daily evaporation amounts calculated from the 28 d time averages and the measured evaporation amounts for (a) the aerodynamic approach, (b) the BREB method, (c) the Priestley-Taylor equation, (d) the Penman equation, and their sensitivity studies (Table 2). The red numbers show the total deviation of the accumulated annual evaporation estimate from the accumulated measured evaporation amount.

a linear function of $Rn$ and thus results in a new Priestley-Taylor coefficient of 1.09. With V3 the seasonal bias is reduced but still high evaporation rates are overestimated and low ones underestimated (Table 6). The total annual amount is overestimated by 9 % (Fig. 7 c, V3).

The last equation tested is the Penman equation. In its original form (V0) it results in correlation coefficients of 0.78 for time averages of 1 d to 0.91 for 28 d (Table 6). However, the slopes of the regression lines vary between 1.44 and 1.76, respectively, and indicate an overestimation. The mean differences also show a strong variability. Evaporation values are strongly overestimated from spring until autumn (Fig. 6 d, V0), exceeding the measured daily evaporation amounts by up to 100 % (Fig. 7 d, V0). The total annual amount is thus also overestimated by 51 % showing that the original Penman equation is not applicable for the investigation of intra-annual variations. The consideration of the heat storage using the hysteresis model (V2) yields considerable improvements. The correlation coefficient is improved to values between 0.87 and 0.97, and the mean difference and its standard deviation reduced, meaning that the spread of the calculated values is smaller (Table 6). The slopes for V2 are



all below unity and the offsets above $0.94\,\mathrm{mm\,d^{-1}}$, meaning that small values are highly overestimated (Fig. 6 d,V2). This is also apparent in the intra-annual deviation of the estimated values from the measured amounts. Deviations are below or around $1\,\mathrm{mm\,d^{-1}}$ for all months, resulting in a total overestimation of the annual amount by 24 % (Fig. 7 d,V2).

The calculation of the heat storage as a linear function of the net radiation results in $\Delta Q = 0.46 \cdot Rn$. Using this function for the
heat storage term in V3, the results are strongly improved. The slopes of the regression lines are close to one, offsets are small and also the mean differences are smaller (Table 6). Correlation coefficients vary between 0.82 and 0.92 and the overall annual evaporation amount is with 5 % within the range of the measurement uncertainties. However, the results show a seasonal bias with an overestimation in spring and summer and a underestimation in autumn and winter (Fig. 7 d,V3).

Another commonly used variation of the Penman equation is the removal of the surface water temperature from the calcula-
tion of the net radiation. This is tested in V4. However, in V4 the heat storage term is still missing an thus does not result in reliable evaporation values (Fig. 6 and 7 d,V4). The combination of the hysteresis model with the removal of the surface water temperature (V5) yields an improvement of the correlation coefficients, the slope of the regression lines and also the standard deviations, but the calculated values show an offset of over $0.93\,\mathrm{mm\,d^{-1}}$ (Table 6). This results in a constant overestimation of evaporation values throughout the year and results in a total evaporation amount which is 41 % higher than the measured
one (Fig. 7 d,V5). The last test for the Penman equation combines the removal of the surface water temperature with a derived linear function for the heat storage term from V4. With an heat storage term $\Delta Q = 0.77 \cdot Rn$ the discrepancy of the calculated from the measured values can be minimized. The regression line is still slightly tilted (Fig. 6 d ,V6), small evaporation values are overestimated, and large ones underestimated, but the mean difference is nearly zero and the standard deviation is in the range of 0.29 to $0.5\,\mathrm{mm\,d^{-1}}$ (Table 6). The total evaporation amount is well represented with this adjustments of the equation
(Fig. 7 d ,V6).

## 5    Discussion

Results from former studies, which investigated Dead Sea evaporation by using indirect approaches, varied strongly. No direct evaporation measurements were performed at the Dead Sea so far to validate these results. Hence, there was a need for such direct evaporation measurements. The eddy covariance method is recognised internationally to be a very accurate method to
directly measure evaporation (e.g. Rimmer et al., 2009) and was therefore chosen for this study. The measurement strategy was based on the installation of the station on a headland, which was surrounded by water from 320°. This setup at the shoreline was chosen to avoid influence on the measurements by raft motion and sea spray, where the latter one leads to a serious soiling of the instrument and influences data quality strongly. However, land based eddy covariance measurements have their limitations in measuring evaporation from the water surface, as part of the flux footprint is located over land. Therefore, a
multiple regression model was applied to the data and the results show that evaporation from the Dead Sea water surface is driven by wind speed and vapour pressure deficit. The model was tested using a MCCV, and it was confirmed that the model is reliable for calculating the Dead Sea evaporation and has a small model error of 4.8 %. Additionally, a slight overestimation of model based evaporation might occur, as the vapour pressure deficit for offshore wind conditions is probably higher directly



**Table 6.** Slope and offset of the regression lines between the evaporation estimates calculated with the different equations and the evaporation measurements and the corresponding correlation coefficient $R$, for averaging periods of 30 min, 1, 7, 14, and 28 days. Mean difference and standard deviation are shown for 1 d and 28 d as no relevant differences for the other time intervals exist. V0 to V6 indicate the different sensitivity studies (see Table 2). The best fitting solutions are indicated with bold numbers.

| | | Slope | | | | | Offset | | | | | R | | | | | MD $\pm$ std | |
|---|---|---|---|---|---|---|---|---|---|---|---|---|---|---|---|---|---|---|
| | | 30 min | 1 d | 7 d | 14 d | 28 d | 30 min | 1 d | 7 d | 14 d | 28 d | 30 min | 1 d | 7 d | 14 d | 28 d | 1 d | 28 d |
| Aero- | **V0** | **1.26** | **1.13** | **1.08** | **1.10** | **1.12** | **-0.01** | **-0.33** | **-0.24** | **-0.29** | **-0.34** | **0.85** | **0.94** | **0.94** | **0.98** | **0.99** | **0.02±0.54** | **-0.02±0.24** |
| dynamic | V1 | 1.27 | 1.16 | 1.10 | 1.13 | 1.14 | -0.01 | -0.30 | -0.17 | -0.23 | -0.26 | 0.85 | 0.94 | 0.94 | 0.98 | 0.99 | 0.13±0.54 | 0.11±0.24 |
| | V0 | – | 1.27 | 1.51 | 1.63 | 1.72 | – | -0.13 | -0.78 | -1.11 | -1.35 | | 0.67 | 0.78 | 0.84 | 0.87 | 0.60±1.78 | 0.61±1.26 |
| BREB | V2 | – | 0.45 | 0.57 | 0.63 | 0.67 | – | 1.21 | 0.89 | 0.70 | 0.59 | – | 0.69 | 0.83 | 0.90 | 0.96 | -0.30±0.89 | -0.30±0.40 |
| | V3 | – | 1.17 | 1.39 | 1.50 | 1.58 | – | -0.12 | -0.72 | -1.02 | -1.24 | – | 0.67 | 0.78 | 0.84 | 0.87 | 0.33±1.62 | 0.33±1.11 |
| Priestley- | V0 | – | 1.35 | 1.61 | 1.74 | 1.84 | – | -0.28 | -0.98 | -1.33 | -1.58 | – | 0.69 | 0.80 | 0.86 | 0.89 | 0.69±1.81 | 0.70±1.30 |
| Taylor | V2 | – | 0.49 | 0.62 | 0.70 | 0.74 | – | 1.17 | 0.81 | 0.59 | 0.47 | – | 0.73 | 0.87 | 0.93 | 0.98 | -0.24±0.84 | -0.23±0.32 |
| | V3 | – | 1.17 | 1.39 | 1.51 | 1.59 | – | -0.24 | -0.85 | -1.15 | -1.37 | – | 0.69 | 0.80 | 0.86 | 0.89 | 0.23±1.53 | -0.24±1.04 |
| | V0 | – | 1.44 | 1.58 | 1.69 | 1.76 | – | 0.19 | -0.20 | -0.49 | -0.69 | – | 0.78 | 0.83 | 0.88 | 0.91 | 1.38±1.52 | 1.38±1.17 |
| | V1 | – | 1.44 | 1.57 | 1.68 | 1.76 | – | 0.24 | -0.12 | -0.41 | -0.61 | – | 0.78 | 0.83 | 0.88 | 0.91 | 1.44±1.52 | 1.45±1.16 |
| | V2 | – | 0.75 | 0.80 | 0.85 | 0.89 | – | 1.34 | 1.21 | 1.04 | 0.94 | – | 0.87 | 0.89 | 0.94 | 0.97 | 0.65±0.61 | 0.64±0.25 |
| Penman | **V3** | **–** | **0.94** | **0.99** | **1.05** | **1.09** | **–** | **0.29** | **0.16** | **0.00** | **-0.11** | **–** | **0.82** | **0.84** | **0.90** | **0.92** | **0.13±0.82** | **0.13±0.51** |
| | V4 | – | 1.54 | 1.73 | 1.80 | 1.87 | – | 0.45 | -0.09 | -0.28 | -0.48 | – | 0.89 | 0.91 | 0.94 | 0.96 | 1.92±1.16 | 1.91±1.07 |
| | V5 | – | 0.96 | 1.01 | 1.04 | 1.06 | – | 1.22 | 1.10 | 1.00 | 0.93 | – | 0.92 | 0.93 | 0.96 | 0.97 | 1.10±0.51 | 1.10±0.27 |
| | **V6** | **–** | **0.78** | **0.80** | **0.81** | **0.84** | **–** | **0.59** | **0.54** | **0.49** | **0.42** | **–** | **0.92** | **0.92** | **0.95** | **0.97** | **-0.02±0.50** | **-0.02±0.29** |

at the shoreline in comparison to the open water surface. Considering these uncertainties the model can be used to calculate evaporation values for offshore wind conditions, enabling, for the first time, the analysis of the full diurnal and intra-annual cycle of Dead Sea evaporation.

The results show that the diurnal cycle of evaporation is mainly driven by the diurnal cycle of the wind systems and their related
5   wind velocities. This leads to maximum evaporation rates after sunset, caused by westerly winds with high wind velocities. These westerly winds occur from spring until autumn. The results are consistent with findings for Lake Kinneret, where these westerly winds also occur in the evening (Assouline and Mahrer, 1993; Shilo et al., 2015). However, the daily evaporation rates are notably lower compared to the evaporation at Lake Kinneret through the much higher salinity of the Dead Sea water and thus the reduced saturation vapour pressure. The median daily evaporation ranges from 1.1 mm d$^{-1}$ in December to 4.3 mm d$^{-1}$
10   in July, but the absolute maximum of the measured daily evaporation rates was measured in November with 6.9 mm d$^{-1}$. This is extremely high compared to the median values in winter and highlights the stronger synoptic influence on the region during the wet season (Bitan, 1974, 1976). One of the typical synoptic systems during the wet season is the Red Sea Trough, which





can cause high wind velocities and high vapour pressure deficits in the valley and thus leads to very high evaporation rates. This is particularly important as Alpert et al. (2004) found that the frequency of such Red Sea Trough systems nearly doubled since the 1960s from 50 to 100 days per year.

The total measured evaporation for the period 1 March 2014 until 1 March 2015 was $994.5 \pm 81.2$ mm, which agrees with pre-
vious findings such as Stanhill (1994) with 1005 mm a$^{-1}$ and is close to the results from Lensky et al. (2005) ($1100 - 1200$ mm a$^{-1}$), which both estimated the evaporation based on theoretical energy balance approaches. However, it is far away from the 2 m from Salameh and El-Naser (1999), who estimated evaporation based on water balance calculations, which could indicate uncertainties in the assessment of the water balance components.

Eddy covariance measurements provide high resolution and accurate evaproation data but they are costly and need frequent
maintenance. Therefore, it is difficult to maintain an operational system in remote areas. Hence, the third aim of this paper was to evaluate the applicability of commonly used indirect evaporation equations, which use standard meteorological measurements. The BREB, Priestley-Taylor, and Penman method, are difficult to apply for intra-annual calcuations. The main difficulty is the heat storage term. For these three methods the knowledge of the heat storage term is essential to achieve reliable results, as neglecting the heat storage results in a strong seasonal bias, with an overestimation of daily evaporation rates of up to 100
%. Using estimates of the heat storage term does not provide acceptable results for the BREB and the Priestley-Taylor method either. For the Penman equation an applicable solution is achieved when a linear function for the heat storage is empirically gained from the data set. We conclude that the BREB and Priestley-Taylor method can only be applied for the Dead Sea if heat storage is measured, which requires a raft station or ship measurements, or for long time periods, i.e. one year, where the heat storage term can be neglected. The Penman equation is applicable for the Dead Sea, if the heat storage is considered
using the described approaches. The aerodynamic approach yields the best results with respect to the diurnal and intra-annual calculation of evaporation. They were in best agreement with the measurements. It was also shown that the consideration of the atmospheric stability in the calculations has an neglegible effect on the results. This again coincides with results for Lake Kinneret (Shilo et al., 2015; Rimmer et al., 2009) and makes this method easily applicable for evaporation calculations, as only wind velocity and vapour pressure deficit are required.

## 6 Summary

This study focuses on providing an applicable method to investigate the diurnal and intra-annual variability of evaporation from the Dead Sea water surface using an eddy covariance system located at the shoreline. Furthermore, it investigates the application of commonly used indirect methods to calculate evaporation with shoreline data.

When using a station located at the shoreline, the use of a new model is proposed to calculate Dead Sea evaporation for
offshore wind conditions on sub-daily time scales. It was shown that a model consisting of a linear combination of wind speed and vapour pressure deficit results in a robust model for the calculation of evaporation. This approach can also be applied to other lakes, making expensive raft measurements expendable. From the evaluation of the indirect methods we conclude that for a reliable estimate of the Dead Sea evaporation the aerodynamic method is advisable and that the influence of the





atmospheric stability is negligible. Like the new model, the aerodynamic method connects evaporation with its governing variables, which are wind velocity and vapour pressure deficit, and allows the calculation of sub-daily or multi-day evaporation amounts without a seasonal bias. The advantage thereby is clearly the use of cost-efficient measurements, which can be installed on the shoreline, to estimate evaporation values on a sub-daily time scale. This is beneficial for economic purposes, such as

5   the production of minerals from the saline water, as well as for further investigations of the water budget of the lake, and the resulting environmental changes on a longer time scale.





## Appendix A: Measurement of the latent heat of vaporisation

The latent heat of vaporisation and the activity of water $\beta$ for the highly saline water of the Dead Sea were measured using a water probe taken at the measurement site of the EBS at the end of 2014. First, the saturation vapour pressure of pure water $E_w$ was measured with a capacitance manometer, which was calibrated by a linear regression to literature values from

the Kilolabor ETH Zurich[1]. Afterwards, the saturation vapour pressure of the saline water, $E_s$, was measured as a function of water temperature with the calibrated manometer. Through this approach possible measurement uncertainties of the manometer could be minimized. The activity of water can then be calculated as:

$$\beta = \frac{E_s}{E_w}. \tag{A1}$$

The averaged activity for the Dea Sea water is $\beta = 0.65$.

The molar latent heat of vaporisation, $\Delta H_{vap}$ (J mol$^{-1}$), can be derived by using the general form of the Clausius-Clapeyron equation, assuming that the molar volume of the liquid can be neglected against the molar volume of the gas, and by using the ideal gas law:

$$\Delta H_{vap} = -R \frac{\mathrm{d}(\ln E_s)}{\mathrm{d}(\frac{1}{T_w})}. \tag{A2}$$

R=8.314 J mol$^{-1}$ K$^{-1}$ is the universal gas constant, the corrected saturation vapour pressure of the saline water is $E_s$ in hPa,

and water temperature is $T_w$ in K. With the molar mass of water $m_{H_2O}$=0.018 kg mol$^{-1}$, the specific latent heat of vaporisation $L_v$ can be calculated:

$$L_v = \frac{\Delta H_{vap}}{m_{H_2O} \cdot 1000}, \tag{A3}$$

in kJ kg$^{-1}$, and can then be fitted to the water temperature $T_w$ (Fig. A1). The regression formula is:

$$L_v = 5150.6561 - 13.9530 \cdot T_w + 0.0162 \cdot T_w^2. \tag{A4}$$

[1]https://cdm.unfccc.int/filestorage/U/4/B/U4BKYDK7NTLWWFQ1OTUFUCKJMTEE3Y/U4BKYDK7.pdf?t=Vm98bzQ0aGx1fDC3cDweIA5
PuHui7yRAOy3k





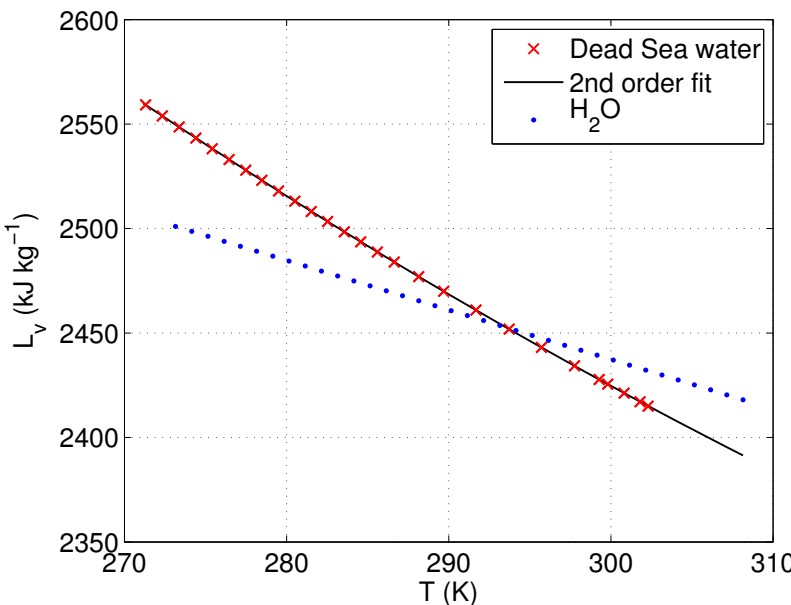

**Figure A1.** Dependency of the specific latent heat of vaporisation ($L_v$) on temperature. Measurements of $L_v$ for the saline water of the Dead Sea, a second order polynomial fit and literature values for pure water ($H_2O$) are shown.

## Appendix B: Calculation of vapour pressure deficit

The vapour pressure deficit for the regression approach is calculated as follows: The vapour pressure deficit is defined as the difference between the saturation vapour pressure above the saline water, $E_s$, and the atmospheric vapour pressure in 2 m height, $e_{a,2\mathrm{m}}$:

$$\Delta e = E_s - e_{a,2\mathrm{m}}. \tag{B1}$$

The saturation vapour pressure of saline water is lower than that of freshwater, $E_w$, by a factor $\beta$, caused by the vapour pressure depression by dissolved salts (Raoult's law) (Atkins, 2014).

$$E_s = \beta \cdot E_w. \tag{B2}$$

The activity $\beta$ depends on the composition of the dissolved salts and is determined to 0.65 for the Dead Sea water in this study (Appendix A). Saturation vapour pressure over water can be calculated using the Magnus equation after Bolton (1980):

$$E_w(T_{0m}) = 6.112 \cdot exp\left(\frac{17.67 \cdot (T_{0m} - 273.15)}{T_{0m} - 29.65}\right), \tag{B3}$$

with water surface temperature $T_{0m}$ in K. As water surface temperature is not directly measured at the station, vapour pressure deficit is calculated using surface temperature obtained by the Monin-Obukhov theory, $T_{MO}$ in K:

$$\Delta e_{MO} = \beta \cdot E_w(T_{MO}) - e_{a,2\mathrm{m}}, \tag{B4}$$




where $T_{MO}$ is calculated with the following profile equation:

$$T_{MO} = T(z_0) = T(z_m) - \frac{\theta^*}{\kappa} \cdot \left( ln\frac{z_m}{z_0} - \Psi_H(\zeta_m, \zeta_0) \right). \qquad (B5)$$

$z_m$ is the measurement height in m, $z_0$ is the roughness length assumed as 0.001 m, $\zeta_m = z_m L_*^{-1}$ and $\zeta_0 = z_0 L_*^{-1}$ are independent dimensionless parameters using the Monin-Obukhov-Length $L_*$, and $\frac{\theta^*}{\kappa}$ is a scaling parameter defined as:

$$\frac{\theta^*}{\kappa} = -\frac{1}{\kappa u^*}\frac{H}{\rho_0 c_p}, \qquad (B6)$$

with $\kappa$=0.4, which is the Kármán constant, sensible heat flux $H$ in W m$^{-2}$, specific heat capacity $c_p$=1004 J K$^{-1}$ kg$^{-1}$ and density of the air $\rho_0$ in kg m$^{-3}$. $\Psi_H$ is the integral over the empirical gained functions $\varphi_H$:

$$\Psi_H(\zeta_m, \zeta_0) = \int\limits_{z_0}^{z_m} = \frac{1 - \varphi_H}{z} dz \qquad (B7)$$

In this work the $\varphi$ functions from Dyer (1974) are used:

$$\varphi_H = 1 + 5\zeta \qquad\qquad\qquad \zeta > 0 \qquad\qquad (B8)$$
$$\varphi_H = (1 - 16\zeta)^{-1/2} \qquad\qquad -1 < \zeta < 0. \qquad\qquad (B9)$$

*Acknowledgements.* The current study was carried out in the framework of the DEad SEa Research VEnue (DESERVE) (http://www.deserve-vi.net), an international project funded by the Helmholtz Association of German Reseach Centres as a Virtual Institute (VH-VI-527). Many thanks to Bernhard Deny and Philipp Gasch for maintaining the stations. Thanks to Pinhas Alpert and Eduard Karat from Tel Aviv University

15 for their support. We also want to thank David Seveloff and Yael Maor from the Dead Sea and Arava Science Center for their support with the measurements and Tamar Regional Council, Ein Gedi Spa, and Mekorot for the provision of the measurement locations.





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
