# Peer review of "Dead Sea evaporation by eddy covariance measurements versus aerodynamic, energy budget, Priestley-Taylor, and Penman estimates"

_Hydrology and Earth System Sciences, 2017_

## Referee Comment (RC1) · Anonymous Referee #1 · 13 Jul 2017

**1    General comments**

I think this paper deserves publication because the dataset is particularly interesting. As said by the authors, it is the first time that one year of turbulent atmospheric fluxes measured by the eddy-correlation (EC) method has been presented for the specific area of the Dead Sea. The data have been professionally processed. They can efficiently be used to assess several parameterizations generally used for long-term measurements when the EC method is not available. The originality is that the authors provide several levels in the parameterizations, according to the measurements that can be performed.

However the paper requires some important corrections before being published. I try to describe them below. Some corrections, related to the methodology, are essential. Other are secondary and consists in numerous details that could be improved.

**2    Specific comments**

Methodology : I think the difficulty comes from the fact you make a local measurement over ground (with the EC method) while you would like to include the close environment in the driving parameters of the turbulent fluxes, considering that the air that is advected on the measurement site is (most of the time) characterized by the water surface temperature and water surface vapor partial pressure. I am not against the idea, but I think the way you deal with this assumption is not always correct. I am aware that you want to prove that measuring on the headland, very close to the seashore is equivalent to measuring with a raft in the middle of the sea, but I am not totally convinced.

Another issue is the fact that you address different time scales for the energy budget you consider, without saying accurately which timescale you refer to.

- When measuring the latent heat flux LE at level 6m with the EC, the Lv value (kJ/kg) that has to be used to change the evaporation rate $\overline{w'a'}$ into a flux is that of the air, i.e. 3148.4-2.37 Ta and not $Tw$ as you use in Eq. 3. $T_a$ is the air temperature in K. Even if the evaporation takes place at the water surface as you mention p 6, line 3, LE is assumed to be constant in the surface layer (in fact, not to vary more than 10% of its surface value). This unique Lv value should be used for both offshore and onshore winds. In fact an internal boundary layer develops either inland or offshore, depending on the wind direction. Perhaps you should mention it and clarify the parameters you use in both situations. [The only opportunity to use Lw as you define it, would be to take into account the heat loss of the water due to water evaporation, as shown by Giadrossich et al., 2015 in their eq (2). This eq. applies to the energy budget of the whole sea, and not the local energy budget that you quantify at the EBS].

- When discussing the various models you apply to your data, you use Tom-Ta or Ew-Ea. The latter should be replaced by $E_{surf} - Ea$, with $E_{surf}$ (or another name) standing for the water vapor partial pressure at the surface (water or ground). The former, by $T_{surf} - Ta$.

— For onshore winds, as the source area is over water, I think that the similarity profile you use in Appendix B, p26, to deduce the water surface temperature is not appropriate since the regressions you wish to establish in the following will depend on this profile. I suggest that you'd try to find independent remote-sensed measurements of the Dead Sea surface temperature instead, which is not exactly the air temperature at the surface, but is the closer you can find. If you do so, I am almost sure that the discrepancy between panels 2 and 3 in Fig. 2 will be larger. In that way, the models you will apply in the following will include independent measurements (since the temperature difference will not result from the similarity profile). For offshore winds, the source area is partly over water, partly over land. The $\Delta T$ estimation should be a combination of ground surface temperature and water surface temperature. There again, an estimation of the ground surface temperature (perhaps after some assumptions of the emissivity) would be appropriate. You could perhaps also use the upward longwave radiation flux measured at the neighbouring ground station.

— For offshore winds, I do not know any mean to deduce $E_{surf}$, unless you make assumptions on the water vapour at the ground surface. You could use the similarity profile, but meanwhile you would make the choice of a model and Sections 3.3 and 4.4 would become useless. For onshore winds, $E_{surf} = 0.65\,Ew(Tw)$

could be an appropriate estimation, Tw being the satellite sea surface temperature, instead of Tmo.

— A sensitivity study of the regressions to the error in Tw (which determines ew) could also be informative.

- If we consider the 3 objectives you propose to fulfill : i) is fulfilled by the EC method, but you do not need to use any multiple regression model to quantify the offshore conditions : you directly measure them. The way you can link these flux measurements with local air or surface parameters is another issue. ii) I think you cannot totally achieve this aim since you can only access to the local terms of the energy budget and make assumptions on the terms you have to neglect. iii) OK if you define what 'evaporation' is. You could also add that you want to assess the capacity of these models to retrieve the 'evaporation' term, in the future, when the EC sensors are not available any more (you suggest this in the conclusion).

- In general, you could perhaps take more benefit from the literature you quote. For instance, you mention Giadrossich et al.'s work for minor reasons whereas you could compare their results to yours, since the measurements they carried on, are characteristic of a semi-arid area, like the area you study.

I think modifications explained in 1 to 4 should be done before I read the paper again. However, I add below more accurate remarks and some comments, that could be useful to improve the paper (at least the beginning).

**3 Other remarks**

[Note : there is no intention in numbering e.g. 18a, 18b, 18c : I simply inserted this lettering because I wanted to add remarks without changing the numbers ...].

- The term 'evaporation' you frequently use is not accurate enough. I will point it out in the following.

- p1, line 6 : 'total annual amount measured' $\longrightarrow$ 'total annual amount of evaporation measured' (in this case, you do not need to be more accurate since you provide the evaporation unit).

- p1, line 7 and further on : 'vapour pressure deficit' $\longrightarrow$ 'water vapour pressure deficit'

- p1, line 8 : 'Consequently' is not appropriate. Perhaps 'in fact' could be used instead. What do you mean by 'evaporation amounts'?

- p1, line 10 could be changed to $\longrightarrow$ 'during daytime. During nighttime, evaporation rates are also larger than the daytime evaporation rates, due to strong ...'. Why do you use 'evaporation rate' this time ? Note that this result will perhaps require corrections, in light of what is said in the following (see my final remarks for instance).

- p1, line 11 : The link 'Furthermore' is somewhat awkward. You should explain here why you calculated the regressions. By the way, I think that the multiple regressions should be established for another purpose (see my remark 46-).

- p1, line 14 is clear and nice.

I skip lines 15, p1 to the Introduction, p2.

- p2, line 6 : you could add 'down' after '90%'.

- p2, line 7 : during which period this decrease happened ? And 60-400 denotes a very large variability. Can you explain why ? (variability among the authors ?)

- p2, line 9 : 'The total amount is about' $\longrightarrow$ 'The total amount of loss is about'

- You could add a budget equation such as : $10^6 (400 + 240 - 250) + evaporation = -650 \, 10^6$, which gives evaporation $= -1060 \, 10^6 \, m^3 a^{-1}$, which is in the range $700 - 1400 \, 10^6$, indicated by Gavrieli et al., 2006.

- p2, lines 13-14-15 : I would replace ' Evaporation is not only ....environmental problems.' by something like 'It is important to assess the water budget components of the Dead Sea for a climatological purpose, but it is also a priority for the people and the socio-economic development of the region to anticipate the evolution of these components and the consequence for the environment. For instance, the lake level decline causes severe environmental problems . ' (you may of course change the words).

- p2, line 16 : $\longrightarrow$ 'shifting of the fresh/saline groundwater interface' ('of the' has been added). Please define sinkholes, I did not know this phenomenon.

18a - p2, line 25 : why westerly winds would be harmful, compared with easterly winds ? You could also delete 'the' at the end of the line, before '1940'.

I suppose this harm comes from the fact that easterlies carry drier (continental) air, whereas westerlies carry moister air. But this cannot be guessed from what you wrote here.

18b - p 2, line 29 (no link with 18a) : I do not understand 'especially' in this context ('In addition'?). Perhaps you could also mention whether it is a fresh water fish. Ein Feshkha reserve?

18c - p3, line 5 : I would add 'in the evaporation estimations' just before '(Stanhill, 1994'.

18d - p3, lines 6-7 : I would remove : 'Furthermore, the governing factors of the Dead Sea evaporation, e.g. wind velocity, vapour pressure deficit, or net radiation, have to be identified, to validate the indirect methods'. These parameters are governing factors every where in the world and you do not have to prove it. I think this sentence is confusing at this point.

18e - p3, line 8 : 'with a high temporal resolution' instead of ',in high temporal resolution'.

- p3, line 10 : 'continues' ⟶ 'continuous'

- p3, line 12 : according to Wikipedia, it seems that Lake Kinneret is the same as Lake Tiberias mentioned by Kottmeier et al. (2016) (it is only a remark, you choose the name you prefer). You could perhaps add that it is crossed by the Jordan river which partly feeds the Dead Sea. It is not a major piece of information but I find it nice to provide the reader an idea of the geographical environment.

- p3, line 13 : 'to the authors knowledge' instead of 'as to the authors knowledge'.

- p3, line 14 : 'Therefore, long-term eddy covariance measurements are conducted' ⟶ 'That is why, in the frame of the international DESERVE project (Kottmeier et al., 2016), long-term eddy covariance measurements were conducted'

- p3, line 17 : 'provided' and 'was' instead of 'provides' and 'is'

- p3, lines 17-19 : I perhaps misunderstood but it seems to me you did not use the data from these stations. Is it useful to quote them?

- p3, lines 19-21 : 'Provide', 'Evaluate' and 'Evaluate' instead of the same with 'ing'.

- p3, Measurement site : it is difficult to distinguish in Fig. 1, how far the Judean mountains highest submits are from the lake and what their height is (hills or mountains?). The Moab mountains are clearly shown.

- Fig. 1 : you forgot to write 'Jordan river' in panel (a). Landsat with an L in the caption. The red arrows are a little confusing in panel (b) : simple lines instead of arrows would be enough.

- p5, line 4 : Rotronic

- p5, line 5 : is it a tipping bucket rain gauge?

- p5, line 7 : please add 'open path' before 'integrated gas analyser'. I suppose that at 6m, it did not suffer from spray, even under strong onshore wind conditions ...

- p5, line 9 : 'From the 20 Hz data evaporation was calculated using the eddy covariance technique' ⟶ 'The latent heat flux was calculated from the 20 Hz data using the eddy covariance method.'

32a - p5, line 9-10 : ⟶ 'The principle of the method, the post-processing and data quality control steps are presented in Sec. 3.1'.

32b - p5, lines 11-12 : please consider the multiple regressions again (after 46-)

- p5, eq (1) and line 23 : the ultrasonic anemometer provides the virtual air temperature and not the air temperature Ta. The Schotanus correction is made, as you say in Sec. 3.1.1, to take this point into account.

- p5, eq (2) and line 24 : usually, $LE = \rho_a \, Lv \, \overline{w'r'}$ or $LE = \rho_a \, Lv \, \overline{w'q'}$ where r and q are the water vapour mixing ratio and specific humidity, respectively. Brutsaert (which you refer to in the following) uses the latter definition for the evaporation rate : $Ev = \rho_a \, \overline{w'q'}$. In addition, if I remember well, the hygrometer converts the absolute humidity in water vapour mixing ratio, using T=20°C and P=1013.25 hPa. Perhaps you could consider using $r$ or $q$ instead of $a$.

- p5, eq (3) and line 27 : as said before, Tw should be Ta.

- p5-6, lines 27 to 5 : the text should be deleted from 'For salt water ..' to the end of the subsection.

- p6, line 8 : 'measurement limitations' you could add 'of the sensors'

38- p6, lines 18 to 24 : it seems to me that the order should be : spectral corrections, Schotanus correction and Webb correction unless you applied an iterative process.

39- p6 or before (p5) : did you use a constant calibration coefficient for the IRGASON (when was the calibration done?) or did you calibrate the hygrometer measurements against the low frequency humidity measurements?

40- p6, line 28 : 0.5 g/kg?

41- p6, line 29 : 'when the variability of the signal' . Could you define the variability (standard deviation/average?)? I think it is 0.6 and not 0.6%

- p6 line 31 to p7 line 2 : you say too much or not enough. It would be nice to describe the tests and to say what ITC is.

- p7 and further on : I would replace 'fetch' by 'source area'.

- p7, lines 6-7 : please consider again the rejection of these data : I agree that it is important to distinguish them from the onshore measurements data, but the fluxes are what they are and do not have to be rejected. Nevertheless, it is important to quantify this contribution since the source area may be different.

- p7, lines 7 to 9 : I would replace the 2 sentences 'For southerly wind directions ... 600m away from the headland.' by 'For southerly and northerly wind directions the source area is over water and the average source area contributing to 80% of the flux ranges from 0 to 300 m and 0 to 600 m, respectively, ahead of EBS.'

- p7, subsection 3.2 : once you have made the corrections indicated in point 2, I suggest that you keep on calculating the multiple regressions, but not with the aim of parameterizing the offshore conditions. Rather, to show how the classical relationship between the latent heat flux and the wind and/or vapour pressure deficit behaves, under the specific conditions of a semi-arid area that is influenced by sea breeze, slope breeze or both. The multiple regressions should be done for onshore and offshore data separately (provided that 19% of the dataset is enough to apply your method to the offshore data). Note that the regressions you examine are similar to the aerodynamic (or bulk model) from Brutsaert. That is why I am not surprised by the good correlations you obtained in Table 4 for V0. That is also the reason why I do not agree with the idea of using these multiple regressions to calculate offshore winds.

I also think that the presentation of the simple regressions should be shorter since they are known to fail to represent the flux but they can serve as a base to compare the multiple regression $H = f(U, \Delta T)$, to the regression $H = f(U \Delta\theta)$, and the same with LE, U and $\Delta e$. Please also think of using $\Delta\theta$ instead of $\Delta T$, although the difference will be very small.

You may have noticed that I added a multiple regression $H((U, \Delta T)$. The reason is because the BREB method is partly based on this correlation.

- p8, line 1 : 'Indirect methods to estimate evaporation' $\longrightarrow$ 'Description of four indirect methods to estimate evaporation'

48a - p8 : I would move the first two (essential) sentences of subsection 4.4 ('For the calculation of evaporation -please insert a comma here-, several equations, based on .... at least 7 days') to the beginning of section 3.3.

48b - p8, line 5 'an overview of which sensitivity study is performed' (I added 'of').

- Ev is not defined. I suppose it is the evaporation rate defined as Ev=$\rho_a \overline{w'q'}$ (Brutsaert, 1982).

- p8 : I would substitute 'aerodynamic method' for 'aerodynamic or mass transfer method'. In fact it is the bulk method, frequently used to estimate surface fluxes over the sea, where the EC method or dissipative method are not easy to implement. Brutsaert (1982) refers to it as the 'bulk transfer' method and it is based on similarity profiles assumptions and the relationship between fluxes and wind or scalar gradients (through the Dalton or Stanton numbers). According to Brutsaert (p88 in my edition, reprinted in 1984)

$$Ev = C_e \, \rho_a \, v_a (q_{surf} - q_a) = C_e \, \rho_a \, v_a \, (e_{surf} - e_a) \frac{0.622}{p}$$

Without telling it, you assume equal transfer coefficients for evaporation and momentum ($C_e = C_d$). $C_e$ is a mass transfer coefficient for evaporation and $C_d$ is the drag coefficient = $\frac{u^2_*}{v^2_a}$. Introducing the logarithmic wind velocity gradient under neutral conditions, which is a second assumption, Ev becomes :

$$Ev = \frac{k^2}{(ln\frac{Z}{zo})^2} \, \rho_a \, v_a \, (e_{surf} - e_a) \frac{0.622}{p}$$

So $K_E$ you identify in Table 1 should not contain $\rho_w$. I suppose you needed to add it since the kinematic flux you calculated is in term of absolute humidity (but you should have divided Ev by $\rho_a$ and not $\rho_w$). To conclude, I suggest you add a remark concerning the assumption that $C_e = C_d$. Perhaps this could be explained in an additional Appendix (3). I also suggest that you move into the present subsection, your sentence from p17, lines 12-13 : 'the aerodynamic approach is the only approach designed for sub-daily time intervals'. And you could add '(typically 30 min in this study)'. I find that you well address this timescale issue in the following, specifically with Table 6 where you show the results. However, it should be also clearly mentioned in subsection 3.3 (for the 4 models).

- The sensitivity study, referred to as V1, is performed to address the stability issue. The presentation of the stability factors you refer to (Cline 1977) p. 17 should also be moved to the present subsection. I would also describe the new $K_E$ (including the stability factor) in Appendix 3. You could also add that the stability functions for wind and heat are expressed in terms of the bulk Richardson number, which allows estimating the stability when the turbulent fluxes are not known.

- Energy budget : Here again, $\rho_w$ should vanish if you use the specific humidity instead of the absolute.

- I suggest that you write down the budget equation as Giadrossssich, 2015 did (their eq 1) : $Rn + A_{net} = LE + H + \Delta Q$ , where $A_{net}$ is the net heat advected into the lake (by stream flow and precipitation minus the heat loss due to evaporation minus the heat transferred at the bottom of the lake) and $\Delta Q$ is the heat storage per unit area in the lake (for most cases) or in the ground (for specific cases with strong offshore winds. Under these conditions, $A_{net}$ can be ignored). This energy balance applies to timescales larger than the day due to the advection term that cannot be known at a short timescale.

- With V0, you neglect $\Delta Q$ and $A_{net}$ . [Note that the resulting reduced budget equation can also be applied at a sub-daily scale.] Neglecting $\Delta Q$ is a coarse assumption that is valid only under specific and occasional conditions. I think you should mention it at first and say that V0, even unrealistic, is a basis for the BREB method that will be improved by V1 and V2. I mean that V1 and V2 should not only be considered as sensitivity studies for V0, but as 2 alternative methods for V0.

- p9, line 8 : you forgot to mention the water vapour deficit.

- $\beta$, the Bowen ratio should be defined as H/LE. When fluxes are unknown, $\beta$ can be approximated by the expression you give, provided that $K_\theta$, the Stanton number for temperature $= C_e$, the equivalent for evaporation.

Also, be careful not to use the same symbol for the Bowen ratio and the activity of water in Appendix A (I would keep $\beta$ for the Bowen ratio).

- The hysteresis model from Duan and Bastiaanssen, $\Delta Q = aRn + b + c\frac{dRn}{dt}$ should be described including the discussion about the term $\frac{dRn}{dt}$. Note also the dependance of c on the range and variability of the water surface temperature. V3 is a specific case of V2 where b=0 and c=0, $a$ being obtained as 'the deviation of the default version from the measurements'. Did you try to determine specific (a,b,c) for your own dataset, just to quantify the deviation relative to Duan and Bastiaanssen's results ? I do not suggest to include them in the models you use, since, doing so, you would invalidate the V5 regressions.

- Could you please be careful to discuss the assumptions of the other two models and give additional information for Kohler and Parmele's work ? I would not say that $T_s$ has been removed (if I understood correctly) but that it has been estimated from the long-wave radiation flux. This is no-doubt an improvement, relative to your initial estimation from the similarity profile. Please use the same symbol to design the water temperature as the one you have used in the previous section, unless you want to distinguish it on purpose.

- Table 1 : caption : default versions (V0) in Sec. 3.3 and 4.4. (just add 3.3)

Fn and $\Delta t$ are not used and $\rho_w$ should not be used. Please take into account my remarks in 2-

Priestley-Taylor is presented as the 3rd method in Sec 3.3.

- Meteorological conditions : this subsection will have to be read again after new LE, Tsurf, $\Delta_e$ values ...

- For some parameters, the daily values and their evolution are also interesting to discuss in addition to the extreme values.

- Fig. 2 : Please add 'prec' after 'daily precipitation amount' in the caption. Which temperature is $T_a$ : ultrasonic at 6m, BetaTherm probe at 6m, HC2S3 probe at 2m ? Please represent Tsurf instead of Tmo. You could also show $T_{surf} - T_a$. Please represent $e_{surf} - e_a$, and perhaps also $q_a$. I suppose the air is very dry during summer. It would be interesting to show the annual evolution of the air specific humidity. 200 $Wm^{-2}$ are enough for the H vertical axis. It would be convenient to add a thin line for 0 $Wm^{-2}$ in the lower panel.

Did you try to represent the daily average parameters (Rn, H, LE, $\Delta Q$) on the same graph (with a more appropriate scale on the y-axis), to be able to see the phase shift between the annual maxima and also whether the Rn variation relative to the time is linked or not with $\Delta Q$ (in relation with the hysteresis model from Duan and Bastiannssen).

- p10, line 4 : 'long term annual mean' : during which period ?

- p10, line 5 : please add 'sometimes' before 'exceeded'

- p10, line 6 : 'the annual precipitation normal of 80 mm' : the word normal is not accurate. What is the period considered by Goldreich, 2003 ?

- p10, line 7 : 'made' instead of 'makes'

- p10, line 9 : 'only during the winter seasons a different behaviour was found' $\longrightarrow$ except during winter, when the wind increased in connection with the convective activity, a different behaviour was found'

- p10, lines 14 and 15 : are 32 and 26 % relative to lake breeze or lake breeze+ synoptic conditions ?

- p10, line 15 : Please indicate the direction of the downslope breeze (north-westerly)

- p10, line 16 : 'yielded' instead of 'lead to'

- p10, lines 18, 19, 24 : 'exceeding' instead of 'of over'

- p10, line 22 : 'November a Red Sea Trough with a central axis advected dry and warm air' $\longrightarrow$ 'November, when a Red Sea Trough advected dry and warm air'

- p10, line 25 : 'However, at' $\longrightarrow$ 'However, on'

- p10, line 26 : 'in winter latent heat flux values' $\longrightarrow$ 'in winter some latent heat flux values'

- p10, line 27 : the energy balance equation should have been shown in subsection 3.3. You only need to say that at this timescale (24h), $A_{net}$ is ignored. Do you think it is still correct after rainfall ?

- p10, discussion on the energy budget : you assume that the energy budget is closed and you never discuss the frequent non-closure energy budget problem that is reported in several studies in the literature (see Foken et al. in Aubinet et al. p108-109). You cannot avoid this discussion, even if there is no mean to estimate the error, especially because your measurements are done at the boundary between the marine and the continental surface layers. Under these conditions, the surface change may generate large scale heterogeneities that are unlikely to be correctly taken into account by the local measurements.

You show in Fig. 3, extreme values of $\Delta Q$ that are of the order of the net radiative flux in spring and summer. It is unlikely to be true. Anyway, it is known that the shorter the timescale, the larger the non-closure. By contrast, the average $\Delta Q$ (daily average) is about 100 $\mathrm{Wm}^{-2}$ during summer and decreases down to a few tenth of $\mathrm{Wm}^{-2}$ in winter, which are rational values (I remember that LE has to be recalculated, but it should not be very different).

- p10, line 31 : please reword 'used for heating the lake, which is stronger in spring than in winter'. This is grammatically false.

- p10, line 33 : I do not see that $\Delta Q$ is negative in winter.

- p 12, fig 3 : it is usually required to indicate the delay between the UTC and local time. You can keep LT and indicate, on the first time (UTC +3h). It would be also interesting to add in the caption the approximate time of sunrise and sunset, especially from Spring to Autumn.

- p12, line 4 : $\longrightarrow$ 'and, thus, [at] most of the [days] data within this time frame' (remove the [words]).

- p12, line 6 : $\longrightarrow$ 'also for the study of the intra-annual .... this gap' (instead of these gaps).

- p12, line 6 : 'A multiple regression model was applied ... for offshore conditions' : As said before, I do not agree with this method : I'm waiting for your decision, regarding the suggestions I made in 2-

I'm convinced that the comparison of the 4 models is very interesting, but I need to see results that I can rely on. Specifically, I do not trust the results you show in Fig. 4. However, I'm ready to recognize that I am wrong.

I stopped the review here.

---

## Referee Comment (RC2) · Anonymous Referee #2 · 17 Aug 2017

General

This a very interesting paper, addressing an important environmental issue (evaporation from the Dead Sea), presenting very important data and (apparently for the first time) directly measured evaporation rates and thus adding important new information to our knowledge. There is no doubt, therefore, that the paper should eventually be published in one or the other form. There are three main pillars: first, to measure, directly, the lake's evaporation using an EC station on the shore (therewith employing the footprint of the water surface). This provides direct measurements for about 70% of the time. I think this approach is well motivated, well explained and also 'well executed'

(all the necessary data treatment, corrections, QC, etc.).

To estimate, as a second pillar, evaporation during the remaining 30% of the time, a statistical model is trained using the onshore wind conditions and the available information during these conditions, to estimate lake evaporation during offshore wind direction. This is very appropriate (and possibly novel) as an approach, and so is the statistical approach, its presentation (and results). However, it has the drawback that the estimated evaporation rates are among the largest during the whole year and thus contribute a substantial fraction of the total (yearly, monthly, daily – the shorter the more variable) evaporation. Therefore, the statistical model does not only have to be tested using the 'usual' tests (cross-validation, etc.) based on the available onshore conditions (what is convincingly being done), but attempts should be made to support the hypothesis that the same statistical model applies (yields the claimed 'good' statistics) when the input data stem from offshore situations (high wind speed in combination with low water vapor pressure deficit). Some suggestions are provided in major comments 2 and 3.

As a third pillar, empirical estimates (for evaporation) are compared in their performance to the measured evaporation. This is adding a great deal of value to the paper. Unfortunately, for three out of four methods the comparison is not valid (because net radiation is not measured over water – which basically invalidates all the estimates; see major comment 1). Furthermore, the estimation of heat storage (in the water) is also flawed (see again major comment 1), so that all the different 'versions' tested are not really conclusive (with the exception of V1). Indeed, the results show that all the empirical methods using net radiation give results quite different from the observations (if only measured – and not estimated – values would be considered, this finding would probably be even more pronounced). The only 'reliable' empirical method is the aerodynamic approach (not using Rn). The problem with the different fields of view (for radiation and turbulent fluxes) could somehow be overcome (for example by using satellite – or other – observations [even literature values would be better than nothing]

to correct for albedo differences between land and water, and by using the [simultane-ously measured] land surface temperature [from the two additional EC sites over land] and the estimated lake surface temperature to correct for different longwave outgoing radiation). To actually estimate heat storage in the water from the available data I consider virtually impossible – so that the hysteresis model is possibly the only available source.

Overall, the addressed issues call for truly major revisions before this paper can e published in HESS.

Major comments

1) Heat storage is calculated as the residuum (P10, l. 26): the authors use the same notation (delta_Q) as above for the 'heat storage of the lake' (p9, l. 5). So, is this meant to yield the heat storage of the lake using the local energy balance? This is not appropriate for two reasons. First of all, the energy balance is based on the turbulent fluxes (which can, in my opinion, be interpreted as reflecting the heat fluxes in the footprint, i.e. over water – of course, if the wind direction is accordingly). Net radiation, however, has a much smaller 'field of view' (a circle with a radius of maybe 2 m a for measurement height of 6 m), so that the albedo is that of the land surface, and the same is true for the longwave outgoing radiation. The considered energy balance, therefore, is reflecting (a combination of) two different surfaces – and the difference cannot be attributed to 'anything' (if not a careful disentangling of the differences between radiation conditions is performed). Second, even if the various sensors would see the same surface, the energy balance (measured at 6 m agl) cannot be expected to be closed. There will be mean advection (possibly even in the vertical – as this location is so close to a step change in surface conditions; water – land), storage (in the air layer between the surface and the sensors!), possibly even vertical flux divergence. Finally, for the local energy balance (still assuming that all the observations correspond to the same surface type), one would also need the ground heat flux. Give the importance of this term (P10, l. 30) the authors need definitely to do something about it.

2) Statistical model to estimate latent heat flux during offshore conditions: (P14, l. 8) Values up to 200 Wm-2...: Interestingly, the estimated values are larger than the measured values (even on average!). Given the statistical model and the high wind speeds especially during the evening hours – in combination with presumed small water vapor pressure deficit - in spring and summer (Fig. 3), this suggests that the statistical model has possibly been used outside the conditions, for which it has been 'constructed'. In other words, the statistical model is trained for cases of high wind speeds (but possibly not even as high as the downslope winds) in combination with (relatively) small water vapor pressure deficit while it is being used for high wind speed and large delta_e. Since these estimated values are not only to fill some gaps in the measured time series but produce the largest values for characteristic times (i.e., after the evening transition), the authors should try to make a very strong case for these estimated latent heat fluxes. In this sense, the statistical model should not only be evaluated in the 'classical sense' (as it is being done – and very convincingly!), but also the question (hypothesis) should be addressed, whether the onshore (training) and offshore (application) conditions are comparable. In other words, how is delta_e over land related to delta_e over water? For this, potentially the two additional EC sites (p3, l. 17 - they are apparently available but not used in this study) could be employed. Similarly, the question should be addressed how the strong downslope winds are related to typical wind speed over water. The question here is therefore whether there is any suitable information (possibly from other studies or sources), which would support the hypothesis that the observed (offshore) wind speed can be used to estimate the wind speed over the [entire] Dead Sea.

3) The Discussion Section as a whole is, first of all, more a summary than a discussion. Much of what has been stated before is repeated (and the 'discussion' consists to some extent in adding some literature values). The statistical model, for example, is repeated to be good enough (no discussion), rather than addressing potential difficulties (see major comment 2). The 'problem' with the downslope winds (having a stronger wind speed that over the lake, and (probably much) larger water vapor pressure deficit is mentioned - but only mentioned to yield a 'slight overestimation'. Based on what is this called 'slight'? Is it 10% (and would 10% be slight)? Or 50%? (but only occurring during 30% of the hours? – and what is then slight?). I think this would be a discussion. Another point that apparently needs discussion is the fact that the radiation measurement does not 'see' the same surface as the authors want to probe with their EC system, i.e. the lake surface. In fact, I think that either all the aspects, which include Rn have either to be removed from the paper, or an estimate has to be made to establish a method to estimate Rn over water from measured Rn over land (see major comment 1). The discussion then, would consist of the associated uncertainty and the potential impact on the interpretation of the empirical methods (i.e. their performance). Given the relatively large uncertainty in the statistical model, a useful contribution to the discussion would be to test the empirical relations for only a subset of days (for the 1 day averaging period, say), for which the impact of the statistical regression model is minimal (only a few or no estimated evaporation hours, mostly measured values). If then the comparison to the 4 empirical methods (Tab 6) would be robust, this would indeed be an indication that the conclusions regarding the appropriateness of the empirical relations are supported by data (not the statistical model). This last discussion, of course, would only make sense if the 'radiation problem' had somehow been overcome. Finally, an important point for the discussion seems to be that the empirical estimates are relatively good 'average estimates' (28 days) – but do only have reduced predictive (diagnostic, that is) skill for short time scales.

Minor comments

P5, l.4 I don't think I have ever heard of a Rototronic sensor

P5, l. 4 the height of measurement is not given for the radiation, precipitation and pressure sensors. Please specify.

P5, l. 25 I agree with the present authors (in contrast to the other reviewer) that the appropriate conversion should be based on the water temperature and not the air temperature (the differences will be small, though). <w'a'> is already an energy flux (i.e., the kinematic flux of absolute humidity), which is only converted into energy units by multiplication with $L_v$. The corresponding energy (enthalpy) is calculated at the location where the process of evaporation takes place and the relevant temperature is the lake surface temperature (whether this is called $T_w$, the very top of the water or $T_s$, the very bottom of the air, is the same). I don't think that the 'constancy of the fluxes' in the SL is a valid argument for using the air temperature (at 6 m agl in the present case) since the fluxes are only 'constant' within the SL, but not below (note that the SL has also a lower boundary, i.e. the laminar layer with a thickness of some millimeters over water wherein the turbulent fluxes are zero by definition – and rapidly change to their 'atmospheric surface value' at its top). The 'SL theory' (including the constant flux assumption) produces a temperature profile (e.g., eq. 11.12 in the text book of Arya (1988)), which is based on a 'surface temperature, $T_o$ or $T_s$, which actually corresponds to the temperature at the height of the (thermal) roughness length. If we assume an (ideal) stratified SL, stable or unstable, and perfect measurements with a given 'surface' latent heat flux, each measurement height would have a (slightly) different $L_v$ (because of the temperature profile) and hence a different latent heat flux – which is inconsistent with the 'constant flux'. In order to obtain the 'surface flux' from a measurement at any height (within the SL, of course), we must therefore relate the conversion into energy units to a common height, i.e. the thermal roughness length. The actual task therefore in determining $L_v$ is to find the temperature at the height of the thermal roughness length. I think the water temperature is a better estimate for this than the temperature at measurement height. This 'surface latent heat flux' then can be used to assess how many mm of water had been evaporated (what is one of the primary goals in the present study). In any case, two comments have to be added: first, it is in fact potential temperature that has to be used (more precisely, virtual potential temperature) – again the differences (for a 6 m level) are negligible. Second, if $L_v$ is estimated using eq. (3), latent heat fluxes will be in [kW m-2] (since $L_v$ is is in [kJ kg-1]) and not comparable to the sensible heat fluxes from eq (2).
P6, l.17 . . .the mean vertical velocity: this could be mixed up with the mean vertical velocity over the averaging period that is zero in the 'double rotation approach'. In PF, it is the mean vertical velocity over the period that is used to define the plane.

P6, l.20 . . .calculations at sites..

P6, l. 28 below 0.5 'what'? (units)

P7, l.3 . . . data, which . . .

P7, l. 11 is reasonably good

P7, l. 28 is built

P9, l. 5 on longer time scales (or: on a longer time scale)

P8, l. 11ff The presentation of the methods to estimate evaporation is somewhat difficult to comprehend. I try to exemplify this for the Energy balance method. First, one is referred to Tab. 1. The first mentioned aspect of this method is, what is difficult to obtain. I then try to check (find) F_n in Tab 1 (which is apparently difficult to obtain). F_n does not appear in the given equation (but it is 'explained' below in the list of symbols – even if it is not present in any of the given equations). So, I cannot at least judge how this variable appears in the full equation or is related to other variables that do so. The same with the other neglected variables. The 'result' is called V0 (it has an 'X' in Tab 2). Estimating 'somehow' delta_Q produces then 'V2' (why is it V2 for this method, and V1 for the first?). Anyway, it also has an 'X' in Tab 2 (why not a V2?). And some of the other methods also have an 'X' in this table for delta_Q. Overall, I think the overview on the employed approaches should be presented in a much more concise manner. The reader should be able to judge what has actually been done.

P8, l. 15 to make the confusion complete, the 'third method' is then – not the third line in Tab 1 but the fourth. . ...

P10, l. 15 on 26% of the days

[Figure]

P10, l. 16 on about 57%...

P10, l. 18 if the along-valley flow is northerly (with what I concur judging from Fig. 1) the lake breeze would be expected to be perpendicular (easterly on the western shore). Wouldn't this mean that, what was called a 'lake breeze' before (p10, l. 11) is rather a superposition of the along-valley flow and the lake breeze? Can the authors comment on that?

P10, l. 22 in the beginning

P0, l. 25 on individual days

P12, l. 4 on most of the days ...

P14, l. 2 what does 'uncorrected' mean? No Webb correction, etc.? Or do the authors refer to 'only measured, no estimated (with the multiple regression model) values'? (same in Fig. 4). I think the term 'uncorrected' is not appropriate.

P15, l. 1 Maximum values are reached....: see major comment 2.

P15, l.11 the uncertainty...: so, how large was it found to be?

P16, l. 10 evaporation rates of 5.1 mm d-1, …. are measured: do these 'extreme days' contain any estimated values (using the statistical model)? How about the other 'large days' throughout the year?

P16, l. 14 not shown: I think that this 'case' (if it were shown) could serve as a good example to gain some confidence in the statistical model (see major comment 2).

Tab 6 'MD' must be defined.

P17, l. 16 MD is probably mean difference, right? Anyway, the mean differences are not given in the table to demonstrate this (for V0 they are essentially the same...).

P17, l. 20 I suggest to start a new paragraph for the BREB method.

P17, l. 25 the largest

P25, l. 30 22%: judging from Fig. 7 this number is probably valid for a 28 d averaging period

p25, l. 31 coefficients

p25, l. 31 improved the results: I do not really agree. Indeed, the correlation coefficients do somewhat increase (but look at Fig. 6 – both versions would probably serve as examples for 'statistics 101' students for data sets, for which a linear model is not appropriate). At the same time, slope and offset are getting worse (this is why we usually use different statistical measures. . ..). In my view the results of V2 (as compared to V0) simply demonstrate that the calculation of heat storage is not appropriate (see major comment 1)

p17, l. 34 11%: same as above (and also in the following) these values seem to apply for the 28 day averages

p19, l. 7 up to 100%: I don't think I can see this from Fig. 7d

p20, l. 16 with a heat storage term. . ..: in fact, I only now understand what actually V6 does: it fits the Penman equation to account for the missing storage term, right? So, how is this fitting process being done? If it is fitted – as I assume – using the measured evaporation, it does not come as a surprise that a negligible mean difference results. The results from this exercise have to discussed in this light (major comment 3).

p21, l. 8 due to the much higher

p22, l. 12 The BREB, Priestly-Taylor and Penman. . . All these methods do not employ radiation (which has a different field of view – and does not represent the water surface, see major comment 1). I would hypothesize that this is, in the first place, the reason for their bad performance. Only if the Penman method is fitted to the data, it can also produce some reasonable results.

Reference Arya SP (1988) Âă Introduction to Âă Micrometeorology, Academic Press ( San Diego), 1988. No. of Pages: 307

---

## Referee Comment (RC3) · Anonymous Referee #3 · 17 Aug 2017

**Dead Sea evaporation by eddy covariance measurements versus aerodynamic, energy budget, Priestley-Tylor and Penman estimates.**

By Metzger et al.

The paper shows, for the first time, results of direct annual evaporation (E) measurements from the Dead Sea (DS) based on eddy covariance (EC) technic Understanding the annual and the short-term dynamic of the lake evaporation rate is important scientifically in many aspects, for the regional managers and for the future fate of the whole region.

The paper is a clearly written, covering both measurements aspects and evaporation modeling aspects over free water body in exceptional conditions, and one can assume that the measurements were carried under very harsh conditions. Last, there are not many E measurements over water bodies that are based on eddy covariance technique and are comparing measurements results versus different evaporation rate models as the Authors presented here.

Having said that, there are a few significant points the Authors need to address before any publications:

1. Comparing annual evaporation results with previous estimation.
   Comparing to previous works need caution which the Authors have to mention and discuss, including; A. The change in the water level likely changed as well the DS surface area between the different estimation years (e.g., in the case of Stanhill 1994 the lake level was probably 30 m higher and surface area much larger). B. Changes of the climatic conditions due to large-scale changes as well as due to the lake shrinkage. The Authors already mentioned the rapid changes in the reginal Persian trough frequency. C. Likely salinity changes over the years and possibly also the amounts of water removal to the mineral production pools in those years? And D. This work is based on a single measurement year that the Authors mentioned as a relatively wet one.

2. H and $L_v$ calculations (section 3.1) were needed for the energy budget models (as in Tab1). And I assume, though not clearly presented, that ET was derived directly from EC evapotranspiration calculation, not from $L_v$ ?
   However, figure A1 is important in showing that compared with pure water, saline water $L_v$ is lower for temperature higher than ~22C , which likely means that for most times of the year $L_v$ of Dead Sea water is lower than that of a pure water. In this respect, the sentence in L27, page P5 is confusing and future warming and increase water salinity will possibly increase E?

3. Gap filling model for E values when wind direction is coming from the land enhances considerably the total evaporation, especially during the afternoons. However, this model uses VPD (and wind speed) derived from humidity values of air coming from the lake. While the humidity of the land air is probably lower compared to wind coming from the lake. But, it is likely that RH of this dry air  increases as it is blowing over the lake for some distance., Thus VPD and E should decrease.  Shouldn't such effects be estimated, considering its large effect on E? Do the Authors have any information on the RH difference between the two sides of the lake (e.g., west vs. east) for wind blowing to either directions?

4.  Combining or incorporating variables with previous works that have been carried out over the DS in the past to check estimations and assumptions. For example, I found published works on DS surface temperature (*T*om) measurements, and others on the lake heat storage on different time scales. I am wondering why the Authors did not refer to this data?  Δe is highly dependent on *T*om and close to the shore *T*om is warmer than in the open sea, thus it would be valuable if the authors could compare their estimations with independent measurements and its effects on E estimation.
5.  This leads to the last main point: The basis for the uncertainty around E (± 82.2 mm) is unclear. For ecosystems over land, it is generally assume to be ~10%; is it about the same here or? However, although the uncertainty value is about 8% of E it is likely still a substantial large number for water management of the region. Can Authors suggest ways to reduce this in future activities?

A few detailed comments:

1.  L. 9 p. 3. I would look for additional citation(s) for the EC approach reliability to measure E over water bodies.
2.  Is the IRGASON a close or open path IRGA?
    And generally, did the researcher had any problems with the presumable high rusty environment down there, with salt particles etc.?
3.  Heat storage in section 4; can the Authors add 'zero' line in Figure 2, ΔQ value. The impression from inspecting that figure is that the annual value deviate considerably from zero? Is it due to negative heat transfer (e.g., by rain)?
4.  Please add the units for MD and std in Table 6.

---

## Referee Comment (RC4) · Anonymous Referee #4 · 30 Aug 2017

The paper contains very important information on evaporation from the Dead Sea that should be eventually published for the benefit of the scientific and water management communities. The paper presents measured heat fluxes using eddy covariance system over a year; the eddy covariance data is presented with a solid data analysis. In addition the paper evaluates the validity of common used indirect estimations of evaporation from the Dead Sea, which is very important for extending the flux estimates when EC is not available.

However I think that at the present form the paper cannot be published in HESS due to the following reasons:

[Figure]

The authors state that (end of P3) "To measure the energy budget components of the water surface, a fully equipped energy balance station was installed...", however this is not the case. Using eddy covariance - latent and sensible heat are measured directly, but heat storage is not measured nor the net radiation. The radiometer in use is a 4 components CNR4 (P5 L4-5), but as can be seen in Fig 1b the two lower half spaces are directed to the ground not to water surface; this is also acknowledged later on, but this should clearly be stated upfront. Water temperature is calculated not measured, this is found later on in the paper, should appear upfront when declaring for fully equipped energy balance station. In the results it is stated that the heat storage is calculated as the residuum of the energy balance (Rn-LE-H), where Rn by itself is not really measured, again, this is not a fully equipped system. I think the paper should be written with emphasis on the existing data, which are very important and worth publication, but not declaring for measuring energy balance, this gives a wrong impression for the reader. As stated above, water temperature was calculated, not measured, termed TMO. TMO is used for the examining the equations of evaporation versus measured evaporation (e.g. table 1 and large portions of the figures). TMO is also used to determine the saturation vapor pressure at water surface temperature, Ew. Ew is used to determine the vapor pressure deficit and for the evaporation estimates. So both water temperature and Ew are not measured, but they have a very important role in the analysis and the conclusions of the paper. Again, I this think that it would be better to orient the paper to the existing information that was measured and analyzed and not to rely so much on the computed meteorological parameters. Typically, when EC measured evaporation is compared with the evaporation equations it is done based on measured meteorological parameters, not on computed parameters.

Overall, due to these weaknesses, there is a gap between the actual measurements and the interpretations and conclusions. The scientific methods and assumptions should be better declared earlier and should appear clearer. I think that the first half of the title "Dead Sea evaporation by eddy covariance measurements" is good and representing the novel aspects of this work, but the second half of the title represents the

weaker part of the paper.

---

## Author Comment (AC1) · 29 Sep 2017

**Reply to Reviewer 1**

**1 General Comments**

*I think this paper deserves publication because the dataset is particularly interesting. As said by the authors, it is the first time that one year of turbulent atmospheric fluxes measured by the eddy-correlation (EC) method has been presented for the specific area of the Dead Sea. The data have been professionally processed. They can efficiently be used to assess several parameterizations generally used for long-term measurements when the EC method is not available. The originality is that the authors provide several levels in the parameterizations, according to the measurements that can be performed. However the paper requires some important corrections before being published. I try to describe them below. Some corrections, related to the methodology, are essential. Other are secondary and consists in numerous details that could be improved.*

Thank you for the very detailed and insightful review. We are sure they will help to improve the paper. Responses to individual comments are provided below. Reviewer's comments are in italic.

**2 Specific Comments**

*Methodology : I think the difficulty comes from the fact you make a local measurement over ground (with the EC method) while you would like to include the close environment in the driving parameters of the turbulent fluxes, considering that the air that is advected on the measurement site is (most of the time) characterized by the water surface temperature and water surface vapor partial pressure. I am not against the idea, but I think the way you deal with this assumption is not always correct. I am aware that you want to prove that measuring on the headland, very close to the seashore is equivalent to measuring with a raft in the middle of the sea, but I am not totally convinced. Another issue is the fact that you address different time scales for the energy budget you consider, without saying accurately which timescale you refer to.*

*C1) When measuring the latent heat flux LE at level 6m with the EC, the Lv value (kJ/kg) that has to be used to change the evaporation rate $\overline{w'a'}$ into a flux is that of the air, i.e. 3148.4-2.37 Ta and not Tw as you use in Eq. 3. Ta is the air temperature in K. Even if the evaporation takes place at the water surface as you mention p 6, line 3, LE is assumed to be constant in the surface layer (in fact, not to vary more than 10% of its surface value).*

Thank you for this comment. As Lv is a water specific variable and defined by the temperature of the liquid, in our opinion Tw is the appropiate variable to calculate Lv. The literature equation $Lv = 3148.4 - 2.37 \cdot T$ is derived via the Clausius-Clapeyron equation, by measuring water temperature and saturation vapour pressure. This results in the enthalpy of vaporization. Dividing the enthalpy of vaporization by the molar mass of water gives us the latent heat of vaporization. So it does not include air temperature measurements and is only a function of water temperature.
When using Lv to convert the evaporation rate $\overline{w'a'}$ into LE, which is the energy used to evaporate the water at the surface, from a physical point of view, the water surface temperature and not air temperature has to be used as Lv is a water specific variable Lv=fct(Tw).

*This unique Lv value should be used for both offshore and onshore winds. In fact an internal boundary layer develops either inland or offshore, depending on the wind direction. Perhaps you should mention it and clarify the parameters you use in both situations. [The only opportunity to use Lw as you define it, would be to take*

*into account the heat loss of the water due to water evaporation, as shown by Giadrossich et al., 2015 in their eq (2). This eq. applies to the energy budget of the whole sea, and not the local energy budget that you quantify at the EBS].*

As in this paper we are only interested in fluxes from the water surface only fluxes measured during onshore wind conditions are used, because then the source area of the flux is over water. All flux measurements for offshore wind conditions are neglected as they represent the fluxes from the land surface. To fill the so created gaps in the time series the regression model is used.

At this point we take advantage of the very periodic wind systems at the Dead Sea. There are only very short time frames with westerly offshore winds where the flux measurements have to be neglected and therefore a large data set with onshore wind conditions is available to establish the regression of the latent heat flux from the water surface with wind velocity and water vapour pressure deficit. With this so gained regression we calculate evaporation from the water surface for the time steps with westerly winds as no measurements of the fluxes from the water surface for this conditions are available. With this procedure we get the full diurnal cycle of the evaporation from the water surface, just as it would be measured with a raft station in the lake. As only flux data for onshore wind conditions are used, $\overline{w'a'}$ is always converted into LE with Eq. 4:

$$Lv = 5150.6561 - 13.9530 \cdot T_w + 0.0162 \cdot T_w^2$$

(This equation is slightly different to Eq. 3, which is for pure water, as the salinity of the water also influences the latent heat of vaporization.)

For the investigation of the energy balance of the land surface no data from this station is used as we have another station further inland, which is not affected by the water surface. The results of the energy balance of the land surface are not discussed here as they are beyond the scope of this paper.

We will revise the paper to make the work flow and the used equations and parameters clear. Thank you for pointing this out.

*C2) When discussing the various models you apply to your data, you use Tom-Ta or Ew-Ea. The latter should be replaced by $E_{surf}$ - Ea, with $E_{surf}$ (or another name) standing for the water vapor partial pressure at the surface (water or ground). The former, by $T_{surf}$ - Ta:*

- *For onshore winds, as the source area is over water, we think that the similarity profile you use in Appendix B, p26, to deduce the water surface temperature is not appropriate since the regressions you wish to establish in the following will depend on this profile. We suggest that you'd try to find independent remote-sensed measurements of the Dead Sea surface temperature instead, which is not exactly the air temperature at the surface, but is the closer you can find. If you do so, we am almost sure that the discrepancy between panels 2 and 3 in Fig. 2 will be larger. In that way, the models you will apply in the following will include independent measurements (since the temperature difference will not result from the similarity profile).*

  We agree that remotely sensed water surface temperature would be favourable as it is independent from the air temperature measurements. This was also considered and discussed when analysing the data but it was discarded because of the following reasons:

  – Nehorai et al. (2009) used Meteosat Second Generation (MSG) data to estimate water surface temperature from the Dead Sea. For retrieving the water surface temperature from the satellite data the operational SST algorithms could not be applied as they are calibrated to mean sea level and do not take the additional 421 m atmospheric layer in the Dead Sea valley into account. They derived the water surface temperature by calibrating their algorithm against in-situ measurements. Unfortunately, we did not have the necessary in-situ measurements of the water surface temperature to follow their procedure to derive the water surface temperature from satellite data.

  – Furthermore, Nehorai et al. (2009) raised concerns, that on days where the Mediterranean Sea Breeze, a strong westerly wind, enters the valley in the afternoon the enhanced evaporation causes enhanced water vapour and thus a stronger absorption of thermal IR radiation which leads to a screening of the

Dead Sea surface and thus incorrect estimates of the water surface temperature. For their studies they excluded all data with these conditions. This would lead to data gaps in the time series of water surface temperature during westerly (offshore) wind conditions, meaning that no water surface temperature data would be available for the timesteps where it is actually needed to estimate evaporation from the water surface, as the station measures evaporation from the land surface for offshore wind.

– Another point why satellite data was not used is the need of a continuous time series. Satellite data can not be used for cloudy conditions. So especially for the winter months cloud cover would reduce data availability strongly.

– Because of the aforementioned problems with satellite data we followed the advice of another paper from Nehorai et al. (2013), which shows that "SST is highly correlated to air temperature ($R^2 = 0.93 - 0.98$) in all seasons". Based on these results the similarity approach was used to calculate water surface temperature from air temperature.

– Of course there is dependence in the data, but we also checked the results of the similarity approach with a short term experiment of about 5 days, where longwave and shortwave radiation was measured directly over the water surface. From the outgoing longwave radiation radiation temperature of the water surface was calculated and compared to the calculated Tw using the similarity approach. A correlation of 0.8 was achieved. This is quite good, considering the uncertainty of the radiation measurements, as the radiation sensor was getting covered with salt through the spray over the course of the 5 day experiment.

*For offshore winds, the source area is partly over water, partly over land. The $\Delta T$ estimation should be a combination of ground surface temperature and water surface temperature. There again, an estimation of the ground surface temperature (perhaps after some assumptions of the emissivity) would be appropriate. You could perhaps also use the upward longwave radiation flux measured at the neighbouring ground station.*

For the multiple linear regression approach, data from offshore winds are excluded. So land surface temperature is not needed. As the regression model is used to calculate the evaporation from the water surface for offshore wind conditions, the water surface temperature is needed and can be calculated with the MO Theory.

- *For offshore winds, I do not know any mean to deduce $E_{surf}$ , unless you make assumptions on the water vapour at the ground surface. You could use the similarity profile, but meanwhile you would make the choice of a model and Sections 3.3 and 4.4 would become useless. For onshore winds, $E_{surf} = 0.65 \cdot Ew(Tw)$ could be an appropriate estimation, Tw being the satellite sea surface temperature, instead of Tmo. A sensitivity study of the regressions to the error in Tw (which determines ew) could also be informative.*

This paper focuses only on the energy balance of the water surface. The energy balance of the land surface is not discussed and beyond the scope of this paper. For investigating the energy balance of the land surface we have a different station which is further inland and therefore not influenced by the water. Therefore there is no need to deduce $E_{surf}$ from the data set.

*C3) If we consider the 3 objectives you propose to fulfill : i) is fulfilled by the EC method, but you do not need to use any multiple regression model to quantify the offshore conditions : you directly measure them. The way you can link these flux measurements with local air or surface parameters is another issue.*
Yes the offshore conditions are measured but not the fluxes from the water surface for offshore conditions. For offshore wind conditions the source area of the measured flux is the land surface. As the aim is to get the full diurnal cycle of the flux from the water surface (which would be measured if the station would be located on a raft), it is necessary to apply the multiple regression model and estimate the flux values for offshore wind conditions.

*ii) I think you cannot totally achieve this aim since you can only access to the local terms of the energy budget and make assumptions on the terms you have to neglect.*

We fully agree. We can only get the local evaporation at the measurement site and not the evaporation for the whole lake. This point will be rephrased that it will become clear that these are local values.

*iii) OK if you define what 'evaporation' is. You could also add that you want to assess the capacity of these models to retrieve the 'evaporation' term, in the future, when the EC sensors are not available any more (you suggest this in the conclusion).*
we will add your suggestions to point iii.

**3   Other Remarks**

Thank you very much for the detailed and helpful remarks. First, we will provide answers to the questions raised by the reviewer. Afterwards we list the comments about readability and linguistical problems. We will consider all of these comments in our revised version of the paper and will revise the text accordingly.

**3.1   Questions**

*13 - p2, line 7 : during which period this decrease happened ? And 60-400 denotes a very large variability. Can you explain why ? (variability among the authors ?)*

The decrease was caused by the construction of dams and canals along the Yarmouk river and Lake Tiberias/Kinneret mainly between 1955 and 1964. Since then only about 10% of the natural discharge of the Jordan river enters the Dead Sea. The variability of the inflow results from variability among the authors.

*17 - p2, line 16 : ← 'shifting of the fresh/saline groundwater interface' ('of the' has been added). Please define sinkholes, I did not know this phenomenon.*

Sinkholes are holes or depressions in the ground formed by subsurface erosion or removal of soluble bedrock and the collapse of the surface layer.

*18a - p2, line 25 : why westerly winds would be harmful, compared with easterly winds ? You could also delete 'the' at the end of the line, before '1940'. I suppose this harm comes from the fact that easterlies carry drier (continental) air, whereas westerlies carry moister air. But this cannot be guessed from what you wrote here.*

Thee westerly winds have often high wind velocities enhancing the evaporation and thus accelerating the lake level decline.

*29 - p5, line 5 : is it a tipping bucket rain gauge ?*

Yes, it is a tipping bucket rain gauge.

*34 - p5, eq (2) and line 24 : usually, $LE = \rho_a Lv\overline{w'r'}$ or $LE = \rho_a Lv\overline{w'q'}$ where r and q are the water vapour mixing ratio and specific humidity, respectively. Brutsaert (which you refer to in the following) uses the latter definition for the evaporation rate : $Ev = \rho_a\overline{w'q'}$. In addition, if I remember well, the hygrometer converts the absolute humidity in water vapour mixing ratio, using T=20°C and P=1013.25 hPa. Perhaps you could consider using r or q instead of a.*

We can of course change eq(2) to the more common form with specific humidity. In our calculations of the turbulent fluxes with the EC software we use absolute humidity as this is the raw output of the IRGASON.

*39- p6 or before (p5) : did you use a constant calibration coefficient for the IRGASON (when was the calibration done ?) or did you calibrate the hygrometer measurements against the low frequency humidity*

*measurements ?*

The IRGASON was calibrated before the experiment but not in between, so constant caplibration coefficients were used.

*41- p6, line 29 : 'when the variability of the signal' . Could you define the variability (standard deviation/average ?) ? I think it is 0.6 and not 0.6%*

This means that if 10 min averaged normalized signal strength varied from one time step to the next more than 0.006, which is 0.6%, the corresponding 30 min flux value was excluded. This procedure was introduced due to the fact, that we found an increasing variation of the signal strength with the decrease of the total signal strength. Most likely due to contamination of the glass window of the instrument.

*46 - p7, subsection 3.2 : once you have made the corrections indicated in point 2, I suggest that you keep on calculating the multiple regressions, but not with the aim of parameterizing the offshore conditions. Rather, to show how the classical relationship between the latent heat flux and the wind and/or vapour pressure deficit behaves, under the specific conditions of a semi-arid area that is influenced by sea breeze, slope breeze or both. The multiple regressions should be done for onshore and offshore data separately (provided that 19% of the dataset is enough to apply your method to the offshore data). Note that the regressions you examine are similar to the aerodynamic (or bulk model) from Brutsaert. That is why I am not surprised by the good correlations you obtained in Table 4 for V0. That is also the reason why I do not agree with the idea of using these multiple regressions to calculate offshore winds. I also think that the presentation of the simple regressions should be shorter since they are known to fail to represent the flux but they can serve as a base to compare the multiple regression $H = f(U;\Delta T)$, to the regression $H = f(U \Delta\Theta)$, and the same with LE, U and $\Delta e$. Please also think of using $\Delta\Theta$ instead of $\Delta T$, although the difference will be very small. You may have noticed that I added a multiple regression $H((U;\Delta T)$. The reason is because the BREB method is partly based on this correlation.*

We think that the reviewer suggested the additional multiple regression for offshore data based on its wrong impression of our work flow at the beginning (also discussed in C1+C2). As we already clarified in our answer to C1+C2 that we do not use the offshore data for any calculations, there is, in our opinion, no need to establish a regression model for this part of the data.

The second point of the reviewer was to shorten the results of the simple regression results (P.13 l.1-6) and add a comparison of the multiple regression with the simple regressions. We will go over the paragraph and will add the following conlcusion to it: it can be seen from the comparison that wind speed has a much stronger impact on evaporation that the vapour pressure deficit (VDP) in spring, summer and winter, but that in autumn the VDP has an considerable impact as well.

The third point raised in this comment was a regression for the sensible heat flux with different variables. This was already done during the analysis. The multiple regression of $H=f(U,\Delta T)$ achieved a correlation of 0.93. These results were not presented in the paper as we wanted to keep the focus on the latent heat flux.

*50 - p8 : I would substitute 'aerodynamic method' for 'aerodynamic or mass transfer method'. In fact it is the bulk method, frequently used to estimate surface fluxes over the sea, where the EC method or dissipative method are not easy to implement. Brutsaert (1982) refers to it as the 'bulk transfer' method and it is based on similarity profiles assumptions and the relationship between fluxes and wind or scalar gradients (through the Dalton or Stanton numbers). According to Brutsaert (p88 in my edition, reprinted in 1984)*

$$Ev = C_e\rho_a v_a(q_{surf} - q_a) = C_e\rho_a v_a(e_{surf} - e_a)\frac{0.622}{p} \tag{1}$$

*Without telling it, you assume equal transfer coefficients for evaporation and momentum ($C_e = C_d$). $C_e$ is a mass transfer coefficient for evaporation and $C_d$ is the drag coefficient $= \frac{u_*^2}{v_a^2}$ . Introducing the logarithmic wind velocity gradient under neutral conditions, which is a second assumption, Ev becomes :*

$$Ev = \frac{k^2}{(ln\frac{Z}{zo})^2} \rho_a v_a (e_{surf} - e_a) \frac{0.622}{p} \tag{2}$$

*So $K_E$ you identify in Table 1 should not contain $\rho_w$. I suppose you needed to add it since the kinematic flux you calculated is in term of absolute humidity (but you should have divided Ev by $\rho_a$ and not $\rho_w$).*
*To conclude, I suggest you add a remark concerning the assumption that $Ce = Cd$. Perhaps this could be explained in an additional Appendix (3). I also suggest that you move into the present subsection, your sentence from p17, lines 12-13 : 'the aerodynamic approach is the only approach designed for sub-daily time intervals'. And you could add '(typically 30 min in this study)'. I find that you well address this timescale issue in the following, specifically with Table 6 where you show the results. However, it should be also clearly mentioned in subsection 3.3 (for the 4 models).*

Thank you very much for this comment. We will follow your suggestions and add an explanation about the assumptions and steps we made, including Ce=Cd, and the used of the logarithmic wind profile. We will also address the issue that only the aerodynamic approach can be used for sub-daily calculations and correct the text accordingly.

*57 - The hysteresis model from Duan and Bastiaanssen, $\Delta Q = aRn + b + c\frac{dRn}{dt}$ should be described including the discussion about the term $\frac{dRn}{dt}$ . Note also the dependance of c on the range and variability of the water surface temperature. V3 is a specific case of V2 where b=0 and c=0, a being obtained as 'the deviation of the default version from the measurements'. Did you try to determine specific (a,b,c) for your own dataset, just to quantify the deviation relative to Duan and Bastiaanssen's results ? I do not suggest to include them in the models you use, since, doing so, you would invalidate the V5 regressions.*

Thank you for this comment. We will add some additional explanation about the hysteresis model to the paper. We will also explain that in our analysis we first used the proposed ansatz from Duan and Bastiaanssen $\Delta Q = aRn + b + c\frac{dRn}{dt}$, and calculated specific a,b,c from our data set. This was done, as Duan and Bastiaanssen stated in their paper that 'a,b, and c vary largely among lakes. [...] and that lake-specific coefficients should be determined'. Then we investigated the special case b=0 and c=0. V5 is an additional 'special' case were we repeated the calculations using the hysteresis model and the coefficients from Kohler and Parmele, which means V2 and V5 are not the same.

*58 - Could you please be careful to discuss the assumptions of the other two models and give additional information for Kohler and Parmele's work ? I would not say that Ts has been removed (if I understood correctly) but that it has been estimated from the long-wave radiation flux. This is no-doubt an improvement, relative to your initial estimation from the similarity profile. Please use the same symbol to design the water temperature as the one you have used in the previous section, unless you want to distinguish it on purpose.*
Thank you for the comment. We will revise the description of the sensitivity studies that it will become clearer to the reader. Kohler and Parmele did not estimate the surface temperature from longwave radiation but removed $T_s$ by doing following approximation. He used the first two terms of a binominal expansion

$$L \uparrow = \epsilon \sigma T_s^4 = \epsilon \sigma (T_a^4 + 4T_a^3(T_s - T_a)) \tag{3}$$

and then used $T_s - T_a = (E_s - e_a)/\Delta$ to derive following parameters

$$L' = L \downarrow - L \uparrow = \epsilon_w \cdot L \downarrow - \epsilon_w \cdot \sigma \cdot T_a^4 \tag{4}$$

$$\gamma' = \gamma + \frac{4 \cdot \epsilon_w \cdot \sigma \cdot T_a^3}{K_E \cdot \rho_w \cdot L_v \cdot v_a}. \tag{5}$$

*62 - Fig. 2 : Please add 'prec' after 'daily precipitation amount' in the caption. Which temperature is Ta : ultrasonic at 6m, BetaTherm probe at 6m, HC2S3 probe at 2m? Please represent $T_{surf}$ instead of Tmo. You could also show $T_{surf} - T_a$. Please represent $e_{surf} - e_a$, and perhaps also qa. I suppose the air is very dry during summer. It would be interesting to show the annual evolution of the air specific humidity. 200 $Wm^{-2}$ are enough for the H vertical axis. It would be convenient to add a thin line for 0 $Wm^{-2}$ in the lower panel. Did you try to represent the daily average parameters (Rn, H, LE, $\Delta Q$) on the same graph (with a*

*more appropriate scale on the y-axis), to be able to see the phase shift between the annual maxima and also whether the Rn variation relative to the time is linked or not with $\Delta Q$ (in relation with the hysteresis model from Duan and Bastiannssen).*

Ta is 2m temperature. The use of $T_{MO}$ and not $T_{surf}$ was discussed in C2. As explained, there is no way to determine $T_{surf}$ from satellite data and therefore we will keep $T_{MO}$ and $e_{MO}$ in the graph, but we will try to present the daily averages in the graph as suggested from the reviewer. If the graph is still well readable we will change the graph in the revised version.

*65 - p10, line 6 : 'the annual precipitation normal of 80 mm' : the word normal is not accurate. What is the period considered by Goldreich, 2003 ?*
The period which was considered by Goldreich was the climatological standard normal period 1961-1990.

*68 - p10, lines 14 and 15 : are 32 and 26 % relative to lake breeze or lake breeze+ synoptic conditions ?*
These values are relative to the total amount of days in winter.

*76 - p10, discussion on the energy budget : you assume that the energy budget is closed and you never discuss the frequent non-closure energy budget problem that is reported in several studies in the literature (see Foken et al. in Aubinet et al. p108-109). You cannot avoid this discussion, even if there is no mean to estimate the error, especially because your measurements are done at the boundary between the marine and the continental surface layers. Under these conditions, the surface change may generate large scale heterogeneities that are unlikely to be correctly taken into account by the local measurements. You show in Fig. 3, extreme values of $\Delta Q$ that are of the order of the net radiative flux in spring and summer. It is unlikely to be true. Anyway, it is known that the shorter the timescale, the larger the non-closure. By contrast, the average $\Delta Q$ (daily average) is about 100 $Wm^{-2}$ during summer and decreases down to a few tenth of $Wm^{-2}$ in winter, which are rational values (I remember that LE has to be recalculated, but it should not be very different).*
Thank you for the comment. We agree with the reviewer that for every EC system there are uncertainties and a possible non closure of the energy balance. The reviewer is right, that we should address this point in context of the calculation of the heat storage as a residuum. So we will address this point in the context of section 4.1.

**3.2   Linguistical Comments**

Thank you very much for these detailed comments. We will consider all of these comments in our revised version of the paper and will revise the text accordingly.

*5 - The term 'evaporation' you frequently use is not accurate enough. I will point it out in the following.*
*6 - p1, line 6 : 'total annual amount measured' →'total annual amount of evaporation measured' (in this case, you do not need to be more accurate since you provide the evaporation unit).*
*7 - p1, line 7 and further on : 'vapour pressure deficit' ←'water vapour pressure deficit'*
*8 - p1, line 8 : 'Consequently' is not appropriate. Perhaps 'in fact' could be used instead. What do you mean by 'evaporation amounts' ?*
*9 - p1, line 10 could be changed to ←'during daytime. During nighttime, evaporation rates are also larger than the daytime evaporation rates, due to strong ...'. Why do you use 'evaporation rate' this time ?*
*Note that this result will perhaps require corrections, in light of what is said in the following (see my final remarks for instance).*
*10 - p1, line 11 : The link 'Furthermore' is somewhat awkward. You should explain here why you calculated the regressions. By the way, I think that the multiple regressions should be established for another purpose (see my remark 46-).*
*11 - p1, line 14 is clear and nice. I skip lines 15, p1 to the Introduction, p2.*
*12 - p2, line 6 : you could add 'down' after '90%'.*
*14 - p2, line 9 : 'The total amount is about' ←'The total amount of loss is about'*
*15 - You could add a budget equation such as : $10^6$ (400+240-250)+evaporation = -650 $10^6$, which gives*

evaporation = -1060 $10^6 m^3 a^{-1}$, which is in the range 700 - 1400 $10^6$, indicated by Gavrieli et al., 2006.

16 - p2, lines 13-14-15 : I would replace ' Evaporation is not only ....environmental problems.' by something like 'It is important to assess the water budget components of the Dead Sea for a climatological purpose, but it is also a priority for the people and the socio-economic development of the region to anticipate the evolution of these components and the consequence for the environment. For instance, the lake level decline causes severe environmental problems . ' (you may of course change the words).

18b - p 2, line 29 (no link with 18a) : I do not understand 'especially' in this context ('In addition' ?). Perhaps you could also mention whether it is a fresh water fish. Ein Feshkha reserve ?

18c - p3, line 5 : I would add 'in the evaporation estimations' just before '(Stanhill, 1994'.

18d - p3, lines 6-7 : I would remove : 'Furthermore, the governing factors of the Dead Sea evaporation, e.g. wind velocity, vapour pressure deficit, or net radiation, have to be identified, to validate the indirect methods'. These parameters are governing factors every where in the world and you do not have to prove it. I think this sentence is confusing at this point.

18e - p3, line 8 : 'with a high temporal resolution' instead of ',in high temporal resolution'.

19 - p3, line 10 : 'continues' ← 'continuous'

20 - p3, line 12 : according to Wikipedia, it seems that Lake Kinneret is the same as Lake Tiberias mentioned by Kottmeier et al. (2016) (it is only a remark, you choose the name you prefer). You could perhaps add that it is crossed by the Jordan river which partly feeds the Dead Sea. It is not a major piece of information but I find it nice to provide the reader an idea of the geographical environment.

21 - p3, line 13 : 'to the authors knowledge' instead of 'as to the authors knowledge'.

22 - p3, line 14 : 'Therefore, long-term eddy covariance measurements are conducted' ← 'That is why, in the frame of the international DESERVE project (Kottmeier et al., 2016), long-term eddy covariance measurements were conducted'

23 - p3, line 17 : 'provided' and 'was' instead of 'provides' and 'is'

24 - p3, lines 17-19 : I perhaps misunderstood but it seems to me you did not use the data from these stations. Is it useful to quote them ?

25 - p3, lines 19-21 : 'Provide', 'Evaluate' and 'Evaluate' instead of the same with 'ing'.

26 - p3, Measurement site : it is difficult to distinguish in Fig. 1, how far the Judean mountains highest submits are from the lake and what their height is (hills or mountains ?). The Moab mountains are clearly shown.

27 - Fig. 1 : you forgot to write 'Jordan river' in panel (a). Landsat with an L in the caption. The red arrows are a little confusing in panel (b) : simple lines instead of arrows would be enough.

28 - p5, line 4 : Rotronic

30 - p5, line 7 : please add 'open path' before 'integrated gas analyser'. I suppose that at 6m, it did not suffer from spray, even under strong onshore wind conditions ...

31 - p5, line 9 : 'From the 20 Hz data evaporation was calculated using the eddy covariance technique'→ 'The latent heat flux was calculated from the 20 Hz data using the eddy covariance method.'

32a - p5, line 9-10 : ← 'The principle of the method, the post-processing and data quality control steps are presented in Sec. 3.1'.

32b - p5, lines 11-12 : please consider the multiple regressions again (after 46-)

33 - p5, eq (1) and line 23 : the ultrasonic anemometer provides the virtual air temperature and not the air temperature Ta. The Schotanus correction is made, as you say in Sec. 3.1.1, to take this point into account.

35 - p5, eq (3) and line 27 : as said before, Tw should be Ta.

36 - p5-6, lines 27 to 5 : the text should be deleted from 'For salt water ..' to the end of the subsection.

37 - p6, line 8 : 'measurement limitations' you could add 'of the sensors'

38- p6, lines 18 to 24 : it seems to me that the order should be : spectral corrections, Schotanus correction and Webb correction unless you applied an iterative process.

40- p6, line 28 : 0.5 g/kg ?

42 - p6 line 31 to p7 line 2 : you say too much or not enough. It would be nice to describe the tests and to say what ITC is.

43 - p7 and further on : I would replace 'fetch' by 'source area'.

44 - p7, lines 6-7 : please consider again the rejection of these data : I agree that it is important to distinguish them from the onshore measurements data, but the fluxes are what they are and do not have to be rejected. Nevertheless, it is important to quantify this contribution since the source area may be different.

*45 - p7, lines 7 to 9 : I would replace the 2 sentences 'For southerly wind directions ... 600m away from the headland.' by 'For southerly and northerly wind directions the source area is over water and the average source area contributing to 80% of the flux ranges from 0 to 300 m and 0 to 600 m, respectively, ahead of EBS.'*

*47 - p8, line 1 : 'Indirect methods to estimate evaporation' ←'Description of four indirect methods to estimate evaporation'*

*48a - p8 : I would move the first two (essential) sentences of subsection 4.4 ('For the calculation of evaporation -please insert a comma here-, several equations, based on .... at least 7 days') to the beginning of section 3.3.*

*48b - p8, line 5 'an overview of which sensitivity study is performed' (I added 'of').*

*49 - Ev is not defined. I suppose it is the evaporation rate defined as $Ev=\rho_a \ w0q0$ (Brutsaert, 1982).*

*51 - The sensitivity study, referred to as V1, is performed to address the stability issue. The presentation of the stability factors you refer to (Cline 1977) p. 17 should also be moved to the present subsection. I would also describe the new KE (including the stability factor) in Appendix 3. You could also add that the stability functions for wind and heat are expressed in terms of the bulk Richardson number, which allows estimating the stability when the turbulent fluxes are not known.*

*52 - Energy budget : Here again, $\rho_w$ should vanish if you use the specific humidity instead of the absolute.*

*53 - I suggest that you write down the budget equation as Giadrossssich, 2015 did (their eq 1) : Rn + Anet = LE + H +$\Delta Q$ , where Anet is the net heat advected into the lake (by stream flow and precipitation minus the heat loss due to evaporation minus the heat transferred at the bottom of the lake) and $\Delta Q$ is the heat storage per unit area in the lake (for most cases) or in the ground (for specific cases with strong offshore winds. Under these conditions, Anet can be ignored). This energy balance applies to timescales larger than the day due to the advection term that cannot be known at a short timescale.*

*54 - With V0, you neglect $\Delta Q$ and Anet . [Note that the resulting reduced budget equation can also be applied at a sub-daily scale.] Neglecting $\Delta Q$ is a coarse assumption that is valid only under specific and occasional conditions. I think you should mention it at first and say that V0, even unrealistic, is a basis for the BREB method that will be improved by V1 and V2. I mean that V1 and V2 should not only be considered as sensitivity studies for V0, but as 2 alternative methods for V0.*

*55 - p9, line 8 : you forgot to mention the water vapour deficit.*

*56 - $\beta$, the Bowen ratio should be defined as H/LE. When fluxes are unknown, $\beta$ can be approximated by the expression you give, provided that $K_\Theta$, the Stanton number for temperature = Ce, the equivalent for evaporation. Also, be careful not to use the same symbol for the Bowen ratio and the activity of water in Appendix A (I would keep $\beta$ for the Bowen ratio).*

*59 - Table 1 : caption : default versions (V0) in Sec. 3.3 and 4.4. (just add 3.3) Fn and $\Delta t$ are not used and $\rho_w$ should not be used. Please take into account my remarks in 2- Priestley-Taylor is presented as the 3rd method in Sec 3.3.*

*60 - Meteorological conditions : this subsection will have to be read again after new LE, $T_{surf}$, $\Delta e$ values ...*

*61 - For some parameters, the daily values and their evolution are also interesting to discuss in addition to the extreme values.*

*63 - p10, line 4 : 'long term annual mean' : during which period ?*

*64 - p10, line 5 : please add 'sometimes' before 'exceeded' 66 - p10, line 7 : 'made' instead of 'makes'*

*67 - p10, line 9 : 'only during the winter seasons a different behaviour was found' ←except during winter, when the wind increased in connection with the convective activity, a different behaviour was found'*

*68 - p10, lines 14 and 15 : are 32 and 26 % relative to lake breeze or lake breeze+ synoptic conditions ?*

*69 - p10, line 15 : Please indicate the direction of the downslope breeze (north-westerly)*

*70 - p10, line 16 : 'yielded' instead of 'lead to'*

*71 - p10, lines 18, 19, 24 : 'exceeding' instead of 'of over'*

*72 - p10, line 22 : 'November a Red Sea Trough with a central axis advected dry and warm air'→ 'November, when a Red Sea Trough advected dry and warm air'*

*73 - p10, line 25 : 'However, at' ←'However, on'*

*74 - p10, line 26 : 'in winter latent heat flux values' ←'in winter some latent heat flux values'*

*75 - p10, line 27 : the energy balance equation should have been shown in subsection 3.3. You only need to say that at this timescale (24h), Anet is ignored. Do you think it is still correct after rainfall ?*

*77 - p10, line 31 : please reword 'used for heating the lake, which is stronger in spring than in winter'. This*

*is grammatically false.*

*78 - p10, line 33 : I do not see that $\Delta Q$ is negative in winter.*

*79 - p 12, fig 3 : it is usually required to indicate the delay between the UTC and local time. You can keep LT and indicate, on the first time (UTC +3h). It would be also interesting to add in the caption the approximate time of sunrise and sunset, especially from Spring to Autumn.*

*80 - p12, line 4 : ←'and, thus, [at] most of the [days] data within this time frame' (remove the [words]).*

*81 - p12, line 6 : ←'also for the study of the intra-annual .... this gap' (instead of these gaps).*

*82 - p12, line 6 : 'A multiple regression model was applied ... for offshore conditions' : As said before, I do not agree with this method : I'm waiting for your decision, regarding the suggestions I made in 2-*

**References**

*Nehorai, R., Lensky, I. M., Lensky, N. G., and Shiff, S. (2009). Remote sensing of the Dead Sea surface temperature.* J. Geophys. Res. Oceans, *114(C5). C05021.*

*Nehorai, R., Lensky, N., Brenner, S., and Lensky, I. (2013). The dynamics of the skin temperature of the Dead Sea.* Advances in Meteorology, *2013:1–9.*

---

## Author Comment (AC2) · 29 Sep 2017

**Reply to Reviewer 2**

**1 General Comment**

*This a very interesting paper, addressing an important environmental issue (evapora- tion from the Dead Sea), presenting very important data and (apparently for the first time) directly measured evaporation rates and thus adding important new information to our knowledge. There is no doubt, therefore, that the paper should eventually be published in one or the other form. There are three main pillars: first, to measure, directly, the lake's evaporation using an EC station on the shore (therewith employing the footprint of the water surface). This provides direct measurements for about 70% of the time. I think this approach is well motivated, well explained and also 'well executed' (all the necessary data treatment, corrections, QC, etc.). To estimate, as a second pillar, evaporation during the remaining 30% of the time, a statistical model is trained using the onshore wind conditions and the available in- formation during these conditions, to esti- mate lake evaporation during offshore wind direction. This is very appropriate (and possibly novel) as an approach, and so is the statistical approach, its presentation (and results). However, it has the drawback that the estimated evaporation rates are among the largest during the whole year and thus contribute a substantial fraction of the total (yearly, monthly, daily – the shorter the more variable) evaporation. Therefore, the statistical model does not only have to be tested using the 'usual' tests (cross-validation, etc.) based on the available onshore conditions (what is convincingly being done), but attempts should be made to support the hypothesis that the same statistical model applies (yields the claimed 'good' statistics) when the input data stem from offshore situations (high wind speed in combination with low water vapor pressure deficit). Some suggestions are provided in major comments 2 and 3. As a third pillar, empirical estimates (for evaporation) are compared in their performance to the measured evaporation. This is adding a great deal of value to the paper. Unfortunately, for three out of four methods the comparison is not valid (because net radiation is not measured over water – which basically invalidates all the estimates; see major comment 1). Furthermore, the estimation of heat storage (in the water) is also flawed (see again major comment 1), so that all the different 'versions' tested are not really conclusive (with the exception of V1). Indeed, the results show that all the empirical methods using net radiation give results quite different from the observations (if only measured – and not estimated – values would be considered, this finding would probably be even more pronounced). The only 'reliable' empirical method is the aerodynamic approach (not using Rn). The problem with the different fields of view (for radiation and turbulent fluxes) could somehow be overcome (for example by using satellite – or other – observations [even literature values would be better than nothing] to correct for albedo differences between land and water, and by using the [simultaneously measured] land surface temperature [from the two additional EC sites over land] and the estimated lake surface temperature to correct for different longwave outgoing radiation). To actually estimate heat storage in the water from the available data I con- sider vir- tually impossible – so that the hysteresis model is possibly the only available source. Overall, the addressed issues call for truly major revisions before this paper can e published in HESS.*

We thank the reviewer for the detailed and insightful review. We are sure they will help to improve the paper. Responses to individual comments are provided below. Reviewer's comments are in italic.

**2 Major Comments**

*C1) Heat storage is calculated as the residuum (P10, l. 26): the authors use the same notation (delta_Q) as above for the 'heat storage of the lake' (p9, l. 5). So, is this meant to yield the heat storage of the lake using the local energy balance? This is not appropriate for two reasons. First of all, the energy balance is*

*based on the turbulent fluxes (which can, in my opinion, be interpreted as reflecting the heat fluxes in the footprint, i.e. over water – of course, if the wind direction is accordingly). Net radiation, however, has a much smaller 'field of view' (a circle with a radius of maybe 2 m a for measurement height of 6 m), so that the albedo is that of the land surface, and the same is true for the longwave outgoing radiation. The considered energy balance, therefore, is reflecting (a combination of) two different surfaces – and the difference cannot be attributed to 'anything' (if not a careful disentangling of the differences between radiation conditions is performed). Second, even if the various sensors would see the same surface, the energy balance (measured at 6 m agl) cannot be expected to be closed. There will be mean advection (possibly even in the vertical – as this location is so close to a step change in surface conditions; water – land), storage (in the air layer between the surface and the sensors!), possibly even vertical flux divergence. Finally, for the local energy balance (still assuming that all the observations correspond to the same surface type), one would also need the ground heat flux. Give the importance of this term (P10, l. 30) the authors need definitely to do something about it*

Thank you for this comment. Yes, heat storage was meant to be calculated from the local energy balance. Addressing the first point of the reviewer: it's right that the sensor mounted at the station is located over land and thus, outgoing longwave and reflected shortwave radiation do not represent the water surface. We considered this problem using following approach:

To calculate reflected shortwave radiation, we used literature values of the albedo for the Dead Sea and additionally performed a short-term experiment, with radiation measurements directly over the water surface to confirm the literature values for our site. Stanhill (1987) calculated the albedo of the Dead Sea surface from ship measurements and reports values of 0.06 in the summer months and 0.09 in the winter months and an annual average of 0.07. He also reported albedo values from Kondrat'Ev (1969) for the latitude of the Dead Sea and the cloud cover observed in the northern part of the Dead Sea, which was 0.08 for November and 0.07 as an average annual albedo value. The results of the short-term experiment concurred well with the literature values and thus the annual average of 0.07 was used in our calculations. For the longwave outgoing radiation we used the Stefan-Boltzmann equation with a water surface emissivity of $\epsilon = 0.98$ (e.g. Konda et al. (1994)) and the surface water temperature calculated with the Monin-Obukhov approach. We also compared it with the results of the short-term experiment, where we found a good agreement. We will add a paragraph to the paper where we will explain this procedure.

The second point of the reviewer discussed the problem of energy balance closure. For every EC system there are uncertainties and a possible non closure of the energy balance through e.g. mean advection. The reviewer is right, that through calculating the heat storage as a residuum, the mentioned amount of heat storage on P10 l.30 also includes the possible non closure of the energy balance. It is definitely important to mention it in this context and we will add some comments on that to the text and to the discussion.

*C2) Statistical model to estimate latent heat flux during offshore conditions: (P14, l. 8) Values up to 200 Wm-2 ...: Interestingly, the estimated values are larger than the measured values (even on average!). Given the statistical model and the high wind speeds especially during the evening hours – in combination with presumed small water vapor pressure deficit - in spring and summer (Fig. 3), this suggests that the statistical model has possibly been used outside the conditions, for which it has been 'constructed'. In other words, the statistical model is trained for cases of high wind speeds (but possibly not even as high as the downslope winds) in combination with (relatively) small water vapor pressure deficit while it is being used for high wind speed and large delta_e. Since these estimated values are not only to fill some gaps in the measured time series but produce the largest values for characteristic times (i.e., after the evening transition), the authors should try to make a very strong case for these estimated latent heat fluxes. In this sense, the statistical model should not only be evaluated in the 'classical sense' (as it is being done – and very convincingly!), but also the question (hypothesis) should be addressed, whether the onshore (training) and offshore (application) conditions are comparable. In other words, how is delta_e over land related to delta_e over water? For this, potentially the two additional EC sites (p3, l. 17 - they are apparently available but not used in this study) could be employed. Similarly, the question should be addressed how the strong downslope winds are related to typical wind speed over water. The question here is therefore whether there is any suitable information (possibly from other studies or sources), which would support the hypothesis that the observed (offshore) wind speed can be used to estimate the wind speed over the [entire] Dead Sea.*

Thank you for this comment. To assure, that the model is not applied outside the conditions for which it has

been 'constructed', the extreme values of offshore wind velocity and vapour pressure deficit, were not considered to calculate evaporation and it was always checked that data were not extrapolated. Extreme values in this case were considered to be the 1st and 99th percentile of the data and with this regulation wind velocity and vapour pressure deficit values, which were used to calculate evaporation were within the model boundaries. Evaporation values, which could not be calculated because wind velocity or vapour pressure deficit were outside the boundaries were treated as missing values and filled with the corresponding value of the median diurnal cycle. We know that this might lead to an underestimation of evaporation, but with this procedure we took care that training and application data were comparable and that the model was not used outside the training conditions.

The wind velocity measured at the station is a point measurement, so it is not valid for the entire Dead Sea water surface. However, the offshore wind velocities measured at the station are in our opinion suitable for the water area, and thus the fetch around the station. This is also confirmed by wind lidar measurements, which were performed during the DESERVE project. The measurements showed that westerly winds in the evening regularly reached several km over the lake without loosing its strength. Furthermore other studies from Weiss et al. (1988) and Hecht and Gertman (2003), evaluated data aquired in the middle of the lake. They both observed westerly winds in their data and listed strong wind events with hourly averaged velocities between 8 to 9 $\mathrm{m\,s^{-1}}$ in the study of Weiss et al. (1988) and 10 to 12 $\mathrm{m\,s^{-1}}$ in the study from Hecht and Gertman (2003).

For the vapour pressure deficit we don't have data from the middle of the lake to directly compare it, but we made following observations: 1) We have seen in the data that the strong westerly winds are connected with high turbulence, and even rotor formation was observed. This means that vertical mixing and air mass exchange is enhanced and thus VDP decrease should be low. 2) The fetch of the station is around 600 m. In our opinion the decrease of VDP within such a distance is not very strong considering the turbulent mixing. 3) Evaporation has a stronger dependence on wind velocity than on VDP which makes the influence of VDP variations on the results weaker. But we agree with the reviewer that this is an uncertainty which has to be mentioned and we will discuss in the discussion chapter.

We also agree that the measured evaporation is only valid for the footprint of the measurement location and that it can vary from other areas of the water surface. Estimates of the entire Dead Sea evaporation are not possible by EC measurements, only by applying models. So for future work, the presented regression model can be used to estimate evaporation for the whole water area, when using model data (wind velocity and vapour pressure deficit) at multiple locations over the Dead Sea water surface.

*C3) The Discussion Section as a whole is, first of all, more a summary than a discussion. Much of what has been stated before is repeated (and the 'discussion' consists to some extent in adding some literature values). The statistical model, for example, is repeated to be good enough (no discussion), rather than addressing potential difficulties (see major comment 2).*

*The 'problem' with the downslope winds (having a stronger wind speed that over the lake, and (probably much) larger water vapor pres- sure deficit is mentioned - but only mentioned to yield a 'slight overestimation'. Based on what is this called 'slight'? Is it 10% (and would 10% be slight)? Or 50%? (but only occurring during 30% of the hours? – and what is then slight?). I think this would be a discussion.*

*Another point that apparently needs discussion is the fact that the radiation measurement does not 'see' the same surface as the authors want to probe with their EC system, i.e. the lake surface. In fact, I think that either all the aspects, which include Rn have either to be removed from the paper, or an estimate has to be made to establish a method to estimate Rn over water from measured Rn over land (see major comment 1). The discussion then, would consist of the associated uncertainty and the potential impact on the interpretation of the empirical methods (i.e. their performance). Given the relatively large uncertainty in the statistical model, a useful contribution to the discussion would be to test the empirical relations for only a subset of days (for the 1 day averaging period, say), for which the impact of the statistical regression model is minimal (only a few or no estimated evaporation hours, mostly measured values). If then the comparison to the 4 empirical methods (Tab 6) would be robust, this would indeed be an indication that the conclusions regarding the appropriateness of the empirical relations are supported by data (not the statistical model). This last discussion, of course, would only make sense if the 'radiation problem' had somehow been overcome. Finally, an important point for the discussion seems to be that the empirical estimates are relatively good 'average estimates' (28 days) – but do only have reduced predictive (diagnostic, that is) skill for short time scales*

Thank you for the detailed remarks about the discussions sections.

We agree that the discussion of the possible uncertainties of the applied method is probably too short. We will revise this section and will address also address the 4 points of the reviewer:

1) The first point of the reviewer refers to C2 and if the regression model was used 'outside' its boundaries. As already discussed this was not the case and so an overestimation does not take place but we will discuss the possible underestimation of evaporation, due to the fact that 'extreme' values of wind velocity and VDP were not used for calculations but instead a median value was used (see also answer to C2). Furthermore, we will discuss the question of overestimating evaporation due to the possibility that VDP decreases with increasing distance to the shoreline. As already mentioned in the answer to C2, in our opinion, the influence on the evaporation results is weak but nevertheless will be discussed in the revised version of the paper.

2) Regarding the net radiation measurements, we will explain our methods for calculating net radiation in section 3 and then discuss the possible uncertainties caused by the applied methods and their impact on the empirical methods in the discussion section.

3) We also want to thank the reviewer for the idea to test the empirical formulas on a subset of days to show that the regression model does not influence the results of the empirical relations. We can realise those tests and add the results to the paper.

4) The last point of the reviewer, refers to the reduced predictive skills of the empirical approaches on short time scales. We will discuss the reduced predictive skill of the empirical approaches, especially on the sub-daily time scales and that only the aerodynamic approach can reliably be applied to short time scales.

**3    Minor comments**

*P5, l.4 I don't think I have ever heard of a Rototronic sensor*
that was a typo. The company is called Rotronic.

*P5, l. 4 the height of measurement is not given for the radiation, precipitation and pressure sensors. Please specify.*
The measurement height for radiation is 2 m. Precipitation and pressure at 1 m height. It will be added to the paper.

*P5, l. 25 I agree with the present authors (in contrast to the other reviewer) that the appropriate conversion should be based on the water temperature and not the air temperature (the differences will be small, though). ¡w'a'¿ is already an energy flux (i.e., the kinematic flux of absolute humidity), which is only converted into energy units by multiplication with L_v. The corresponding energy (enthalpy) is calculated at the location where the process of evaporation takes place and the relevant temperature is the lake surface temperature (whether this is called T_w, the very top of the water or T_s, the very bottom of the air, is the same). I don't think that the 'constancy of the fluxes' in the SL is a valid argument for using the air temperature (at 6 m agl in the present case) since the fluxes are only 'constant' within the SL, but not below (note that the SL has also a lower boundary, i.e. the laminar layer with a thickness of some millimeters over water wherein the turbulent fluxes are zero by definition – and rapidly change to their 'atmospheric surface value' at its top). The 'SL theory' (including the constant flux assumption) produces a temperature profile (e.g., eq. 11.12 in the text book of Arya (1988)), which is based on a 'surface temperature, T_o or T_s, which actually corresponds to the temperature at the height of the (thermal) roughness length. If we assume an (ideal) stratified SL, stable or unstable, and perfect measurements with a given 'surface' latent heat flux, each measurement height would have a (slightly) different L_v (because of the temperature profile) and hence a different latent heat flux – which is inconsistent with the 'constant flux'. In order to obtain the 'surface flux' from a measurement at any height (within the SL, of course), we must therefore relate the conversion into energy units to a common height, i.e. the thermal roughness length. The actual task therefore in determining L_v is to find the temperature at the height of the thermal roughness length. I think the water temperature is a better estimate for this than the temperature at measurement height. This 'surface latent heat flux' then can be used to assess how many mm of water had been evaporated (what is one of the primary goals in the present study). In any case, two comments have to be added: first, it is in fact potential temperature that has to be used (more precisely, virtual potential temperature) – again the differences (for a 6 m level) are negligible. Second, if L_v is estimated using eq. (3), latent heat fluxes will be in [kW m-2] (since L_v is is in [kJ kg-1]) and not comparable to the sensible heat fluxes from eq (2).*
Thank you for your comment. For better comparison of eq. 3 it's changed to $LE = L_v \cdot \overline{w'a'} \cdot 1000$

*P6, l.17 … the mean vertical velocity: this could be mixed up with the mean vertical velocity over the averaging period that is zero in the 'double rotation approach'. In PF, it is the mean vertical velocity over the period that is used to define the plane.*
Thank you for the comment. We will make this clearer in the text and will add '..over the period that is used to define the plane'.

*P6, l.20…calculations at sites..*
will be changed accordingly.

*P6, l. 28 below 0.5 'what'? (units)*
0.5 is 50%. This will be changed.

*P7, l.3 …data, which…*
will be changed accordingly.

*P7, l. 11 is reasonably good*
will be changed accordingly..

*P7, l. 28 is built*
will be changed accordingly.

*P9, l. 5 on longer time scales (or: on a longer time scale)*
will be changed accordingly.

*P8, l. 11ff The presentation of the methods to estimate evaporation is somewhat difficult to comprehend. I try to exemplify this for the Energy balance method. First, one is referred to Tab. 1. The first mentioned aspect of this method is, what is difficult to obtain. I then try to check (find) F_n in Tab 1 (which is apparently difficult to obtain). F_n does not appear in the given equation (but it is 'explained' below in the list of symbols – even if it is not present in any of the given equations). So, I cannot at least judge how this variable appears in the full equation or is related to other variables that do so. The same with the other neglected variables. The 'result' is called V0 (it has an 'X' in Tab 2). Estimating 'somehow' delta_Q produces then 'V2' (why is it V2 for this method, and V1 for the first?). Anyway, it also has an 'X' in Tab 2 (why not a V2?). And some of the other methods also have an 'X' in this table for delta_Q. Overall, I think the overview on the employed approaches should be presented in a much more concise manner. The reader should be able to judge what has actually been done.*
*P8, l. 15 to make the confusion complete, the 'third method' is then – not the third line in Tab 1 but the fourth.....*
Thank you for this remark. We will go over this section and revise this part that it will be easier for the reader to follow and identify the different sensitivity tests which were performed.

*P10, l. 15 on 26% of the days* will be changed accordingly.

*P10, l. 16 on about 57%...* will be changed accordingly.

*P10, l. 18 if the along-valley flow is northerly (with what I concur judging from Fig. 1) the lake breeze would be expected to be perpendicular (easterly on the western shore). Wouldn't this mean that, what was called a 'lake breeze' before (p10, l. 11) is rather a superposition of the along-valley flow and the lake breeze? Can the authors comment on that?*
We don't think that it is a superposition of the described nocturnal along-valley flow and the lake breeze, as the data showed the following: (1) When we analyse data of the station further inland and a little bit south of the shoreline station, wind during daytime has a much stronger easterly direction. (2) when we look at the third station (which is in the north) data shows a south-easterly flow during daytime. (3) Other studies, e.g. Bitan (1974, 1976); Alpert et al. (1990), all show a lake breeze development during the day, but depending on the

location of the station the exact wind direction of the lake breeze changes. Alpert et al. (1990) analyzed data in the south and found a northerly lake breeze, whereas Bitan (1974) in the north found a south-easterly flow (which corresponds to ()).

So in our opinion the north-easterly flow shown at the shoreline is not a superposition with an along valley flow but rather caused by the specific geographic location of the station, meaning the shape of the shoreline and the nearby orography which modulates the lake breeze.

*P10, l. 22 in the beginning*
will be changed accordingly.

*P0, l. 25 on individual days*
will be changed accordingly.

*P12, l. 4 on most of the days ...*
will be changed accordingly.

*P14, l. 2 what does 'uncorrected' mean? No Webb correction, etc.? Or do the authors refer to 'only measured, no estimated (with the multiple regression model) values'? (same in Fig. 4). I think the term 'uncorrected' is not appropriate.*
uncorrected means only measured values. To avoid confusion we will rename it to "only measured"

*P15, l. 1 Maximum values are reached....: see major comment 2.*
as explained in the answer to comment 2, the values of the model should not overestimate evaporation, as it was only used in the reliable boundaries.

*P15, l.11 the uncertainty...: so, how large was it found to be?*
The uncertainty due to the gap filling method was estimated using the corresponding MAD of the used timestep of the respective month. Overall the mean MAD for the different months varied between 0.019 and 0.029 mm 30 min$^{-1}$. For the total annual evaporation amount the uncertainty due to gap filling resulted in 81.2 mm

*P16, l. 10 evaporation rates of 5.1 mm d-1,.... are measured: do these 'extreme days' contain any estimated values (using the statistical model)? How about the other 'large days' throughout the year?*
Only 3 out of the 72 values for these 3 'extreme' days had to be estimated as northerly winds prevailed. The estimated values were also not the maximum values reached on these days. We can also check this for other 'large days'

*P16, l. 14 not shown: I think that this 'case' (if it were shown) could serve as a good example to gain some confidence in the statistical model (see major comment 2). Tab 6 'MD' must be defined.*
Thank you for this comment. Indeed this event serves as a good example to show the performance of the statistical model. As mentioned in the previous comment on these 3 days only 3 out of the 72 evaporation values were estimated. When the model is applied to estimate the evaporation of these days, we get quite a good agreement for day 1 and 3 the model underestimates evaporation by only 4% and 5%. But we also see that the model potentially underestimates the extreme values as on day 2 the value was underestimated by 18%. We will add the information about this case to the paper.

*P17, l. 16 MD is probably mean difference, right? Anyway, the mean differences are not given in the table to demonstrate this (for V0 they are essentially the same ...).*
Thank you for this comment. MD is mean difference. The mean difference for the 30 min interval is only given in the text (P.17, l.14) and not in the table to keep the table clear and better readable. To make the comparison easier we will add the value of the mean difference for the 1d interval to the text.

*P17, l. 20 I suggest to start a new paragraph for the BREB method.*
will be changed accordingly.

*P17, l. 25 the largest*

will be changed accordingly.

*P17, l. 30 22%: judging from Fig. 7 this number is probably valid for a 28 d averaging period* Yes that's right. It will be added to the text to make it clear and we will add another explanation, that the numbers for the other averaging times are comparable.

*p17, l. 31 coefficients*
will be changed accordingly.

*p17, l. 31 improved the results: I do not really agree. Indeed, the correlation coefficients do somewhat increase (but look at Fig. 6 – both versions would probably serve as examples for 'statistics 101' students for data sets, for which a linear model is not appropriate). At the same time, slope and offset are getting worse (this is why we usually use different statistical measures....). In my view the results of V2 (as compared to V0) simply demonstrate that the calculation of heat storage is not appropriate (see major comment 1)* we will revise this description.
*p17, l. 34 11%: same as above (and also in the following) these values seem to apply for the 28 day averages* That's right, please see also answer to comment P17,l.30

*p19, l. 7 up to 100%: I don't think I can see this from Fig. 7d*
Thank you for pointing this out. Fig. 7d has to be compared to the measured daily evaporation amounts in Fig. 5. We will add the necessary reference.

*p20, l. 16 with a heat storage term....: in fact, I only now understand what actually V6 does: it fits the Penman equation to account for the missing storage term, right? So, how is this fitting process being done? If it is fitted – as I assume – using the measured evaporation, it does not come as a surprise that a negligible mean difference results. The results from this exercise have to discussed in this light (major comment 3).*
Thank you for the comment. For the fitting it was assumed that the heat storage term equals the difference between the calculated evaporation amounts using the Penman equation V4 and the measured evaporation amounts and that it can be described as a linear function of the net radiation $\Delta Q = a \cdot Rn$. With this assumption the coefficient 'a' was derived and averaged over the measurement period, which resulted in $\Delta Q = 0.77Rn$.
As already mentioned, we will add a more detailed part to the discussion to go into detail about the uncertainties, shortcomings and possibilities of the different methods.

*p21, l. 8 due to the much higher*
will be changed accordingly

*p22, l. 12 The BREB, Priestly-Taylor and Penman... All these methods do not employ radiation (which has a different field of view – and does not represent the water surface, see major comment 1). I would hypothesize that this is, in the first place, the reason for their bad performance. Only if the Penman method is fitted to the data, it can also produce some reasonable results.*
Thank you for the comment. As already explained for comment 1, we considered the different field of view of the radiation measurements. We still think that the strongest influence is the correct representation of the heat storage term, but of course there are uncertainties connected to the calculation of the net radiation.

**References**

Alpert, P., Abramsky, R., and Neeman, B. U. (1990). The prevailing summer synoptic system in Israel - Subtropical high, not Persian Trough. *Israel J. Earth Sci.*, 39(2/4):93–102.

Bitan, A. (1974). The wind regime in the north-west section of the Dead-Sea. *Arch. Meteorol. Geophys. Bioklim. Ser B*, 22(4):313–335.

Bitan, A. (1976). The influence of the special shape of the Dead-Sea and its environment on the local wind system. *Arch. Meteorol. Geophys. Bioklim. Ser B*, 24(4):283–301.

Hecht, A. and Gertman, I. (2003). Dead Sea meteorological climate. In Nevo, E., Oren, A., and Wasser, S., editors, _Fungal Life in the Dead Sea_, pages 68–114. International Center for Cryptogamic Plants and Fungi, Haifa.

Konda, M., Imasato, N., Nishi, K., and Toda, T. (1994). Measurement of the sea surface emissivity. _J. Oceanogr._, 50(1):17–30.

Kondrat'Ev, K. Y. (1969). _Radiation in the Atmosphere_. International Geophysics Series, Vol. 12, Academic Press, New York.

Stanhill, G. (1987). The radiation climate of the dead sea. _J. Climatol._, 7(3):247–265.

Weiss, M., Cohen, A., and Mahrer, Y. (1988). Upper atmosphere measurements and meteorological measurements on the dead sea. Technical report, Ministry of Energy and Infrastructure (in Hebrew). 19pp.

---

## Author Response (AR1)

**Reply to Reviewer 1**

Jutta Metzger et al.
(jutta.metzger@kit.edu)

**1 General Comments**

*I think this paper deserves publication because the dataset is particularly interesting. As said by the authors, it is the first time that one year of turbulent atmospheric fluxes measured by the eddy-correlation (EC) method has been presented for the specific area of the Dead Sea. The data have been professionally processed. They can efficiently be used to assess several parameterizations generally used for long-term measurements when the EC method is not available. The originality is that the authors provide several levels in the parameterizations, according to the measurements that can be performed. However the paper requires some important corrections before being published. I try to describe them below. Some corrections, related to the methodology, are essential. Other are secondary and consists in numerous details that could be improved.*

Thank you for the very detailed and insightful review. Your comments helped to improve the paper. Responses to individual comments are provided below. Reviewer's comments are in italic.

**2 Specific Comments**

*Methodology : I think the difficulty comes from the fact you make a local measurement over ground (with the EC method) while you would like to include the close environment in the driving parameters of the turbulent fluxes, considering that the air that is advected on the measurement site is (most of the time) characterized by the water surface temperature and water surface vapor partial pressure. I am not against the idea, but I think the way you deal with this assumption is not always correct. I am aware that you want to prove that measuring on the headland, very close to the seashore is equivalent to measuring with a raft in the middle of the sea, but I am not totally convinced. Another issue is the fact that you address different time scales for the energy budget you consider, without saying accurately which timescale you refer to.*

*C1) When measuring the latent heat flux LE at level 6m with the EC, the Lv value (kJ/kg) that has to be used to change the evaporation rate $\overline{w'a'}$ into a flux is that of the air, i.e. 3148.4-2.37 Ta and not Tw as you use in Eq. 3. Ta is the air temperature in K. Even if the evaporation takes place at the water surface as you mention p 6, line 3, LE is assumed to be constant in the surface layer (in fact, not to vary more than 10% of its surface value).*

A) Thank you for this comment. As Lv is a water specific variable and defined by the temperature of the liquid, in our opinion Tw is the appropiate variable to calculate Lv. The literature equation $Lv = 3148.4 - 2.37 \cdot T$ is derived via the Clausius-Clapeyron equation, by measuring water temperature and saturation vapour pressure. This results in the enthalpy of vaporization. Dividing the enthalpy of vaporization by the molar mass of water gives us the latent heat of vaporization. So it does not include air temperature measurements and is only a function of water temperature.
When using Lv to convert the evaporation rate $\overline{w'a'}$ into LE, which is the energy used to evaporate the water at the surface, from a physical point of view, the water surface temperature and not air temperature has to be used as Lv is a water specific variable Lv=fct(Tw).

*This unique Lv value should be used for both offshore and onshore winds. In fact an internal boundary layer develops either inland or offshore, depending on the wind direction. Perhaps you should mention it and clarify*

*the parameters you use in both situations. [The only opportunity to use Lw as you define it, would be to take into account the heat loss of the water due to water evaporation, as shown by Giadrossich et al., 2015 in their eq (2). This eq. applies to the energy budget of the whole sea, and not the local energy budget that you quantify at the EBS].*

A) As in this paper we are only interested in fluxes from the water surface only fluxes measured during onshore wind conditions are used, because then the source area of the flux is over water. All flux measurements for offshore wind conditions are neglected as they represent the fluxes from the land surface. To fill the so created gaps in the time series the regression model is used.

At this point we take advantage of the very periodic wind systems at the Dead Sea. There are only very short time frames with westerly offshore winds where the flux measurements have to be neglected and therefore a large data set with onshore wind conditions is available to establish the regression of the latent heat flux from the water surface with wind velocity and water vapour pressure deficit. With this so gained regression we calculate evaporation from the water surface for the time steps with westerly winds as no measurements of the fluxes from the water surface for this conditions are available. With this procedure we get the full diurnal cycle of the evaporation from the water surface, just as it would be measured with a raft station in the lake. As only flux data for onshore wind conditions are used, $\overline{w'a'}$ is always converted into LE with Eq. 4:

$$Lv = 5150.6561 - 13.9530 \cdot T_w + 0.0162 \cdot T_w^2$$

(This equation is slightly different to Eq. 3, which is for pure water, as the salinity of the water also influences the latent heat of vaporization.)

For the investigation of the energy balance of the land surface no data from this station is used as we have another station further inland, which is not affected by the water surface. The results of the energy balance of the land surface are not discussed here as they are beyond the scope of this paper.

*C2) When discussing the various models you apply to your data, you use Tom-Ta or Ew-Ea. The latter should be replaced by $E_{surf}$ - Ea, with $E_{surf}$ (or another name) standing for the water vapor partial pressure at the surface (water or ground). The former, by $T_{surf}$ - Ta:*

- *For onshore winds, as the source area is over water, we think that the similarity profile you use in Appendix B, p26, to deduce the water surface temperature is not appropriate since the regressions you wish to establish in the following will depend on this profile. We suggest that you'd try to find independent remote-sensed measurements of the Dead Sea surface temperature instead, which is not exactly the air temperature at the surface, but is the closer you can find. If you do so, we am almost sure that the discrepancy between panels 2 and 3 in Fig. 2 will be larger. In that way, the models you will apply in the following will include independent measurements (since the temperature difference will not result from the similarity profile).*

  A) We agree that remotely sensed water surface temperature would be favourable as it is independent from the air temperature measurements. This was also considered and discussed when analysing the data but it was discarded because of the following reasons:

  - Nehorai et al. (2009) used Meteosat Second Generation (MSG) data to estimate water surface temperature from the Dead Sea. For retrieving the water surface temperature from the satellite data the operational SST algorithms could not be applied as they are calibrated to mean sea level and do not take the additional 421 m atmospheric layer in the Dead Sea valley into account. They derived the water surface temperature by calibrating their algorithm against in-situ measurements. Unfortunately, we did not have the necessary in-situ measurements of the water surface temperature to follow their procedure to derive the water surface temperature from satellite data.

  - Furthermore, Nehorai et al. (2009) raised concerns, that on days where the Mediterranean Sea Breeze, a strong westerly wind, enters the valley in the afternoon the enhanced evaporation causes enhanced water vapour and thus a stronger absorption of thermal IR radiation which leads to a screening of the Dead Sea surface and thus incorrect estimates of the water surface temperature. For their studies they

excluded all data with these conditions. This would lead to data gaps in the time series of water surface temperature during westerly (offshore) wind conditions, meaning that no water surface temperature data would be available for the timesteps where it is actually needed to estimate evaporation from the water surface, as the station measures evaporation from the land surface for offshore wind.

– Another point why satellite data was not used is the need of a continuous time series. Satellite data can not be used for cloudy conditions. So especially for the winter months cloud cover would reduce data availability strongly.

– Because of the aforementioned problems with satellite data we followed the advice of another paper from Nehorai et al. (2013), which shows that "SST is highly correlated to air temperature ($R^2 = 0.93 - 0.98$) in all seasons". Based on these results the similarity approach was used to calculate water surface temperature from air temperature.

– Of course there is dependence in the data, but we also checked the results of the similarity approach with a short term experiment of about 5 days, where longwave and shortwave radiation was measured directly over the water surface. From the outgoing longwave radiation radiation temperature of the water surface was calculated and compared to the calculated Tw using the similarity approach. A correlation of 0.8 was achieved. This is quite good, considering the uncertainty of the radiation measurements, as the radiation sensor was getting covered with salt through the spray over the course of the 5 day experiment.

Following paragraph was added to the paper:

T) For the surface water temperature, $T_S$, no in-situ measurements were available. Also remotely sensed surface water temperature products could not be used as operational SST algorithms are calibrated to mean sea level and do not take the additional 421 m atmospheric layer in the Dead Sea valley into account. Nehorai et al. (2009) showed that a calibration of satellite data with in-situ measurements is necessary. Furthermore, Nehorai et al. (2009) raised concerns that enhanced water vapour input into the atmosphere through evaporation causes stronger absorption of thermal IR radiation, which leads to a screening of the Dead Sea surface and thus incorrect estimates of the surface water temperature. Following the results of Nehorai et al. (2013) , which showed that "SST is highly correlated to air temperature ($R^2 = 0.93 - 0.98$) in all seasons" the Monin-Obukhov similarity approach was used to calculate surface water temperature from the measured air temperature (see Appendix A), and referred to as $T_{MO}$ in the following.

*For offshore winds, the source area is partly over water, partly over land. The $\Delta T$ estimation should be a combination of ground surface temperature and water surface temperature. There again, an estimation of the ground surface temperature (perhaps after some assumptions of the emissivity) would be appropriate. You could perhaps also use the upward longwave radiation flux measured at the neighbouring ground station.*

A) For the multiple linear regression approach, data from offshore winds are excluded. So land surface temperature is not needed. As the regression model is used to calculate the evaporation from the water surface for offshore wind conditions, the water surface temperature is needed and can be calculated with the MO Theory.

T) Through the installation of the EBS at the shoreline flux data from the water surface are only available for onshore wind conditions and all data for offshore wind conditions, i.e. wind directions between 230° and 330°,  are rejected for the analysis

- *For offshore winds, I do not know any mean to deduce $E_{surf}$ , unless you make assumptions on the water vapour at the ground surface. You could use the similarity profile, but meanwhile you would make the choice of a model and Sections 3.3 and 4.4 would become useless. For onshore winds, $E_{surf} = 0.65 \cdot Ew(Tw)$ could be an appropriate estimation, Tw being the satellite sea surface temperature, instead of Tmo. A sensitivity study of the regressions to the error in Tw (which determines ew) could also be informative.*

A) This paper focuses only on the energy balance of the water surface. The energy balance of the land surface is not discussed and beyond the scope of this paper. For investigating the energy balance of the

land surface we have a different station which is further inland and therefore not influenced by the water. Therefore there is no need to deduce $E_{surf}$ from the data set. We explain this point in the introduction:

T)That is why, in the framework of the international DESERVE project (Kottmeier et al., 2016), a new concept of assessing lake evaporation from onshore measurements was applied.  Long-term eddy covariance measurements  were conducted at the Dead Sea shore, which provided evaporation data for onshore wind conditions.  These measurements were combined with a statistical model to calculate evaporation for offshore wind conditions.

*C3) If we consider the 3 objectives you propose to fulfill : i) is fulfilled by the EC method, but you do not need to use any multiple regression model to quantify the offshore conditions : you directly measure them. The way you can link these flux measurements with local air or surface parameters is another issue.*

A) Yes the offshore conditions are measured but not the fluxes from the water surface for offshore conditions. For offshore wind conditions the source area of the measured flux is the land surface. As the aim is to get the full diurnal cycle of the flux from the water surface (which would be measured if the station would be located on a raft), it is necessary to apply the multiple regression model and estimate the flux values for offshore wind conditions.

T) (i)  Provide an applicable method for measuring lake evaporation, using a station located at the shoreline.

*ii) I think you cannot totally achieve this aim since you can only access to the local terms of the energy budget and make assumptions on the terms you have to neglect.*

A) We fully agree. We can only get the local evaporation at the measurement site and not the evaporation for the whole lake. This point is rephrased that it becomes clear that these are local values.

T) (ii) Evaluate the  local evaporation rate of the Dead Sea at the measurement location and its diurnal and intra-annual variability,

*iii) OK if you define what 'evaporation' is. You could also add that you want to assess the capacity of these models to retrieve the 'evaporation' term, in the future, when the EC sensors are not available any more (you suggest this in the conclusion).*

A) we added your suggestions to point iii.

T) (iii) evaluate the applicability of the commonly used indirect methods to calculate evaporation rates for different time scales, from sub-daily to biweekly time intervals, from the Dead Sea, and assess the capacity of the methods to retrieve the evaporation term, in the future, when eddy covariance measurements are not available.

**3 Other Remarks**

Thank you very much for the detailed and helpful remarks. First, we provide answers to the questions raised by the reviewer. Afterwards we list the comments about readability and linguistical problems. We considered all of these comments in our revised version of the paper and revised the text accordingly.

**3.1 Questions**

*13 - p2, line 7 : during which period this decrease happened ? And 60-400 denotes a very large variability. Can you explain why ? (variability among the authors ?)*

A) The decrease was caused by the construction of dams and canals along the Yarmouk river and Lake Tiberias/Kinneret mainly between 1955 and 1964. Since then only about 10% of the natural discharge of the Jordan river enters the Dead Sea. The variability of the inflow results from variability among the authors.

T) The main water inflow to the Dead Sea is the Jordan river, but through anthropogenic interferences the

discharge of the Jordan river into the Dead Sea decreased by 90 % down to $60 - 400 \cdot 10^6 \, \mathrm{m^3 \, a^{-1}}$ (Asmar and Ergenzinger, 2002; Holtzman et al., 2005) compared to its natural discharge before 1955.

*17 - p2, line 16 : ←'shifting of the fresh/saline groundwater interface' ('of the' has been added). Please define sinkholes, I did not know this phenomenon.*
A) Sinkholes are holes or depressions in the ground formed by subsurface erosion or removal of soluble bedrock and the collapse of the surface layer.

*18a - p2, line 25 : why westerly winds would be harmful, compared with easterly winds ? You could also delete 'the' at the end of the line, before '1940'. I suppose this harm comes from the fact that easterlies carry drier (continental) air, whereas westerlies carry moister air. But this cannot be guessed from what you wrote here.*
A) The westerly winds have often high wind velocities enhancing the evaporation and thus accelerating the lake level decline.
T) Furthermore, it increases the diurnal penetration of the westerly winds into the valley in the afternoon. These westerly winds have often high wind velocities enhancing the evaporation and thus accelerating the lake level decline.

*29 - p5, line 5 : is it a tipping bucket rain gauge ?*
A) Yes, it is a tipping bucket rain gauge.

*34 - p5, eq (2) and line 24 : usually, $LE = \rho_a Lv\overline{w'r'}$ or $LE = \rho_a Lv\overline{w'q'}$ where r and q are the water vapour mixing ratio and specific humidity, respectively. Brutsaert (which you refer to in the following) uses the latter definition for the evaporation rate : $Ev = \rho_a\overline{w'q'}$. In addition, if I remember well, the hygrometer converts the absolute humidity in water vapour mixing ratio, using T=20°C and P=1013.25 hPa. Perhaps you could consider using r or q instead of a.*
A) We decided to keep eq(2) as in our calculations of the turbulent fluxes with the EC software we use absolute humidity as this is the raw output of the IRGASON.

*39- p6 or before (p5) : did you use a constant calibration coefficient for the IRGASON (when was the calibration done ?) or did you calibrate the hygrometer measurements against the low frequency humidity measurements ?*
A) The IRGASON was calibrated before the experiment but not in between, so constant calibration coefficients were used.

*41- p6, line 29 : 'when the variability of the signal' . Could you define the variability (standard deviation/average ?) ? I think it is 0.6 and not 0.6%*
A) This means that if 10 min averaged normalized signal strength varied from one time step to the next more than 0.006, which is 0.6%, the corresponding 30 min flux value was excluded. This procedure was introduced due to the fact, that we found an increasing variation of the signal strength with the decrease of the total signal strength. Most likely due to contamination of the glass window of the instrument.
T) To assure data quality of the flux measurements, several quality criteria were applied. Latent heat flux data were rejected when the signal strength of the radiation source to measure the water vapour was below 0.550%, when the variability of the signal from one 10 min average to the next one was higher than 0.6 % within a the 30 min time interval, and during precipitation events, as a disturbance of the water vapour measurements was expected for these conditions. Due to these quality criteria 10 % of the latent heat flux data were rejected.

*46 - p7, subsection 3.2 : once you have made the corrections indicated in point 2, I suggest that you keep on calculating the multiple regressions, but not with the aim of parameterizing the offshore conditions. Rather, to show how the classical relationship between the latent heat flux and the wind and/or vapour pressure deficit behaves, under the specific conditions of a semi-arid area that is influenced by sea breeze, slope breeze or both. The multiple regressions should be done for onshore and offshore data separately (provided that 19% of the dataset is enough to apply your method to the offshore data). Note that the regressions you examine*

*are similar to the aerodynamic (or bulk model) from Brutsaert. That is why I am not surprised by the good correlations you obtained in Table 4 for V0. That is also the reason why I do not agree with the idea of using these multiple regressions to calculate offshore winds. I also think that the presentation of the simple regressions should be shorter since they are known to fail to represent the flux but they can serve as a base to compare the multiple regression $H = f(U;\Delta T)$, to the regression $H = f(U \Delta\Theta)$, and the same with LE, U and $\Delta e$. Please also think of using $\Delta\Theta$ instead of $\Delta T$, although the difference will be very small. You may have noticed that I added a multiple regression $H((U;\Delta T)$. The reason is because the BREB method is partly based on this correlation.*

A) We think that the reviewer suggested the additional multiple regression for offshore data based on its wrong impression of our work flow at the beginning (also discussed in C1+C2). As we already clarified in our answer to C1+C2 that we do not use the offshore data for any calculations, there is, in our opinion, no need to establish a regression model for this part of the data.

The second point of the reviewer was to shorten the results of the simple regression results (P.13 l.1-6) and add a comparison of the multiple regression with the simple regressions. We went over the paragraph and added the following conclusion to it: it can be seen from the comparison that wind speed has a much stronger impact on evaporation that the vapour pressure deficit (VDP) in spring, summer and winter, but that in autumn the VDP has an considerable impact as well.

The third point raised in this comment was a regression for the sensible heat flux with different variables. This was already done during the analysis. The multiple regression of H=f(U,$\Delta T$) achieved a correlation of 0.93. These results were not presented in the paper as we wanted to keep the focus on the latent heat flux.

*50 - p8 : I would substitute 'aerodynamic method' for 'aerodynamic or mass transfer method'. In fact it is the bulk method, frequently used to estimate surface fluxes over the sea, where the EC method or dissipative method are not easy to implement. Brutsaert (1982) refers to it as the 'bulk transfer' method and it is based on similarity profiles assumptions and the relationship between fluxes and wind or scalar gradients (through the Dalton or Stanton numbers). According to Brutsaert (p88 in my edition, reprinted in 1984)*

$$Ev = C_e \rho_a v_a (q_{surf} - q_a) = C_e \rho_a v_a (e_{surf} - e_a) \frac{0.622}{p} \tag{1}$$

*Without telling it, you assume equal transfer coefficients for evaporation and momentum ($C_e = C_d$). $C_e$ is a mass transfer coefficient for evaporation and $C_d$ is the drag coefficient $= \frac{u_*^2}{v_a^2}$ . Introducing the logarithmic wind velocity gradient under neutral conditions, which is a second assumption, Ev becomes :*

$$Ev = \frac{k^2}{(ln\frac{Z}{zo})}^2 \rho_a v_a (e_{surf} - e_a) \frac{0.622}{p} \tag{2}$$

*So $K_E$ you identify in Table 1 should not contain $\rho_w$. I suppose you needed to add it since the kinematic flux you calculated is in term of absolute humidity (but you should have divided Ev by $\rho_a$ and not $\rho_w$).*

*To conclude, I suggest you add a remark concerning the assumption that Ce = Cd. Perhaps this could be explained in an additional Appendix (3). I also suggest that you move into the present subsection, your sentence from p17, lines 12-13 : 'the aerodynamic approach is the only approach designed for sub-daily time intervals'. And you could add '(typically 30 min in this study)'. I find that you well address this timescale issue in the following, specifically with Table 6 where you show the results. However, it should be also clearly mentioned in subsection 3.3 (for the 4 models).*

A) Thank you very much for this comment. We followed your suggestions and added an explanation about the assumptions and steps we made, including Ce=Cd, and the use of the logarithmic wind profile. We also addressed the issue that only the aerodynamic approach can be used for sub-daily calculations. The use of $\rho_w$ is necessary to convert evaporation $Ev$ in mm d$^{-1}$.

T) An aerodynamic approach also known as mass transfer approach, the energy budget method, and two combination approaches, namely the  Priestley-Taylor  and Penman equation, will be evaluated on time intervals of 1, 7, 14, and 28 days. The aerodynamic approach is the only approach which is also designed for sub-daily time intervals and will thus also be tested for 30 min time intervals. [...] With the assumption of equal transfer coefficients for evaporation and momentum ($C_e = C_d$) under neutral conditions the logarithmic wind profile can be used (Van Bavel, 1966)(Table 1, V0). This is the default version of the aerodynamic method for the sensitivity studies.

*57 - The hysteresis model from Duan and Bastiaanssen, $\Delta Q = aRn + b + c\frac{dRn}{dt}$ should be described including the discussion about the term $\frac{dRn}{dt}$ . Note also the dependance of c on the range and variability of the water surface temperature. V3 is a specific case of V2 where b=0 and c=0, a being obtained as 'the deviation of the default version from the measurements'. Did you try to determine specific (a,b,c) for your own dataset, just to quantify the deviation relative to Duan and Bastiaanssen's results ? I do not suggest to include them in the models you use, since, doing so, you would invalidate the V5 regressions.*

A) Thank you for this comment. We added some additional explanation about the hysteresis model to the paper. We also explain that in our analysis we first used the proposed ansatz from Duan and Bastiaanssen $\Delta Q = aRn + b + c\frac{dRn}{dt}$, and calculated specific a,b,c from our data set. This was done, as Duan and Bastiaanssen stated in their paper that 'a,b, and c vary largely among lakes. [...] and that lake-specific coefficients should be determined'. Then we investigated the special case b=0 and c=0. V5 is an additional 'special' case were we repeated the calculations using the hysteresis model and the coefficients from Kohler and Parmele, which means V2 and V5 are not the same.

T) Duan and Bastiaanssen (2015) proposed a hysteresis approach to calculate the heat storage term, depending only on the net radiation $(\Delta Q = a + b \cdot R_n + c \cdot dR_n/dt)$. This approach is applied to the measurement data and the rresulting coefficients (a, b, c) are used in sensitivity version V2 to calculate $\Delta Q$.

*58 - Could you please be careful to discuss the assumptions of the other two models and give additional information for Kohler and Parmele's work ? I would not say that Ts has been removed (if I understood correctly) but that it has been estimated from the long-wave radiation flux. This is no-doubt an improvement, relative to your initial estimation from the similarity profile. Please use the same symbol to design the water temperature as the one you have used in the previous section, unless you want to distinguish it on purpose.*

A) Thank you for the comment. We revised the description of the sensitivity studies that it hopefully becomes clearer to the reader. Kohler and Parmele did not estimate the surface temperature from longwave radiation but removed $T_s$ by doing following approximation. He used the first two terms of a binominal expansion

$$L \uparrow = \epsilon \sigma T_s^4 = \epsilon \sigma (T_a^4 + 4T_a^3(T_s - T_a)) \tag{3}$$

and then used $T_s - T_a = (E_s - e_a)/\Delta$ to derive following parameters

$$L' = L \downarrow - L \uparrow = \epsilon_w \cdot L \downarrow - \epsilon_w \cdot \sigma \cdot T_a^4 \tag{4}$$

$$\gamma' = \gamma + \frac{4 \cdot \epsilon_w \cdot \sigma \cdot T_a^3}{K_E \cdot \rho_w \cdot L_v \cdot v_a}. \tag{5}$$

T) In V4 the uncertainty caused by the calculated longwave outgoing radiation with $T_{MO}$ was eliminated by using an approximation from Kohler and Parmele (1967) where they calculated the longwave net radiation and the psychromatric constant using air temperature only.

*62 - Fig. 2 : Please add 'prec' after 'daily precipitation amount' in the caption. Which temperature is Ta : ultrasonic at 6m, BetaTherm probe at 6m, HC2S3 probe at 2m? Please represent $T_{surf}$ instead of Tmo. You could also show $T_{surf} - T_a$. Please represent $e_{surf} - e_a$, and perhaps also qa. I suppose the air is very dry during summer. It would be interesting to show the annual evolution of the air specific humidity. 200 $Wm^{-2}$ are enough for the H vertical axis. It would be convenient to add a thin line for 0 $Wm^{-2}$ in the lower panel. Did you try to represent the daily average parameters (Rn, H, LE, $\Delta Q$) on the same graph (with a more appropriate scale on the y-axis), to be able to see the phase shift between the annual maxima and also whether the Rn variation relative to the time is linked or not with $\Delta Q$ (in relation with the hysteresis model from Duan and Bastiannssen).*

A) Ta is 2m temperature. The use of $T_{MO}$ and not $T_{surf}$ was discussed in C2. As explained, there is no way to determine $T_{surf}$ from satellite data and therefore we will keep $T_{MO}$ and $e_{MO}$ in the graph. Specific humidity was added to the graph, but presenting the daily averages made the graph very overloaded and was thus not included.

*65 - p10, line 6 : 'the annual precipitation normal of 80 mm' : the word normal is not accurate. What is the period considered by Goldreich, 2003 ?*

A) The period which was considered by Goldreich was the climatological standard normal period 1961-1990.

T) The  precipitation amount for the observation period is high compared to the  mean annual precipitation of the standard normal period 1961 to 1990 of 80 mm (Goldreich, 2003).

*68 - p10, lines 14 and 15 : are 32 and 26 % relative to lake breeze or lake breeze+ synoptic conditions ?*

A) These values are relative to the total amount of days in winter.

T) In winter, the synoptic conditions gained more influence and often superimposed the local wind field such that a north-easterly lake breeze was only observed  on about 32 % of the days and a south-easterly flow  on 26 % of the days  in winter 2014/15.

*76 - p10, discussion on the energy budget : you assume that the energy budget is closed and you never discuss the frequent non-closure energy budget problem that is reported in several studies in the literature (see Foken et al. in Aubinet et al. p108-109). You cannot avoid this discussion, even if there is no mean to estimate the error, especially because your measurements are done at the boundary between the marine and the continental surface layers. Under these conditions, the surface change may generate large scale heterogeneities that are unlikely to be correctly taken into account by the local measurements. You show in Fig. 3, extreme values of $\Delta Q$ that are of the order of the net radiative flux in spring and summer. It is unlikely to be true. Anyway, it is known that the shorter the timescale, the larger the non-closure. By contrast, the average $\Delta Q$ (daily average) is about 100 $Wm^{-2}$ during summer and decreases down to a few tenth of $Wm^{-2}$ in winter, which are rational values (I remember that LE has to be recalculated, but it should not be very different).*

A) Thank you for the comment. We agree with the reviewer that for every EC system there are uncertainties and a possible non closure of the energy balance. The reviewer is right, that we should address this point in context of the calculation of the heat storage as a residuum. We addressed this point in section 2 (Measurement site and Instrumentation.

T) Furthermore, the heat storage of the lake was not measured and was therefore calculated as the residuum of the energy balance equation $(Rn = LE + H + \Delta Q)$. Notable hereby is that $\Delta Q$ also contains the possible non-closure of the energy balance. Considering the values of common energy balance closure studies (Foken, 2008; Wilson et al., 2002) the heat storage is thus most likely about 20 % smaller than calculated.

**3.2 Linguistical Comments**

A) Thank you very much for these detailed comments. We considered the comments in our revised version of the paper.

*5 - The term 'evaporation' you frequently use is not accurate enough. I will point it out in the following.*
A) we revised the paper and either use 'evaporation rates' or 'annual evaporation'.

*6 - p1, line 6 : 'total annual amount measured' → 'total annual amount of evaporation measured' (in this case, you do not need to be more accurate since you provide the evaporation unit).*

*7 - p1, line 7 and further on : 'vapour pressure deficit' ← 'water vapour pressure deficit'*
A) rephrased

*8 - p1, line 8 : 'Consequently' is not appropriate. Perhaps 'in fact' could be used instead. What do you mean by 'evaporation amounts' ?*
A) rephrased
T) [...] After sunset, the strong winds cause half hourly evaporation rates which are up to 100 % higher than during daytime...

*9 - p1, line 10 could be changed to ← 'during daytime. During nighttime, evaporation rates are also larger than the daytime evaporation rates, due to strong ...'. Why do you use 'evaporation rate' this time ?*
*Note that this result will perhaps require corrections, in light of what is said in the following (see my final*

*remarks for instance).*
A) rephrased
T) The results show that the diurnal evaporation cycle is governed by three local wind systems: a lake breeze during daytime, strong downslope winds  in the evening and strong northerly along-valley flows during the night. After sunset, the strong winds cause half hourly evaporation rates which are up to 100 % higher than during daytime

*10 - p1, line 11 : The link 'Furthermore' is somewhat awkward. You should explain here why you calculated the regressions. By the way, I think that the multiple regressions should be established for another purpose (see my remark 46-).* A) changed
To account for lake evaporation during offshore wind conditions, a robust and reliable multiple regression model was developed using the identified governing factors

*11 - p1, line 14 is clear and nice. I skip lines 15, p1 to the Introduction, p2.*

*12 - p2, line 6 : you could add 'down' after '90%'.*
A) rephrased

*14 - p2, line 9 : 'The total amount is about' ←'The total amount of loss is about'*
A) rephrased to 'The total loss of water is about'

*15 - You could add a budget equation such as : $10^6$ (400+240-250)+evaporation = -650 $10^6$, which gives evaporation = -1060 $10^6 m^3 a^{-1}$, which is in the range 700 - 1400 $10^6$, indicated by Gavrieli et al., 2006.*
A) a budget equation was not added but the sentence was rephrased.
T) The spread of the evaporation estimates ranges from 1.05 to 2 m a$^{-1}$, comparable to a volume loss of $700 - 1334 \cdot 10^6$ m$^3$ a$^{-1}$(Stanhill, 1994; Salameh and El-Naser, 1999).

*16 - p2, lines 13-14-15 : I would replace ' Evaporation is not only ....environmental problems.' by something like 'It is important to assess the water budget components of the Dead Sea for a climatological purpose, but it is also a priority for the people and the socio-economic development of the region to anticipate the evolution of these components and the consequence for the environment. For instance, the lake level decline causes severe environmental problems . ' (you may of course change the words).*
A) we added following sentence:
T) It is important to assess the water budget components of the Dead Sea for a climatological purpose, but it is also of importance for the people and the socio-economic development of the region to anticipate the evolution of these components and the resulting consequences for the environment. For instance, the lake level decline causes severe environmental problems.

*18b - p 2, line 29 (no link with 18a) : I do not understand 'especially' in this context ('In addition' ?). Perhaps you could also mention whether it is a fresh water fish. Ein Feshkha reserve ?*
A) it should just be an example so I rephrased it to:
T) endangering the unique flora and fauna in the Dead Sea region , such as the unique fish population of the Ein Feshkha reserve.

*18c - p3, line 5 : I would add 'in the evaporation estimations' just before '(Stanhill, 1994'.*
A) added
T) To minimise the spread of 1.05 to 2 m a$^{-1}$ in the evaporation estimates (Stanhill, 1994; Salameh and El-Naser, 1999) and reduce uncertainties, direct measurements of the Dead Sea evaporation are required.

*18d - p3, lines 6-7 : I would remove : 'Furthermore, the governing factors of the Dead Sea evaporation, e.g. wind velocity, vapour pressure deficit, or net radiation, have to be identified, to validate the indirect methods'. These parameters are governing factors every where in the world and you do not have to prove it. I think this sentence is confusing at this point.*

A) The sentence was removed.
T)

*18e - p3, line 8 : 'with a high temporal resolution' instead of ',in high temporal resolution'.*
A) rephrased
The eddy covariance technique is the only method to obtain direct evaporation measurements

*19 - p3, line 10 : 'continues' ← 'continuous'*
A) rephrased

*20 - p3, line 12 : according to Wikipedia, it seems that Lake Kinneret is the same as Lake Tiberias mentioned by Kottmeier et al. (2016) (it is only a remark, you choose the name you prefer). You could perhaps add that it is crossed by the Jordan river which partly feeds the Dead Sea. It is not a major piece of information but I find it nice to provide the reader an idea of the geographical environment.*
A) You are right. Lake Kinneret and Lake Tiberias refer to the same name and can be used synonymously. But as you suggested I added some information about the Jordan river:
T) 'Assouline and Mahrer (1993) measured evaporation from Lake Kinneret, a freshwater lake north of the Dead Sea, crossed by the Jordan river, and Tanny et al. (2008) measured evaporation from a small reservoir also north of the Dead Sea using eddy covariance systems. '

*21 - p3, line 13 : 'to the authors knowledge' instead of 'as to the authors knowledge'.*
A) rephrased
However,  to the authors knowledge, ...

*22 - p3, line 14 : 'Therefore, long-term eddy covariance measurements are conducted' ← 'That is why, in the frame of the international DESERVE project (Kottmeier et al., 2016), long-term eddy covariance measurements were conducted'*
A) rephrased
T) That is why, in the framework of the international DESERVE project (Kottmeier et al., 2016), a new concept of assessing lake evaporation from onshore measurements was applied.  Long-term eddy covariance measurements  were conducted at the Dead Sea shore, which provided evaporation data for onshore wind conditions.

*23 - p3, line 17 : 'provided' and 'was' instead of 'provides' and 'is'*
A) rephrased
T) see comment 22

*24 - p3, lines 17-19 : I perhaps misunderstood but it seems to me you did not use the data from these stations. Is it useful to quote them ?*
A) the sentence was deleted:
T)

*25 - p3, lines 19-21 : 'Provide', 'Evaluate' and 'Evaluate' instead of the same with 'ing'.*
A) rephrased
T) (i)  Provide [...] (ii)  Evaluate [...] (iii)  evaluate [...].

*26 - p3, Measurement site : it is difficult to distinguish in Fig. 1, how far the Judean mountains highest submits are from the lake and what their height is (hills or mountains ?). The Moab mountains are clearly*

*shown.*
A) Fig 1 was changed

*27 - Fig. 1 : you forgot to write 'Jordan river' in panel (a). Landsat with an L in the caption. The red arrows are a little confusing in panel (b) : simple lines instead of arrows would be enough.*
A) Fig 1 was changed

*28 - p5, line 4 : Rotronic*
A) corrected
T) temperature and humidity at 2 m height (HC2S3, Rotronic),...

*30 - p5, line 7 : please add 'open path' before 'integrated gas analyser'. I suppose that at 6m, it did not suffer from spray, even under strong onshore wind conditions ...*
A) 'open path' was added to the text. The instrument did not suffer from spray.
T) With a temporal resolution of 20 Hz, water vapour, $CO_2$ concentration, sonic temperature and the three wind components were measured with an open path integrated gas analyzer and sonic anemometer (IRGASON) from Campbell Scientific at 6 m height.

*31 - p5, line 9 : 'From the 20 Hz data evaporation was calculated using the eddy covariance technique'→ 'The latent heat flux was calculated from the 20 Hz data using the eddy covariance method.'*
A) rephrased
T) The latent and sensible heat flux were calculated from the 20 Hz data  using the eddy covariance method.

*32a - p5, line 9-10 : ←'The principle of the method, the post-processing and data quality control steps are presented in Sec. 3.1'.*
A) rephrased
T) The principle of the  method,  post-processing  and data quality control steps are presented in  Sec. 3.2.

*32b - p5, lines 11-12 : please consider the multiple regressions again (after 46-)*
A) see comments to C1

*33 - p5, eq (1) and line 23 : the ultrasonic anemometer provides the virtual air temperature and not the air temperature Ta. The Schotanus correction is made, as you say in Sec. 3.1.1, to take this point into account.*
A) was corrected in eq (1)
T) $H = c_p \cdot \rho_a \cdot \overline{w'T'_{sonic}},$

*35 - p5, eq (3) and line 27 : as said before, Tw should be Ta.*
A) Tw is correct. See answer to C1.

*36 - p5-6, lines 27 to 5 : the text should be deleted from 'For salt water ..' to the end of the subsection.*
A) See answer to C1.

*37 - p6, line 8 : 'measurement limitations' you could add 'of the sensors'*
A) changed
T) In particular, measurement limitations of the sensors, non-stationary conditions over the averaging period, as well as horizontal heterogeneity have to be considered (Foken et al., 2012).

*38- p6, lines 18 to 24 : it seems to me that the order should be : spectral corrections, Schotanus correction and Webb correction unless you applied an iterative process.*
A) thank you for noting the wrong order. The corrections were applied as you said. The order in the text was changed
T) Spectral corrections were performed to account for the loss of energy for high frequencies, due to path-length

averaging and limited sensor frequency response, following the approach after Mauder and Foken (2011). The influence of humidity on sonic temperature plays an important role for the calculation of the sensible heat flux. To account for this influence, the Schotanus correction (Schotanus et al., 1983) was applied. This correction is particularly important for flux calculations  at sites with high humidity fluctuations, such as over the water surface. The water vapour measurements are influenced by temperature and humidity changes, as only the molar density of water vapour is measured and not the mass mixing ratio. To consider the density fluctuations, corrections after Webb et al. (1980) were applied.

*40- p6, line 28 : 0.5 g/kg ?*
A) 0.5 in the sense of normalised signal strength. For better understanding it was changed to 50%
T) Latent heat flux data were rejected when the signal strength of the radiation source to measure the water vapour was below 50%, when the variability of the signal from one time step to the next one was higher than 0.6 % within  the 30 min time interval, and during precipitation events, as a disturbance of the water vapour measurements was expected for these conditions.

*42 - p6 line 31 to p7 line 2 : you say too much or not enough. It would be nice to describe the tests and to say what ITC is.*
A) As these tests are widely known, and commonly used in eddy covariance processing and also implemented in the TK3 software we decided against describing these tests in detail. The reader is referred to the literature.
T) Further quality control was performed using the steady state test after Foken and Wichura (1996), which analyses each 30 min time interval on stationarity  and the integral turbulence characteristics (ITC) test after Foken et al. (2012), which checks data on fully developed turbulent conditions.

*43 - p7 and further on : I would replace 'fetch' by 'source area'.*
A) 'fetch' is a commonly used word to describe the distance from the tower when describing the flux footprint and will therefore not be changed.

*44 - p7, lines 6-7 : please consider again the rejection of these data : I agree that it is important to distinguish them from the onshore measurements data, but the fluxes are what they are and do not have to be rejected. Nevertheless, it is important to quantify this contribution since the source area may be different.*
A) as the aim of the paper is to analyse fluxes from the water surface only and the regression model is used to estimate evaporation for offshore wind conditions, this data has to be rejected for the calculations.

*45 - p7, lines 7 to 9 : I would replace the 2 sentences 'For southerly wind directions … 600m away from the headland.' by 'For southerly and northerly wind directions the source area is over water and the average source area contributing to 80% of the flux ranges from 0 to 300 m and 0 to 600 m, respectively, ahead of EBS.'*
A) rephrased
T) For southerly and northerly wind directions, the fetch is over water and the average fetch contributing to 80 % of the flux  ranges from 0 to 300 m  and 0 to 600 m, respectively.

*47 - p8, line 1 : 'Indirect methods to estimate evaporation' ← 'Description of four indirect methods to estimate evaporation'*
A) Thank you for the comment, but we keep the shorter heading

*48a - p8 : I would move the first two (essential) sentences of subsection 4.4 ('For the calculation of evaporation -please insert a comma here-, several equations, based on …. at least 7 days') to the beginning of section 3.3.*
A) Thank you for this remark we moved the sentence to Sec.3.3
T) For the calculation of evaporation, several equations, based on different physical approaches, exist. Each approach connects evaporation to different meteorological parameters and is designed for different time intervals,

ranging from sub-daily calculations to a time interval of at least 7 days. Four commonly used indirect methods to estimate evaporation ...

*48b - p8, line 5 'an overview of which sensitivity study is performed' (I added 'of').*
A) sentence was rephrased to:
T) An overview of the sensitivity studies and to which of the methods it is applied to, is given in Table 2.

*49 - Ev is not defined. I suppose it is the evaporation rate defined as Ev=$\rho_a$ w0q0 (Brutsaert, 1982).*
A) definition is added:
T) Selection of commonly used equations to calculate evaporation $(Ev)$

*51 - The sensitivity study, referred to as V1, is performed to address the stability issue. The presentation of the stability factors you refer to (Cline 1977) p. 17 should also be moved to the present subsection. I would also describe the new KE (including the stability factor) in Appendix 3. You could also add that the stability functions for wind and heat are expressed in terms of the bulk Richardson number, which allows estimating the stability when the turbulent fluxes are not known.*

*52 - Energy budget : Here again, $\rho_w$ should vanish if you use the specific humidity instead of the absolute.*
A) see answer to comment 50

*53 - I suggest that you write down the budget equation as Giadrossich, 2015 did (their eq 1) : Rn + Anet = LE + H +$\Delta Q$ , where Anet is the net heat advected into the lake (by stream flow and precipitation minus the heat loss due to evaporation minus the heat transferred at the bottom of the lake) and $\Delta Q$ is the heat storage per unit area in the lake (for most cases) or in the ground (for specific cases with strong offshore winds. Under these conditions, Anet can be ignored). This energy balance applies to timescales larger than the day due to the advection term that cannot be known at a short timescale.*
*54 - With V0, you neglect $\Delta Q$ and Anet . [Note that the resulting reduced budget equation can also be applied at a sub-daily scale.] Neglecting $\Delta Q$ is a coarse assumption that is valid only under specific and occasional conditions. I think you should mention it at first and say that V0, even unrealistic, is a basis for the BREB method that will be improved by V1 and V2. I mean that V1 and V2 should not only be considered as sensitivity studies for V0, but as 2 alternative methods for V0.*
A) we added some explanation:
T) . Because of the aforementioned reasons and the difficulty to obtain these three terms, the net advected heat, the ground heat flux and the heat storage term are neglected in many studies. Thus, for the default version (V0) of the BREB method  these three terms are neglected (Table 1). Even though neglecting the heat storage on the time scales investigated is a coarse assumption it serves as a basis for the sensitivity studies V1 and V2.

*55 - p9, line 8 : you forgot to mention the water vapour deficit.*
A) thank you for noting this. It was added to the text:
T) Using V0, only net radiation, surface water temperature, air temperature, and the vapour pressure deficit have to be known, which are relatively easy to obtain and thus an easy approach to calculate evaporation.

*56 - $\beta$, the Bowen ratio should be defined as H/LE. When fluxes are unknown, $\beta$ can be approximated by the expression you give, provided that $K_\Theta$, the Stanton number for temperature = Ce, the equivalent for evaporation. Also, be careful not to use the same symbol for the Bowen ratio and the activity of water in Appendix A (I would keep $\beta$ for the Bowen ratio).*
A) Thank you for noting the duplicate of $\beta$. We changed it to Bo.  *59 - Table 1 : caption : default versions (V0) in Sec. 3.3 and 4.4. (just add 3.3) Fn and $\Delta t$ are not used and $\rho_w$ should not be used. Please take into account my remarks in 2- Priestley-Taylor is presented as the 3rd method in Sec 3.3.*
A) the order of Priestley-Taylor and Penman was changed. also the caption was changed to:
T)Selection of commonly used equations to calculate evaporation $(Ev)$ in mm d$^{-1}$.The original version and the default version (V0) used in Sec. 3.4 and 4.4 are presented.

*60 - Meteorological conditions : this subsection will have to be read again after new LE, $T_{surf}$, $\Delta e$ values ...*

*61 - For some parameters, the daily values and their evolution are also interesting to discuss in addition to the extreme values.*

*63 - p10, line 4 : 'long term annual mean' : during which period ?*
A) rephrased to
T) In the Dead Sea valley the measured average annual air temperature was 26.5°C for the measurement period, which was slightly higher than the long term annual mean of 25.9 °C found by Hecht and Gertman (2003) for the period 1992 to 2002.

*64 - p10, line 5 : please add 'sometimes' before 'exceeded'*
A) was changed to regularly.

*66 - p10, line 7 : 'made' instead of 'makes'*
A) was changed

*67 - p10, line 9 : 'only during the winter seasons a different behaviour was found' ←except during winter, when the wind increased in connection with the convective activity, a different behaviour was found'*
A) rephrased to
T) However, during winter, only during the winter seasons a different behaviour was found when the wind increased in connection with the stronger large scale activity (Fig. 2).

*68 - p10, lines 14 and 15 : are 32 and 26 % relative to lake breeze or lake breeze+ synoptic conditions ?*
A) its relative to the amount of days in winter. Text changed to: T) In winter, the synoptic conditions gained more influence and often superimposed the local wind field such that a north-easterly lake breeze was only observed on about 32 % of the days and a south-easterly flow on 26 % of the days in winter 2014/15.

*69 - p10, line 15 : Please indicate the direction of the downslope breeze (north-westerly)*
A) changed accordingly: T) In the evening, north-westerly downslope winds, often enhanced by the Mediterranean Sea Breeze (MSB)

*70 - p10, line 16 : 'yielded' instead of 'lead to'*
A) thank you for the comment. we think in this case 'lead to' is also suitable'

*71 - p10, lines 18, 19, 24 : 'exceeding' instead of 'of over'*
A) was changed.
T) The downslope winds regularly reached mean wind velocities of over exceeding 10 m s$^{-1}$ (Fig. 3 b). During the night, a northerly along-valley flow prevailed mainly in spring and summer. The along-valley flow also reached wind velocities of over exceeding 10 m s$^{-1}$ (Fig. 3 c). The net radiation reaches maximum values of over exceeding 900 W m$^{-2}$ in summer and about 500 W m$^{-2}$ in winter.

*72 - p10, line 22 : 'November a Red Sea Trough with a central axis advected dry and warm air'→ 'November, when a Red Sea Trough advected dry and warm air'*
A) changed to:
T) at in the beginning of November, when a Red Sea Trough with a central axis advected dry and warm air into the valley over the course of several days.

*73 - p10, line 25 : 'However, at' ←'However, on'*
A) changed to:
T) However, at on individual days in some winter latent heat flux values even exceed the summer values.

*74 - p10, line 26 : 'in winter latent heat flux values' ←'in winter some latent heat flux values'*
A) see comment 73

*75 - p10, line 27 : the energy balance equation should have been shown in subsection 3.3. You only need to say that at this timescale (24h), Anet is ignored. Do you think it is still correct after rainfall ?*
A) The equation is now shown in Table 1

*77 - p10, line 31 : please reword 'used for heating the lake, which is stronger in spring than in winter'. This is grammatically false.*
A) The sentence was deleted

*78 - p10, line 33 : I do not see that $\Delta Q$ is negative in winter.*
A) There is negative heat storage in winter. However, the net annual amount is positive. Several other studies like (Stanhill, 1990; Anati et al., 1987) already documented this and the thereby caused steady increase of the lake temperature over the last couple of years is also documented (e.g Hecht and Gertman (2003)).

*79 - p 12, fig 3 : it is usually required to indicate the delay between the UTC and local time. You can keep LT and indicate, on the first time (UTC +3h). It would be also interesting to add in the caption the approximate time of sunrise and sunset, especially from Spring to Autumn.*
A) thank you for pointing that out. UTC deviation was added to the text:
T) Through the predominant local wind systems, these wind directions occur almost exclusively in the evening between 17:30 to 20:30 LT (LT=UTC+2) from spring until autumn (Fig. 3)

*80 - p12, line 4 : ←'and, thus, [at] most of the [days] data within this time frame' (remove the [words]).*
A) changed accordingly:
T) at most of the days data within this time frame are excluded.

*81 - p12, line 6 : ←'also for the study of the intra-annual .... this gap' (instead of these gaps).*
A) rephrased

*82 - p12, line 6 : 'A multiple regression model was applied ... for offshore conditions' : As said before, I do not agree with this method : I'm waiting for your decision, regarding the suggestions I made in 2-*
A) see answer to C1

(jutta.metzger@kit.edu)

**1 General Comment**

*This a very interesting paper, addressing an important environmental issue (evapora- tion from the Dead Sea), presenting very important data and (apparently for the first time) directly measured evaporation rates and thus adding important new information to our knowledge. There is no doubt, therefore, that the paper should eventually be published in one or the other form. There are three main pillars: first, to measure, directly, the lake's evaporation using an EC station on the shore (therewith employing the footprint of the water surface). This provides direct measurements for about 70% of the time. I think this approach is well motivated, well explained and also 'well executed' (all the necessary data treatment, corrections, QC, etc.). To estimate, as a second pillar, evaporation during the remaining 30% of the time, a statistical model is trained using the onshore wind conditions and the available in- formation during these conditions, to esti- mate lake evaporation during offshore wind direction. This is very appropriate (and possibly novel) as an approach, and so is the statistical approach, its presentation (and results). However, it has the drawback that the estimated evaporation rates are among the largest during the whole year and thus contribute a substantial fraction of the total (yearly, monthly, daily – the shorter the more variable) evaporation. Therefore, the statistical model does not only have to be tested using the 'usual' tests (cross-validation, etc.) based on the available onshore conditions (what is convincingly being done), but attempts should be made to support the hypothesis that the same statistical model applies (yields the claimed 'good' statistics) when the input data stem from offshore situations (high wind speed in combination with low water vapor pressure deficit). Some suggestions are provided in major comments 2 and 3. As a third pillar, empirical estimates (for evaporation) are compared in their performance to the measured evaporation. This is adding a great deal of value to the paper. Unfortunately, for three out of four methods the comparison is not valid (because net radiation is not measured over water – which basically invalidates all the estimates; see major comment 1). Furthermore, the estimation of heat storage (in the water) is also flawed (see again major comment 1), so that all the different 'versions' tested are not really conclusive (with the exception of V1). Indeed, the results show that all the empirical methods using net radiation give results quite different from the observations (if only measured – and not estimated – values would be considered, this finding would probably be even more pronounced). The only 'reliable' empirical method is the aerodynamic approach (not using Rn). The problem with the different fields of view (for radiation and turbulent fluxes) could somehow be overcome (for example by using satellite – or other – observations [even literature values would be better than nothing] to correct for albedo differences between land and water, and by using the [simultaneously measured] land surface temperature [from the two additional EC sites over land] and the estimated lake surface temperature to correct for different longwave outgoing radiation). To actually estimate heat storage in the water from the available data I con- sider vir- tually impossible – so that the hysteresis model is possibly the only available source. Overall, the addressed issues call for truly major revisions before this paper can e published in HESS.*

We thank the reviewer for the detailed and insightful review. We are sure they will help to improve the paper. Responses to individual comments are provided below. Reviewer's comments are in italic.

**2 Major Comments**

*C1) Heat storage is calculated as the residuum (P10, l. 26): the authors use the same notation (delta_Q) as above for the 'heat storage of the lake' (p9, l. 5). So, is this meant to yield the heat storage of the lake*

*using the local energy balance? This is not appropriate for two reasons. First of all, the energy balance is based on the turbulent fluxes (which can, in my opinion, be interpreted as reflecting the heat fluxes in the footprint, i.e. over water – of course, if the wind direction is accordingly). Net radiation, however, has a much smaller 'field of view' (a circle with a radius of maybe 2 m a for measurement height of 6 m), so that the albedo is that of the land surface, and the same is true for the longwave outgoing radiation. The considered energy balance, therefore, is reflecting (a combination of) two different surfaces – and the difference cannot be attributed to 'anything' (if not a careful disentangling of the differences between radiation conditions is performed). Second, even if the various sensors would see the same surface, the energy balance (measured at 6 m agl) cannot be expected to be closed. There will be mean advection (possibly even in the vertical – as this location is so close to a step change in surface conditions; water – land), storage (in the air layer between the surface and the sensors!), possibly even vertical flux divergence. Finally, for the local energy balance (still assuming that all the observations correspond to the same surface type), one would also need the ground heat flux. Give the importance of this term (P10, l. 30) the authors need definitely to do something about it*

A) Thank you for this comment. Yes, heat storage was meant to be calculated from the local energy balance. Addressing the first point of the reviewer: it's right that the sensor mounted at the station is located over land and thus, outgoing longwave and reflected shortwave radiation do not represent the water surface. We considered this problem using following approach:

To calculate reflected shortwave radiation, we used literature values of the albedo for the Dead Sea and additionally performed a short-term experiment, with radiation measurements directly over the water surface to confirm the literature values for our site. Stanhill (1987) calculated the albedo of the Dead Sea surface from ship measurements and reports values of 0.06 in the summer months and 0.09 in the winter months and an annual average of 0.07. He also reported albedo values from Kondrat'Ev (1969) for the latitude of the Dead Sea and the cloud cover observed in the northern part of the Dead Sea, which was 0.08 for November and 0.07 as an average annual albedo value. The results of the short-term experiment concurred well with the literature values and thus the annual average of 0.07 was used in our calculations. For the longwave outgoing radiation we used the Stefan-Boltzmann equation with a water surface emissivity of $\epsilon = 0.98$ (e.g. Konda et al. (1994)) and the surface water temperature calculated with the Monin-Obukhov approach. We also compared it with the results of the short-term experiment, where we found a good agreement. We will add a paragraph to the paper where we will explain this procedure.

The second point of the reviewer discussed the problem of energy balance closure. For every EC system there are uncertainties and a possible non closure of the energy balance through e.g. mean advection. The reviewer is right, that through calculating the heat storage as a residuum, the mentioned amount of heat storage on P10 l.30 also includes the possible non closure of the energy balance. It is definitely important to mention it in this context.

T) We added a subsection on radiation calculations to section 3: The measurements of the radiation components of the lower half space are not conducted directly over the water surface, but over the land surface. Therefore, these two components have to be calculated. The reflected shortwave radiation was calculated using literature values of the Dead Sea albedo. Stanhill (1987) calculated the albedo of the Dead Sea surface from ship measurements and reported values of 0.06 in the summer months, 0.09 in the winter months, and an annual average of 0.07. He also reported albedo values from Kondrat'Ev (1969) for the latitude of the Dead Sea and the cloud cover observed in the northern part of the Dead Sea, which was 0.08 for November and 0.07 as an average annual albedo value. To confirm the validity of the literature values for our site, a short-term experiment was conducted in November 2014. Albedo values of 0.08 to 0.09 concurred well with the literature values for winter. As the literature values for summer could not be compared to measurements, the annual average of 0.07 was used for all calculations. The longwave outgoing radiation was calculated using the Stephan-Boltzmann equation

$$Rl \uparrow = \epsilon \cdot k_B \cdot T_S^4, \tag{1}$$

with the water surface emissivity $\epsilon = 0.98$ (e.g. Konda et al. (1994)) and the Stephan-Boltzman constant, $k_B$. For the surface water temperature, $T_S$, no in-situ measurements were available. Also remotely sensed surface water temperature products could not be used as operational SST algorithms are calibrated to mean sea level and do not take the additional 421 m atmospheric layer in the Dead Sea valley into account. Nehorai et al. (2009) showed that a calibration of satellite data with in-situ measurements is necessary. Furthermore, Nehorai et al. (2009) raised concerns that enhanced water vapour input into the atmosphere through evaporation causes stronger absorption of thermal IR radiation, which leads to a screening of the Dead Sea surface and thus incorrect estimates of the surface water temperature. Following the results of Nehorai et al. (2013), which showed that "SST is highly correlated to

air temperature ($R^2 = 0.93 - 0.98$) in all seasons" the Monin-Obukhov similarity approach was used to calculate surface water temperature from the measured air temperature (see Appendix A), and is further on referred to as $T_{MO}$.

We also explained the influence of the non closure of the energy balance on the heat storage in Sec.2:

Furthermore, the heat storage of the lake was not measured and was therefore calculated as the residuum of the energy balance equation ($R_n = LE + H + \Delta Q$) using half hourly measurements. Notable hereby is that $\Delta Q$ also contains the possible non-closure of the energy balance. Considering the values of common energy balance closure studies (Foken, 2008; Wilson et al., 2002) the heat storage is thus most likely about 20 % smaller than calculated.

*C2) Statistical model to estimate latent heat flux during offshore conditions: (P14, l. 8) Values up to 200 Wm-2 ...: Interestingly, the estimated values are larger than the measured values (even on average!). Given the statistical model and the high wind speeds especially during the evening hours – in combination with pre-sumed small water vapor pressure deficit - in spring and summer (Fig. 3), this suggests that the statistical model has possibly been used outside the conditions, for which it has been 'constructed'. In other words, the statistical model is trained for cases of high wind speeds (but possibly not even as high as the downslope winds) in combination with (relatively) small water vapor pressure deficit while it is being used for high wind speed and large delta_e. Since these estimated values are not only to fill some gaps in the measured time series but produce the largest values for characteristic times (i.e., after the evening transition), the authors should try to make a very strong case for these estimated latent heat fluxes. In this sense, the statistical model should not only be evaluated in the 'classical sense' (as it is being done – and very convincingly!), but also the question (hypothesis) should be addressed, whether the onshore (training) and offshore (application) conditions are comparable. In other words, how is delta_e over land related to delta_e over water? For this, potentially the two additional EC sites (p3, l. 17 - they are apparently avail able but not used in this study) could be employed. Similarly, the question should be addressed how the strong downslope winds are related to typical wind speed over water. The question here is therefore whether there is any suitable information (possibly from other studies or sources), which would support the hypothesis that the observed (offshore) wind speed can be used to estimate the wind speed over the [entire] Dead Sea.*

A) Thank you for this comment. To assure, that the model is not applied outside the conditions for which it has been 'constructed', the extreme values of offshore wind velocity and vapour pressure deficit, were not considered to calculate evaporation and it was always checked that data were not extrapolated. Extreme values in this case were considered to be the 1st and 99th percentile of the data and with this regulation wind velocity and vapour pressure deficit values, which were used to calculate evaporation were within the model boundaries. Evaporation values, which could not be calculated because wind velocity or vapour pressure deficit were outside the boundaries were treated as missing values and filled with the corresponding value of the median diurnal cycle. We know that this might lead to an underestimation of evaporation, but with this procedure we took care that training and application data were comparable and that the model was not used outside the training conditions.

The wind velocity measured at the station is a point measurement, so it is not valid for the entire Dead Sea water surface. However, the offshore wind velocities measured at the station are in our opinion suitable for the water area, and thus the fetch around the station. This is also confirmed by wind lidar measurements, which were performed during the DESERVE project. The measurements showed that westerly winds in the evening regularly reached several km over the lake without loosing its strength. Furthermore other studies from Weiss et al. (1988) and Hecht and Gertman (2003), evaluated data aquired in the middle of the lake. They both observed westerly winds in their data and listed strong wind events with hourly averaged velocities between 8 to 9 m s$^{-1}$ in the study of Weiss et al. (1988) and 10 to 12 m s$^{-1}$ in the study from Hecht and Gertman (2003).

For the vapour pressure deficit we don't have data from the middle of the lake to directly compare it, but we made following observations: 1) We have seen in the data that the strong westerly winds are connected with high turbulence, and even rotor formation was observed. This means that vertical mixing and air mass exchange is enhanced and thus VDP decrease should be low. 2) The fetch of the station is around 600 m. In our opinion the decrease of VDP within such a distance is not very strong considering the turbulent mixing. 3) Evaporation has a stronger dependence on wind velocity than on VDP which makes the influence of VDP variations on the results weaker. But we agree with the reviewer that this is an uncertainty which has to be mentioned and is now

discussed in the discussion chapter. We also agree that the measured evaporation is only valid for the footprint of the measurement location and that it can vary from other areas of the water surface. Estimates of the entire Dead Sea evaporation are not possible by EC measurements, only by applying models. So for future work, the presented regression model can be used to estimate evaporation for the whole water area, when using model data (wind velocity and vapour pressure deficit) at multiple locations over the Dead Sea water surface.

We added following paragraph to Section 4.2:

T) In summary, the regression model $X_{LE}$ provides a suitable and robust method to calculate the latent heat flux for offshore wind conditions. To assure, that the model is not applied outside the conditions for which it has been constructed, the extreme values of offshore wind velocity and vapour pressure deficit were not considered to calculate evaporation and it was checked that data were always within the model boundaries. Evaporation values, which could not be calculated because wind velocity or vapour pressure deficit were outside the boundaries were treated as missing values.

Furthermore we discussed the so introduced uncertainties in the discussion section:

However, there is still some uncertainty to this method which cannot be accounted for directly. On the one hand, extreme values of wind velocity and water vapour pressure deficit were not used to calculate evaporation when they were outside the model boundaries. This leads most likely to an underestimation of the actual evaporation amount. On the other hand, wind velocity and vapour pressure deficit could decrease with increasing distance from the shoreline, which would lead to an overestimation of evaporation. The comparison with results from measurements in the middle of the lake (Weiss et al., 1988; Hecht and Gertman, 2003) shows that even in the middle of the lake westerly winds with hourly averaged velocities between 8 and $12\,\mathrm{m\,s^{-1}}$ were observed. Also wind lidar measurements confirmed, that the westerly winds regularly reached several km over the lake without loosing its strength (Metzger, 2017). So offshore wind measurement seem representative for the calculation of evaporation. A decrease of vapour pressure deficit has to be considered, but is most likely small due to the following reasons. Firstly, the fetch of the station is quite limited with 600 m, and secondly the westerly winds are connected with high turbulence and thus strong vertical mixing (Metzger, 2017).

*C3) The Discussion Section as a whole is, first of all, more a summary than a discussion. Much of what has been stated before is repeated (and the 'discussion' consists to some extent in adding some literature values). The statistical model, for example, is repeated to be good enough (no discussion), rather than addressing potential difficulties (see major comment 2).*

*The 'problem' with the downslope winds (having a stronger wind speed that over the lake, and (probably much) larger water vapor pres- sure deficit is mentioned - but only mentioned to yield a 'slight overestimation'. Based on what is this called 'slight'? Is it 10% (and would 10% be slight)? Or 50%? (but only occurring during 30% of the hours? – and what is then slight?). I think this would be a discussion.*

*Another point that apparently needs discussion is the fact that the radiation measurement does not 'see' the same surface as the authors want to probe with their EC system, i.e. the lake surface. In fact, I think that either all the aspects, which include Rn have either to be removed from the paper, or an estimate has to be made to establish a method to estimate Rn over water from measured Rn over land (see major comment 1). The discussion then, would consist of the associated uncertainty and the potential impact on the interpretation of the empirical methods (i.e. their performance). Given the relatively large uncertainty in the statistical model, a useful contribution to the discussion would be to test the empirical relations for only a subset of days (for the 1 day averaging period, say), for which the impact of the statistical regression model is minimal (only a few or no estimated evaporation hours, mostly measured values). If then the comparison to the 4 empirical methods (Tab 6) would be robust, this would indeed be an indication that the conclusions regarding the appropriateness of the empirical relations are supported by data (not the statistical model). This last discussion, of course, would only make sense if the 'radiation problem' had somehow been overcome. Finally, an important point for the discussion seems to be that the empirical estimates are relatively good 'average estimates' (28 days) – but do only have reduced predictive (diagnostic, that is) skill for short time scales*

A) Thank you for the detailed remarks about the discussions sections.

We agree that the discussion of the possible uncertainties of the applied method is probably too short. We revised this section and addressed also the 4 points of the reviewer:

1) The first point of the reviewer refers to C2 and if the regression model was used 'outside' its boundaries. As already discussed this was not the case and so an overestimation does not take place but we will discuss the

possible underestimation of evaporation, due to the fact that 'extreme' values of wind velocity and VDP were not used for calculations but instead a median value was used (see also answer to C2). Furthermore, we will discuss the question of overestimating evaporation due to the possibility that VDP decreases with increasing distance to the shoreline. As already mentioned in the answer to C2, in our opinion, the influence on the evaporation results is weak but nevertheless is discussed in the revised version of the paper.

2) Regarding the net radiation measurements, we now explain our methods for calculating net radiation in section 3.

3) We also want to thank the reviewer for the idea to test the empirical formulas on a subset of days to show that the regression model does not influence the results of the empirical relations. We did realise those test and it was found that still the aerodynamic approach V0 (R=0.96) and penman V6 (R=0.94) achieved the best results. However, we did not add the results to the paper as we think that the results are not fully comparable to the regressions for the whole data set (Fig. 6) as the number of days where only onshore wind conditions occurred and, thus, no or less than 2 values (1 h) are missing is limited to 13 days and these days are not equally distributed over the year. So this subset of days does not represent the seasonal variation, which obviously is important for the results.

4) The last point of the reviewer, refers to the reduced predictive skills of the empirical approaches on short time scales. We will discuss the reduced predictive skill of the empirical approaches, especially on the sub-daily time scales and that only the aerodynamic approach can reliably be applied to short time scales.

T) **Discussion and Conclusion**

 The eddy covariance method was used for the first-time, high resolution, direct evaporation measurements  of the Dead Sea One aim of this study was to present an applicable method to measure evaporation with a shoreline station. The measurement strategy was based on the installation of the station on a headland, which was surrounded by water from 320°. This setup at the shoreline was chosen to avoid influence on the measurements by raft motion and sea spray, where the latter one leads to a serious soiling of the instrument and influences data quality strongly. However, land based eddy covariance measurements have their limitations in measuring evaporation from the water surface, as part of the flux footprint is located over land. Therefore, a novel approach was presented to account for this limitations. A multiple regression model was  trained with the onshore wind and vapour pressure deficit  data and with this model lake evaporation for offshore wind conditions was calculated. With this method, 90 % of the missing evaporation data due to offshore wind conditions could be calculated. Considering the high amount of rejected data due to the fetch criteria in this and also in other works, 15-25 % (e.g. Mammarella et al., 2015; Nordbo et al., 2011) this approach can improve data availability considerably. The uncertainty due to this method is also small with a prediction error of the calculated values of only 4.8 % which makes it a very reliable method. However, there is still some uncertainty to this method which cannot be accounted for directly. On the one hand, extreme values of wind velocity and water vapour pressure deficit were not used to calculate evaporation when they were outside the model boundaries. This leads most likely to an underestimation of the actual evaporation amount. On the other hand, wind velocity and vapour pressure deficit  could decrease with increasing distance from the shoreline, which would lead to an overestimation of evaporation. The comparison with results from measurements in the middle of the lake (Weiss et al., 1988; Hecht and Gertman, 2003) shows that even in the middle of the lake westerly winds with hourly averaged velocities between 8 and 12 m s$^{-1}$ were observed. Also wind lidar measurements confirmed, that the westerly winds regularly reached several km over the lake without loosing its strength (Metzger, 2017). So offshore wind measurement seem representative for the calculation of evaporation. A decrease of vapour pressure deficit has to be considered, but is most likely small due to the following reasons. Firstly, the fetch of the station is quite limited with 600 m, and secondly the westerly winds are connected with high turbulence and thus strong vertical mixing (Metzger, 2017). From these results we conclude that the approach is also applicable to other

lakes, when the measured wind velocity and vapour pressure deficit value range is equally distributed between onshore and offshore conditions to appropriately train the model, and the fetch of the measurements is small enough that the shoreline measurements are representative for the fetch.

The second aim was to evaluate the diurnal and intra-annual  variability of Dead Sea evaporation.  The results of the measurements showed that the diurnal cycle of evaporation is ~~mainly driven by the diurnal cycle of the wind systems and their related wind velocities. This leads to maximum evaporation rates after sunset, caused by westerly winds with high wind velocities. These westerly winds occur from spring until autumn. The results are consistent with findings for Lake Kinneret, where these westerly winds also occur in the evening (Assouline and Mahrer, 1993; Shilo et al., 2015).However, the daily evaporation rates are notably lower compared to the evaporation at Lake Kinneret through the much higher salinity of the Dead Sea water and thus the reduced saturation vapour pressure. The median daily evaporation ranges from 1.1~~in phase with the wind velocity, which corresponds to findings of other studies in the Jordan valley (e.g. Assouline, 1993; Assouline et al., 2008) and that the strong westerly winds in the evening double evaporation compared to midday values. These findings are also important for other lakes, as there are many places with similar strong and dry wind systems (e.g. Bora, Tramontane, Mistral), which just occur on longer time scales. Bouin et al. (2012) already showed that the Tramontane in France even trebles evaporation from a lagoon compared to normal conditions. In respect to the ongoing climate change these results could motivate a regional study on the impact of climate change on the future evolution of thermally and orography induced wind systems in the Mediterranean region, as there is little information so far, but they are important for the future development of the water bodies. As already expected, evaporation from the Dead Sea is lower compared to other less or non-saline lakes. The ratio to Lake Kinneret, which is located under similar climatic conditions is 0.68 in summer, but only 0.83 in winter. This difference is most likely caused by the different climatic conditions in winter. Lake Kinneret receives a considerable amount of rainfall and more humid air masses as it is Mediterranean climate (Goldreich, 2003), whereas the Dead Sea is located in arid climate conditions where, even in winter, nearly no rainfall occurs. The annual evaporation was found to be $994\pm88.2$ mm. The uncertainty of $8.8\,\%$ results mostly from the gap filling procedure ($81.2$ mm ~~$\mathrm{d}^{-1}$ in July, but the absolute maximum of the measured daily evaporation rates was measured in November with $6.9\,\mathrm{mm\,d}^{-1}$. This is extremely high compared to the median values in winterand highlights the stronger synoptic influence on the region during the wet season (Bitan, 1974, 1976). One of the typical synoptic systems during the wet season is the Red Sea Trough, which can cause high wind velocities and high vapour pressure deficits in the valley and thus leads to very high evaporation rates. This is particularly important as Alpert et al. (2004) found that the frequency of such Red Sea Trough systems nearly doubled since the 1960s from 50 to 100 days per year. The total measured evaporation for the period 1 March 2014 until 1 March 2015 was $994.5 \pm 81.2$ mm, which agrees~~). It could thus be reduced by improving the system performance or by finding a better method to fill the gaps. However, the annual amount coincids well with previous findings such as Stanhill (1994) with 1005 mm a$^{-1}$ and is close to the results from Lensky et al. (2005) $(1100 - 1200$ mm a$^{-1})$, which both estimated the evaporation based on theoretical energy balance approaches. However, it is far away from the 2 m from Salameh and El-Naser (1999), who estimated evaporation based on water balance calculations, which could indicate uncertainties in the assessment of the water balance components. A certain degree of differences between the results is natural as the studies, considered different data sets for different years, which also means different salinities and different weather conditions.

 For the perspective affordable long-term assessment of evaporation different equations to calculate evaporation were tested on their applicability for the Dead Sea. The best suitable, and also the only method applicable on sub-daily time scales, is the aerodynamic approach. It was also shown that the consideration of the atmospheric stability in the calculations has an neglegible effect on the results. This again coincides with results for Lake Kinneret (Shilo et al., 2015; Rimmer et al., 2009) and makes this method easily applicable for evaporation calculations with data from a shoreline station. The other approaches are designed for longer time intervals and are not applicable for sub-daily calculations. The results also confirm the findings from various other studies (Rimmer et al., 2009; Giadrossich et al., 2015; Tanny et al., 2008; Rosenberry et al., 2007) that for the BREB, Priestley-Taylor and Penman method,  the knowledge of the heat storage term is essential to achieve reliable results, as neglecting the heat storage results in a strong seasonal bias

 Using estimates of the heat storage term does not provide acceptable results for the BREB and the Priestley-Taylor method either. For the Penman equation an applicable solution is achieved when ~~a linear function for the heat storage is empirically gained from the data set. We conclude that the BREB and Priestley-Taylor method can only be applied for the Dead Sea if heat storage is measured, which requires a raft station or ship measurements, or for long time periods, i. e. one year, where the heat storage term can be neglected. The Penman equation is applicable for the Dead Sea, if the heat storage is considered using the described approaches. The aerodynamic approach yields the best results with respect to the diurnal and intra-annual calculation of evaporation. They were in best agreement with the measurements. It was also shown that the consideration of the atmospheric stability in the calculations has an neglegible effect on the results. This again coincides with results for Lake Kinneret (Shilo et al., 2015; Rimmer et al., 2009) and makes this method easily applicable for evaporation calculations, as only wind velocity and vapour pressure deficit are required.~~
 using the empirically gained function for the heat storage. Thus, we conclude that the BREB or Priestley-Tayler method are not applicable with a shoreline station only, but the aerodynamic and the  adapted Penman method can be used, making expensive raft measurements expendable. ~~From the evaluation of the indirect methods we conclude that for a reliable estimate of the Dead Sea evaporation the aerodynamic method is advisable and that the influence of the atmospheric stability is negligible. Like the new model, the aerodynamic method connects evaporation with its governing variables, which are wind velocity and vapour pressure deficit, and allows the calculation of sub-daily or multi-day evaporation amounts without a seasonal bias. The advantage thereby is clearly the use of,which can be installed on the shoreline,values on a sub-daily time scale. This, and the resulting environmental changes on a longer time scale.~~ . The results can be implemented into hydrological models to thus study the water budget and its development in the future.

**3   Minor comments**

*P5, l.4 I don't think I have ever heard of a Rototronic sensor*
A) that was a typo. The company is called Rotronic.

*P5, l. 4 the height of measurement is not given for the radiation, precipitation and pressure sensors. Please specify.*
A) The measurement height for radiation is 2 m. Precipitation and pressure at 1 m height. It was added to the paper.
T) temperature and humidity at 2 m height (HC2S3, Rotronic), temperature at 6 m (100KGA1A, BetaTherm), longwave and shortwave radiation components of the upper and lower half space (CNR4, Kipp&Zonen) at 2 m height, precipitation (tipping bucket rain gauge 552202, Young), and atmospheric pressure (PTR330, Vaisala) at 1 m height.

*P5, l. 25 I agree with the present authors (in contrast to the other reviewer) that the appropriate conversion should be based on the water temperature and not the air temperature (the differences will be small, though). ¡w'a'¿ is already an energy flux (i.e., the kinematic flux of absolute humidity), which is only converted into energy units by multiplication with $L_v$. The corresponding energy (enthalpy) is calculated at the location where the process of evaporation takes place and the relevant temperature is the lake surface temperature (whether this is called $T_w$, the very top of the water or $T_s$, the very bottom of the air, is the same). I don't think that the 'constancy of the fluxes' in the SL is a valid argument for using the air temperature (at 6 m*

*agl in the present case) since the fluxes are only 'constant' within the SL, but not below (note that the SL has also a lower boundary, i.e. the laminar layer with a thickness of some millimeters over water wherein the turbulent fluxes are zero by definition – and rapidly change to their 'atmospheric surface value' at its top). The 'SL theory' (including the constant flux assumption) produces a temperature profile (e.g., eq. 11.12 in the text book of Arya (1988)), which is based on a 'surface temperature, T_o or T_s, which actually corresponds to the temperature at the height of the (thermal) roughness length. If we assume an (ideal) stratified SL, stable or unstable, and perfect measurements with a given 'surface' latent heat flux, each measurement height would have a (slightly) different L_v (because of the temperature profile) and hence a different latent heat flux – which is inconsistent with the 'constant flux'. In order to obtain the 'surface flux' from a measurement at any height (within the SL, of course), we must therefore relate the conversion into energy units to a common height, i.e. the thermal roughness length. The actual task therefore in determining L_v is to find the temperature at the height of the thermal roughness length. I think the water temperature is a better estimate for this than the temperature at measurement height. This 'surface latent heat flux' then can be used to assess how many mm of water had been evaporated (what is one of the primary goals in the present study). In any case, two comments have to be added: first, it is in fact potential temperature that has to be used (more precisely, virtual potential temperature) – again the differences (for a 6 m level) are negligible. Second, if L_v is estimated using eq. (3), latent heat fluxes will be in [kW m-2] (since L_v is is in [kJ kg-1]) and not comparable to the sensible heat fluxes from eq (2).*

A) Thank you for your comment. For better comparison of eq. 3 it's changed to $LE = L_v \cdot \overline{w'a'} \cdot 1000$

*P6, l.17 ... the mean vertical velocity: this could be mixed up with the mean vertical velocity over the averaging period that is zero in the 'double rotation approach'. In PF, it is the mean vertical velocity over the period that is used to define the plane.*

A) Thank you for the comment. We added '..over the period that is used to define the plane'.

T) It rotates the coordinate system to the main wind direction and then rotates the system around the y-axis, such that the z-axis is positioned perpendicular to the horizontal plan and that the mean vertical wind  over the period that is used to define the plane is $0\,\mathrm{m\,s^{-1}}$.

*P6, l.20...calculations at sites..*

A) changed accordingly.

T) This correction is particularly important for flux calculations  at sites with high humidity fluctuations, such as over the water surface.

*P6, l. 28 below 0.5 'what'? (units)*

A) 0.5 is 50%. This was changed.

T) Latent heat flux data were rejected when the signal strength of the radiation source to measure the water vapour was below 50%, when the variability of the signal from one from one 10 min average to the next one was higher than 0.6 % within  the 30 min time interval, and during precipitation events, as a disturbance of the water vapour measurements was expected for these conditions.

*P7, l.3 ...data, which...*

A) changed accordingly.

T) Class 1 to 6 describe data, which can be used for the analysis and classes 7 to 9 were rejected.

*P7, l. 11 is reasonably good*

A) changed accordingly.

T) This is  reasonably good...

*P7, l. 28 is built*

A) changed accordingly.

T) After each division a regression model is  built with the training data set and then applied on the data of the validation group.

*P9, l. 5 on longer time scales (or: on a longer time scale)*
A) changed accordingly.
T) On longer time  scales ...

*P8, l. 11ff The presentation of the methods to estimate evaporation is somewhat difficult to comprehend. I try to exemplify this for the Energy balance method. First, one is referred to Tab. 1. The first mentioned aspect of this method is, what is difficult to obtain. I then try to check (find) F_n in Tab 1 (which is apparently difficult to obtain). F_n does not appear in the given equation (but it is 'explained' below in the list of symbols – even if it is not present in any of the given equations). So, I cannot at least judge how this variable appears in the full equation or is related to other variables that do so. The same with the other neglected variables. The 'result' is called V0 (it has an 'X' in Tab 2). Estimating 'somehow' delta_Q produces then 'V2' (why is it V2 for this method, and V1 for the first?). Anyway, it also has an 'X' in Tab 2 (why not a V2?). And some of the other methods also have an 'X' in this table for delta_Q. Overall, I think the overview on the employed approaches should be presented in a much more concise manner. The reader should be able to judge what has actually been done.*

*P8, l. 15 to make the confusion complete, the 'third method' is then – not the third line in Tab 1 but the fourth…..*

A) Thank you for this remark. We revised this section it is now hopefully easier for the reader to follow and identify the different sensitivity tests which were performed. Table 1 was also changed and the original equations with all terms were added.

[revised manuscript text omitted]

*P10, l. 16 on about 57%...*
A) changed accordingly.
T) These downslope winds occurred  on about 57 % of the days in summer,...

*P10, l. 18 if the along-valley flow is northerly (with what I concur judging from Fig. 1) the lake breeze would be expected to be perpendicular (easterly on the western shore). Wouldn't this mean that, what was called a 'lake breeze' before (p10, l. 11) is rather a superposition of the along-valley flow and the lake breeze? Can the authors comment on that?*
A) We don't think that it is a superposition of the described nocturnal along-valley flow and the lake breeze, as the data showed the following: (1) When we analyse data of the station further inland and a little bit south of the shoreline station, wind during daytime has a much stronger easterly direction. (2) when we look at the third station (which is in the north) data shows a south-easterly flow during daytime. (3) Other studies, e.g. Bitan (1974, 1976); Alpert et al. (1990), all show a lake breeze development during the day, but depending on the

location of the station the exact wind direction of the lake breeze changes. Alpert et al. (1990) analyzed data in the south and found a northerly lake breeze, whereas Bitan (1974) in the north found a south-easterly flow (which corresponds to ()).
So in our opinion the north-easterly flow shown at the shoreline is not a superposition with an along valley flow but rather caused by the specific geographic location of the station, meaning the shape of the shoreline and the nearby orography which modulates the lake breeze.

*P10, l. 22 in the beginning*
A) changed accordingly.
T)  in the beginning of November

*P0, l. 25 on individual days*
A) changed accordingly.
T) However,  on individual days...

*P12, l. 4 on most of the days ...*
A) changed
T) Through the predominant local wind systems, these wind directions occur almost exclusively in the evening between 17:30 to 20:30 LT (LT=UTC+2) from spring until autumn (Fig. 3) and, thus,  most of the  data within this time frame are excluded.

*P14, l. 2 what does 'uncorrected' mean? No Webb correction, etc.? Or do the authors refer to 'only measured, no estimated (with the multiple regression model) values'? (same in Fig. 4). I think the term 'uncorrected' is not appropriate.*
A) uncorrected means only measured values. To avoid confusion we renamed it to "measured"
T) The comparison of the mean diurnal cycles of the measured fluxes  with the cycles including the calculated values for offshore wind conditions (corrected fluxes) shows that during the day the differences are small (Fig. 4).

*P15, l. 1 Maximum values are reached....: see major comment 2.*
A) as explained in the answer to comment 2, the values of the model should not overestimate evaporation, as it was only used in the reliable boundaries.
T) see answer to comment 2

*P15, l.11 the uncertainty...: so, how large was it found to be?*
A) The uncertainty due to the gap filling method was estimated using the corresponding MAD of the used timestep of the respective month. Overall the mean MAD for the different months varied between 0.019 and 0.029 mm 30 min$^{-1}$. For the total annual evaporation amount the uncertainty due to gap filling resulted in 81.2 mm
T) Summing the evaporation values over the whole measurement period results in a total amount of 994.5±88.2 mm, where 81.2 mm of the uncertainty result from the gap filling method and 7 mm due to the regression model.

*P16, l. 10 evaporation rates of 5.1 mm d-1,.... are measured: do these 'extreme days' contain any estimated values (using the statistical model)? How about the other 'large days' throughout the year?*
A) Only 3 out of the 72 values for these 3 'extreme' days had to be estimated as northerly winds prevailed. The estimated values were also not the maximum values reached on these days. It's similar for other large days.
T) This case was also used to test the performance of the regression model as on these three consecutive days only 3 out of 72 evaporation values had to be calculated due to the fetch criteria. Applying the regression model to calculate evaporation on these 3 days completely yields good results for day one and three were the difference was only 4-5 % but it also shows the potential underestimation of extreme evaporation rates as the model underestimated the daily evaporation on the second day by 18 %

*P16, l. 14 not shown: I think that this 'case' (if it were shown) could serve as a good example to gain some confidence in the statistical model (see major comment 2). Tab 6 'MD' must be defined.*
A) Thank you for this comment. Indeed this event serves as a good example to show the performance of the

[Figure]

Figure 1: Evaporation, water vapour deficit and wind conditions for 5 to 11 November 2014.

statistical model. As mentioned in the previous comment on these 3 days only 3 out of the 72 evaporation values were estimated. When the model is applied to estimate the evaporation of these days, we get quite a good agreement for day 1 and 3 the model underestimates evaporation by only 4% and 5%. But we also see that the model potentially underestimates the extreme values as on day 2 the value was underestimated by 18% (see Fig. 1 in this document). We added the information about this case to the paper.
T) see answer to comment before

*P17, l. 16 MD is probably mean difference, right? Anyway, the mean differences are not given in the table to demonstrate this (for V0 they are essentially the same …).*
A) Thank you for this comment. MD is mean difference. The mean difference for the 30 min interval is only given in the text (P.17, l.14) and not in the table to keep the table clear and better readable. To make the comparison easier we added the value of the mean difference for the 1d interval to the text.
T) The correlation coefficients vary between 0.94 for 1 d intervals and 0.99 for 28 d intervals, mean differences are smaller, 0.02±0.54 mm d$^{-1}$ for 1 d intervals, and the slopes of the regression lines vary around 1.10 (Table 1, Fig. 6 a,V0).

*P17, l. 20 I suggest to start a new paragraph for the BREB method.*
A) changed accordingly.

*P17, l. 25 the largest*
A) changed accordingly.
T) and the  largest offset

*P17, l. 30 22%: judging from Fig. 7 this number is probably valid for a 28 d averaging period* A) Yes that's right. It was added to the text to make it clear and added another explanation, that the numbers for the other averaging times are comparable.
T) …Compared to the measured values, this results in  an overestimation of the annual evaporation amount by 22 %, calculated from the 28 d averages. For the other time intervals the overestimation of the annual evaporation amount was comparable and is therefore not shown.

*p17, l. 31 coefficients*
A) changed accordingly.
T) Correlation  coefficients are better…

*p17, l. 31 improved the results: I do not really agree. Indeed, the correlation coefficients do somewhat increase (but look at Fig. 6 – both versions would probably serve as examples for 'statistics 101' students for data sets, for which a linear model is not appropriate). At the same time, slope and offset are getting worse (this is why we usually use different statistical measures....). In my view the results of V2 (as compared to V0) simply demonstrate that the calculation of heat storage is not appropriate (see major comment 1)*

A) we revised this description.

T) Correlation  coefficients are better and the mean differences are reduced (Table 1). However, the slope and offset shows that the heat storage term is still not represented correctly. The slopes and the offsets indicate an overestimation of the small evaporation amounts and an underestimation of the high amounts (Fig.7 b,V2).

*p17, l. 34 11%: same as above (and also in the following) these values seem to apply for the 28 day averages*

A) That's right, please see also answer to comment P17,l.30

*p19, l. 7 up to 100%: I don't think I can see this from Fig. 7d*

A) Thank you for pointing this out. Fig. 7d has to be compared to the measured daily evaporation amounts in Fig. 5. We added the necessary reference.

T) Evaporation values are strongly overestimated from spring until autumn (Fig. 6 d,V0), exceeding the measured daily evaporation amounts by up to 100 % (compare Fig. 7 d,V0 to Fig. 5).

*p20, l. 16 with a heat storage term....: in fact, I only now understand what actually V6 does: it fits the Penman equation to account for the missing storage term, right? So, how is this fitting process being done? If it is fitted – as I assume – using the measured evaporation, it does not come as a surprise that a negligible mean difference results. The results from this exercise have to discussed in this light (major comment 3).*

A) Thank you for the comment. For the fitting it was assumed that the heat storage term equals the difference between the calculated evaporation amounts using the Penman equation V4 and the measured evaporation amounts and that it can be described as a linear function of the net radiation $\Delta Q = a \cdot Rn$. With this assumption the coefficient 'a' was derived and averaged over the measurement period, which resulted in $\Delta Q = 0.77 Rn$.

*p21, l. 8 due to the much higher*

A) sentence is not longer in the discussion.

*p22, l. 12 The BREB, Priestly-Taylor and Penman... All these methods do not employ radiation (which has a different field of view – and does not represent the water surface, see major comment 1). I would hypothesize that this is, in the first place, the reason for their bad performance. Only if the Penman method is fitted to the data, it can also produce some reasonable results.*

A) Thank you for the comment. As already explained for comment 1, we considered the different field of view of the radiation measurements. We still think that the strongest influence is the correct representation of the heat storage term, but of course there are uncertainties connected to the calculation of the net radiation.

(jutta.metzger@kit.edu)

*The paper shows, for the first time, results of direct annual evaporation (E) measurements from the Dead Sea (DS) based on eddy covariance (EC) technique. Understanding the annual and the short-term dynamic of the lake evaporation rate is important scientifically in many aspects, for the regional managers and for the future fate of the whole region. The paper is a clearly written, covering both measurements aspects and evaporation modelling aspects over free water body in exceptional conditions, and one can assume that the measurements were carried under very harsh conditions. Last, there are not many E measurements over water bodies that are based on eddy covariance technique and are comparing measurements results versus different evaporation rate models as the Authors presented here. Having said that, there are a few significant points the Authors need to address before any publications.*

The authors thank the reviewer for the insightful review. They helped to improve the paper. Responses to individual comments are provided below. Reviewer's comments are in italic.

**1 Major Comments**

*C1) Comparing annual evaporation results with previous estimation. Comparing to previous works need caution which the Authors have to mention and discuss, including; A. The change in the water level likely changed as well the DS surface area between the different estimation years (e.g., in the case of Stanhill 1994 the lake level was probably 30 m higher and surface area much larger). B. Changes of the climatic conditions due to large-scale changes as well as due to the lake shrinkage. The Authors already mentioned the rapid changes in the regional Persian trough frequency. C. Likely salinity changes over the years and possibly also the amounts of water removal to the mineral production pools in those years? And D. This work is based on a single measurement year that the Authors mentioned as a relatively wet one*

A) Thank you for this comment. We agree that there are differences between the conditions of the former studies and our study. Climate and weather conditions, salinity and water level obviously influence the evaporation rate. We will point this out in the discussion and take your remarks into consideration when comparing our annual evaporation rate with former studies. Nevertheless, our main goal was not to rate former results of yearly evaporation amounts. Our main aim was to provide information on the short-term and intra-annual variability of evaporation as this was so far not provided by other studies and to provide a measurement concept and post-processing methods with which evaporation measurements at the shoreline can be realised.
T) The annual amount coincides well with previous findings such as Stanhill (1994) with $1005\,\text{mm}\,\text{a}^{-1}$ and is close to the results from Lensky et al. (2005) ($1100 - 1200\,\text{mm}\,\text{a}^{-1}$), which both estimated the evaporation based on theoretical energy balance approaches.  A certain degree of differences between the results is inevitable as the studies, considered different data sets and different time periods, meaning different salinities and different weather conditions. However, the measurements are far away from the $2000\,\text{mm}$ from Salameh and El-Naser (1999), who estimated evaporation based on water balance calculations, which could indicate uncertainties in the assessment of the water balance components.

*C2) H and L_v calculations (section 3.1) were needed for the energy budget models (as in Tab1). And I assume, though not clearly presented, that ET was derived directly from EC evapotranspiration calculation, not from L_v? However, figure A1 is important in showing that compared with pure water, saline water L_v*

*is lower for temperature higher than 22C , which likely means that for most times of the year L_v of Dead Sea water is lower than that of a pure water. In this respect, the sentence in L27, page P5 is confusing and future warming and increase water salinity will possibly increase E?*

Yes, the evaporation was derived directly from the EC measurements.
If future warming and an increase in water salinity will increase evaporation can not be concluded from Fig. A1. These measurements were only conducted for the given salinity of the Dead Sea in 2014 and do not capture changes through salinity increase. There are two factors determining the L_v: 1) Temperature. It's correct that the data show that L_v of the Dead Sea is most of the time lower than L_v of pure water in case of water temperatures larger 22°C. 2) Water salinity. It can be assumed that if water salinity increases, L_v increases as well (Salhotra et al., 1987). Yet, it is unkown how these two factors will balance each other in the future. For conclusions about the future development of the evaporation due to L_v changes studies with different salinities, also considering the chemical composition of the Dead Sea water, have to be conducted.

*C3) Gap filling model for E values when wind direction is coming from the land enhances considerably the total evaporation, especially during the afternoons. However, this model uses VPD (and wind speed) derived from humidity values of air coming from the lake. While the humidity of the land air is probably lower compared to wind coming from the lake. But, it is likely that RH of this dry air increases as it is blowing over the lake for some distance., Thus VPD and E should decrease. Shouldn't such effects be estimated, considering its large effect on E? Do the Authors have any information on the RH difference between the two sides of the lake (e.g., west vs. east) for wind blowing to either directions?*

A) The effect of a possible VDP decrease with increasing distance from the shoreline couldn't be directly estimated, but we made following observations: 1) We have seen in the data that the strong westerly winds are connected with high turbulence, and even rotor formation was observed. This means that vertical mixing and air mass exchange is enhanced and thus VDP decrease should be low. 2) The fetch of the station is around 600 m. In our opinion the decrease of delta_e within such a distance is not very strong considering the turbulent mixing. 3) Evaporation has a stronger dependence on wind velocity than on VDP which makes the influence of VDP variations on the results weaker.
The second question of the reviewer was if we have data from both sides of the lake. We don't have information on the VDP variation over the lake or from the eastern shore to validate our assumption. We agree with the reviewer that VDP variations are an uncertainty, but as stated in the above paragraph we don't think that it has an large effect. We discussed this uncertainty in the new discussion section:
T) On the other hand, wind velocity and vapour pressure deficit could decrease with increasing distance from the shoreline, which would lead to an overestimation of evaporation. However, the comparison with results from measurements in the middle of the lake (Weiss et al., 1988; Hecht and Gertman, 2003) shows that even in the middle of the lake westerly winds with hourly averaged velocities between 8 and $12\,\mathrm{m\,s^{-1}}$ were observed. Wind lidar measurements confirmed, that the westerly winds regularly reach several km over the lake without loosing their strength (Metzger, 2017). In conclusion, offshore wind measurement seem representative for lake conditions and reasonable for the calculation of evaporation. A decrease of vapour pressure deficit has to be considered, but is most likely small for the following reasons. Firstly, the fetch of the station is limited with 600 m, meaning that the distance the air mass passes over the water is short. Secondly, the westerly winds are connected with high turbulence and, thus, strong vertical mixing (Metzger, 2017). From these results we conclude that the approach is also applicable to other lakes, in case the measured onshore wind velocity and vapour pressure deficit values are representative for offshore conditions to appropriately train the model, and the fetch of the flux measurements is small enough that the meteorological measurements at the shoreline are representative for the fetch.

*C4) Combining or incorporating variables with previous works that have been carried out over the DS in the past to check estimations and assumptions. For example, I found published works on DS surface temperature (Tom) measurements, and others on the lake heat storage on different time scales. I am wondering why the Authors did not refer to this data? Δe is highly dependent on Tom and close to the shore Tom is warmer than in the open sea, thus it would be valuable if the authors could compare their estimations with independent measurements and its effects on E estimation.*

A) Thank you for your comment. We used the published works on DS surface temperature as a basis for our method. E.g. Nehorai et al. (2013), showed that "SST is highly correlated to air temperature (0.93-0.98) in all seasons". Based on these previous results we used the Monin-Obukhov approach to calculate DS surface temperature from air temperature. Concerning the heat storage we could not find suitable data to directly use as input for our calculations and a direct comparison to independent measurements over the open sea is unfortunately not possible, as we don't have such measurements or access to such data sets. We added some explanation to Sec. 3.1:

T) For the surface water temperature, $T_S$, no in-situ measurements were available. Also remotely sensed surface water temperature products could not be used as operational SST algorithms are calibrated to mean sea level and do not take the additional 421 m atmospheric layer in the Dead Sea valley into account. Nehorai et al. (2009) showed that a calibration of satellite data with in-situ measurements is necessary. Furthermore, Nehorai et al. (2009) raised concerns that enhanced water vapour input into the atmosphere through evaporation causes stronger absorption of thermal IR radiation, leading to a screening of the Dead Sea surface and, thus, incorrect estimates of the surface water temperature. Based on the results of Nehorai et al. (2013), which showed that "SST is highly correlated to air temperature ($R^2 = 0.93 - 0.98$) in all seasons", the Monin-Obukhov similarity approach was used to calculate surface water temperature from the measured air temperature (see Appendix A), and is further on referred to as $T_{MO}$.

*5. This leads to the last main point: The basis for the uncertainty around E ( 82.2 mm) is unclear. For ecosystems over land, it is generally assume to be 10%; is it about the same here or? However, although the uncertainty value is about 8% of E it is likely still a substantial large number for water management of the region. Can Authors suggest ways to reduce this in future activities?*

A) The uncertainty of the total evaporation amount (88.2 mm) contains the uncertainty due to the gap filling method. Gaps were filled using the median evaporation of the corresponding time step of the respective month and the uncertainty of this method was estimated using the Median absolute deviation (MAD) for the used time step (described in Sec. 4.3). The uncertainty due to the gap filling procedure accounts for 81.2 mm of the 88.2 mm uncertainty. The rest of the uncertainty (7 mm) stems from the regression model. Here, the prediction error given by the MCCV was used. To account for the highest possible uncertainty of the regression model the prediction error of the MCCV with randomly chosen validation sectors $er_s = 4.79\%$ was used.

The uncertainty could further be reduced by finding another method to fill the gaps. The use of a median evaporation cycle naturally results in relatively large MADs, as evaporation varies from day to day. Nevertheless, we choose this method instead of the often used interpolation, as interpolating would in some cases not depict the real diurnal cycle. E.g. if we would use linear interpolation between the 18 LT and 24 LT value in Fig. 4 we would completely miss the diurnal maximum. Another way to reduce the uncertainty would be reducing the gaps in the data set itself. Gaps in the data set were caused by: malfunction of the system (2.4% missing data), precipitation events and problems with the radiation source (signal strength was too low, 10%), and quality control (integral turbulence characteristics and steady state test after Foken (1999), 9.2%). For example, shorter maintenance intervals could reduce the amount of data missing through system malfunction or problems with the radiation source, which were mainly caused by the very harsh conditions (see also answer to minor comment 2). However, precipitation events or conditions were the criteria of the EC method (fully developed turbulence and steady state) are not fulfilled can not be controlled. We also added this information to the discussion section:

T) The annual Dead Sea evaporation was found to be 994±88.2 mm for the measurement period. The uncertainty of 8.8 % results mostly from the gap filling procedure (81.2 mm) and not from the regression model. As gaps result from system malfunction or bad data quality, the uncertainty can be reduced by improving the system performance or by finding a better method to fill the gaps.

**2  Detailed Comments**

*1. L. 9 p. 3. I would look for additional citation(s) for the EC approach reliability to measure E over water bodies.*

A) added further citation.

T) The eddy covariance technique is the only method to obtain direct evaporation measurements, in high temporal resolution, which can be linked to meteorological variables afterwards. It . Thus, it is considered the most accurate and reliable method to estimate evaporation (Rimmer et al., 2009). (Rimmer et al., 2009; Tanny et al., 2008). All other methods assess evaporation indirectly, which means that all measurement errors accumulate into the estimated evaporation (Assouline and Mahrer, 1993)

*2.Is the IRGASON a close or open path IRGA? And generally, did the researcher had any problems with the presumable high rusty environment down there, with salt particles etc.?*

T) It is an open path instrument. Through the harsh environment the windows of the IRGASON got dirty in a short time, which influences the signal strength of the radiation source. Through these conditions short maintenance intervals of about 3-4 weeks were necessary.

*3.Heat storage in section 4; can the Authors add 'zero' line in Figure 2, $\Delta Q$ value. The impression from inspecting that figure is that the annual value deviate considerably from zero? Is it due to negative heat transfer (e.g., by rain)?*

T) added the zero line to Figure 2. Yes, there is a positive net heat storage when summing over the whole year. Several other studies like (Stanhill, 1990; Anati et al., 1987) already documented this and the thereby caused steady increase of the lake temperature over the last couple of years is also documented (e.g Hecht and Gertman (2003)).

*4.Please add the units for MD and std in Table 6.*

T) added accordingly

(jutta.metzger@kit.edu)

*The paper contains very important information on evaporation from the Dead Sea that should be eventually published for the benefit of the scientific and water management communities. The paper presents measured heat fluxes using eddy covariance sys- tem over a year; the eddy covariance data is presented with a solid data analysis. In addition the paper evaluates the validity of common used indirect estimations of evap- oration from the Dead Sea, which is very important for extending the flux estimates when EC is not available. However I think that at the present form the paper cannot be published in HESS due to the following reasons:*

*The authors state that (end of P3) "To measure the energy budget components of the water surface, a fully equipped energy balance station was installed … ", however this is not the case. Using eddy covariance - latent and sensible heat are measured directly, but heat storage is not measured nor the net radiation. The radiometer in use is a 4 components CNR4 (P5 L4-5), but as can be seen in Fig 1b the two lower half spaces are directed to the ground not to water surface; this is also acknowledged later on, but this should clearly be stated upfront. Water temperature is calculated not measured, this is found later on in the paper, should appear upfront when declaring for fully equipped energy balance station. In the results it is stated that the heat storage is calculated as the residuum of the energy balance (Rn-LE-H), where Rn by itself is not really measured, again, this is not a fully equipped system. I think the paper should be written with emphasis on the existing data, which are very important and worth publication, but not declaring for measuring energy balance, this gives a wrong impression for the reader. As stated above, water temperature was calculated, not measured, termed TMO. TMO is used for the examining the equations of evaporation versus measured evaporation (e.g. table 1 and large portions of the figures). TMO is also used to determine the saturation vapor pressure at water surface temperature, Ew. Ew is used to determine the vapor pressure deficit and for the evaporation estimates. So both water temperature and Ew are not measured, but they have a very important role in the analysis and the conclusions of the paper. Again, I this think that it would be better to orient the paper to the existing information that was measured and analyzed and not to rely so much on the computed meteorological parameters. Typically, when EC measured evaporation is compared with the evaporation equations it is done based on measured meteorological parameters, not on computed parameters. Overall, due to these weaknesses, there is a gap between the actual measurements and the interpretations and conclusions. The scientific methods and assumptions should be better declared earlier and should appear clearer. I think that the first half of the title "Dead Sea evaporation by eddy covariance measurements" is good and rep- resenting the novel aspects of this work, but the second half of the title represents the weaker part of the paper*

The authors thank the reviewer for the insightful review.
The reviewer raised 5 important points: Firstly, a fundamental statement that the comparison to the indirect methods has significant weaknesses, and furthermore four points related to specific parts of the content: 1) There is no fully equipped energy balance station. 2) TMO, which is important in the analysis is not measured but calculated and then used for the calculation of Ew and the vapour pressure deficit. 3) The analysis relies too much on calculated meteorological variables. 4) Methods and assumptions should be better declared.
The responses to the reviewers comments are provided here:

We understand the concerns of the reviewer regarding the indirect methods. There are uncertainties in the calculations of the indirect methods, but as EC measurements are often not available for the long-term assessment of evaporation it was one of the main goals and thus an important part of the paper to provide a reliable and scientifically sound method for long-term flux estimates in the future. Additionally, the results can serve as a

method to calculate the spatial variability of evaporation over the water surface, when using it with an appropriate model, which delivers high resolution information of wind velocity and vapour pressure deficit.

With the first part of the paper alone we would of course publish important and novel data concerning the evaporation of the Dead Sea, but the second part provides an important basis for future studies and work on the topic of Dead Sea evaporation and evaporation of similar land-water configurations. We therefore think that the comparison to the indirect methods should remain part of the paper.

1) With respect to the comment about the energy balance station, we agree with the reviewer that the sentence on P3: "...a fully equipped energy balance station was installed" can confuse the reader, as not all energy balance components of the water surface were directly measured. We changed this sentence and added a paragraph with an explanation what components of the energy balance of the water surface were actually measured and how we calculated the missing variables, like the reflected shortwave and outgoing longwave radiation, and the water surface temperature.

[revised manuscript text omitted]

2) + 3) both refer to the calculation of TMO and further meteorological variables.

One main goal of this paper was to provide a measurement concept and possible post-processing methods, which includes the common problem of assessing missing variables, with which evaporation measurements at the shoreline can be realised. The problem that not all necessary variables for an analysis are measured is a common problem in the assessment of evaporation (e.g. Lensky et al. (2005) used bulk formulas to estimate longwave radiation, Giadrossich et al. (2015) used a model to estimate stream discharge to the lake). Therefore, we don't see the calculation of e.g. TMO as a weakness of this paper but as part of the possible methods to gain evaporation data from a station on land.

One option to derive the surface water temperature (SST) from satellite data, as suggested from Reviewer 1, was also not possible because of the following reasons:

- Nehorai et al. (2009) used MeteoSat Second Generation data to estimate SST from the Dead Sea. For retrieving the SST the operational SST algorithms could not be applied as they are calibrated to sea level and do not take the additional 421 m atmospheric layer in the Dead Sea valley into account. They derived the SST by calibrating their algorithm against in-situ measurements. As we did not have the necessary in-situ measurements of the SST we could not follow their procedure to derive the SST from satellite data.

- Furthermore, Nehorai et al. (2009) raised concerns, that on days where the Mediterranean Sea Breeze enters the valley in the afternoon the enhanced evaporation causes enhanced water vapour and thus a stronger absorption of thermal IR radiation which leads to a screening of the Dead Sea surface and thus incorrect estimates of the SST. For their studies they excluded all data with these conditions. This would lead to data gaps in the time series of SST especially during offshore wind conditions, meaning that no SST data would be available for the timesteps where it is needed as an input parameter for the regression model to calculate evaporation from the water surface.

- Another point why satellite data was not used is the need of a continuous time series. Satellite data can not be used for cloudy conditions. So especially for the winter months cloud cover would reduce data availability significantly.

Because of the aforementioned problems using satellite data, we followed the advice of another paper from Nehorai et al. (2013), which shows that "SST is highly correlated to air temperature (0.93-0.98) in all seasons". Based on these results we used the similarity approach to calculate surface temperature from air temperature.

4) regarding the methods and assumptions we want to refer to answer 1). We will add a better description of the objectives, assumption and methods used to the introduction and section 3 "Data and Methods". We will e.g. explain what data was not measured and how it was calculated (e.g. the calculation of the net radiation.)

**References**

[revised manuscript text omitted]